# GENERALIZATION ERROR OF SPECTRAL ALGORITHMS

**Maksim Velikanov**[1,2], **Maxim Panov**[3], **Dmitry Yarotsky**[4]
[1]Technology Innovation Institute, [2]Ecole Polytechnique, [3]MBZUAI, [4]Skoltech
`maksim.velikanov@tii.ae, maxim.panov@mbzuai.ac.ae, d.yarotsky@skoltech.ru`

## ABSTRACT

The asymptotically precise estimation of the generalization of kernel methods has recently received attention due to the parallels between neural networks and their associated kernels. However, prior works derive such estimates for training by kernel ridge regression (KRR), whereas neural networks are typically trained with gradient descent (GD). In the present work, we consider the training of kernels with a family of *spectral algorithms* specified by profile $h(\lambda)$, and including KRR and GD as special cases. Then, we derive the generalization error as a functional of learning profile $h(\lambda)$ for two data models: high-dimensional Gaussian and low-dimensional translation-invariant model. Under power-law assumptions on the spectrum of the kernel and target, we use our framework to (i) give full loss asymptotics for both noisy and noiseless observations (ii) show that the loss localizes on certain spectral scales, giving a new perspective on the KRR saturation phenomenon (iii) conjecture, and demonstrate for the considered data models, the universality of the loss w.r.t. non-spectral details of the problem, but only in case of noisy observation.

## 1 INTRODUCTION

Quantitative description of various aspects of neural networks, most notably, generalization performance after training, is an important but challenging question of deep learning theory. One of the central approaches to this question is built on the connection between neural networks and its neural tangent kernel, first established for infinitely wide networks (Jacot et al., 2018; Lee et al., 2020; Chizat et al., 2019), and then further taken to the rich realm of finite practical networks (Fort et al., 2020; Maddox et al., 2021; Long, 2021; Kopitkov & Indelman, 2020; Vyas et al., 2023).

Consider a task of learning target function $f^*(\mathbf{x})$ from training dataset $\mathcal{D}_N = \{\mathbf{x}_i\}_{i=1}^N$ and (possibly) noisy observation $y_i = f^*(\mathbf{x}_i) + \sigma\varepsilon_i$, $\varepsilon_i \sim \mathcal{N}(0,1)$, using given kernel $K(\mathbf{x}, \mathbf{x}')$. Then, many authors (Bordelon et al., 2020; Jacot et al., 2020a; Wei et al., 2022) derive asymptotic $N \to \infty$ generalization error for kernel ridge regression (KRR) algorithm with regularization $\eta$:

$$L_{\mathrm{KRR}}(\eta) = \frac{1}{2} \frac{\partial \eta_{\mathrm{eff}}}{\partial \eta} \sum_l \frac{(\eta_{\mathrm{eff}} c_l)^2 + \lambda_l^2 \frac{\sigma^2}{N}}{(\eta_{\mathrm{eff}} + \lambda_l)^2}, \qquad 1 = \frac{\eta}{\eta_{\mathrm{eff}}} + \frac{1}{N} \sum_l \frac{\lambda_l}{\lambda_l + \eta_{\mathrm{eff}}}, \qquad (1)$$

where $\lambda_l$ are population eigenvalues of $K(\mathbf{x}, \mathbf{x}')$ and $c_l$ are respective eigencoefficients of $f^*(\mathbf{x})$ (see definition in (4)). The main motivation of our work is what happens with (1) when, as required by association with neural networks, KRR is replaced with GD or an even more general learning algorithm.

Importantly, a result of the type (1) may give precise insights for the family of power-law spectral distributions, closely related to *capacity* and *source* assumptions of non-parametric statistics:

$$\lambda_l \sim l^{-\nu} \text{ (with } \nu > 1\text{)}, \quad c_l^2 \sim l^{-\kappa-1} \text{ (with } \kappa > 0\text{)}. \qquad (2)$$

Power-law conditions (2) exhibit a rich picture of convergence rates. For the case of noisy observations $\sigma^2 > 0$, (Caponnetto & De Vito, 2007) gives minimax rate $O(N^{-\frac{\kappa}{\kappa+1}})$. For noiseless observations $\sigma^2 = 0$, the optimal estimation rate significantly improves (Bordelon et al., 2020; Cui et al., 2021) becoming $O(N^{-\kappa})$. However, the optimal rates are not always achievable for some classical algorithms. For example, in the case when $\frac{\kappa}{\nu} > 2$ in the noisy case the rate of the KRR becomes

| | Noisy observations $\sigma^2 > 0$ | | Noiseless observations |
| --- | --- | --- | --- |
| | KRR | GF & Optimal | KRR & GF & Optimal |
| Exponent in $L = O(N^{-\#})$ | $\frac{\kappa}{\kappa+1}$ $\Big\vert$ $\frac{2\nu}{2\nu+1}$ | $\frac{\kappa}{\kappa+1}$ | $\kappa \ \big\vert \ 2\nu$ |
| Spectral localization scale $s$ | $\frac{\nu}{\kappa+1}$ $\Big\vert$ $\left\{0, \frac{\nu}{2\nu+1}\right\}$ | $\frac{\nu}{\kappa+1}$ | $\nu \ \big\vert \ 0$ |
| Universality | yes | yes | no |

Table 1: Our results for power-law spectral distributions (2) and three algorithms: optimally regularized KRR, optimally stopped Gradient Flow (GF), and the *optimal* algorithm (see Section 3). For quantities exhibiting saturation at $\frac{\kappa}{\nu} = 2$, the vertical line $\cdot\big\vert\cdot$ separates saturated and non-saturated values. The spectral localization at scale $s$ means that the error is accumulated at eigenvalues $\lambda$ of the order $N^{-s}$ (see Section 5.1). In the last line, by universality, we mean the asymptotic equality of the errors for different problems with the same population spectrum $\lambda_l, c_l$. While we show the universality only for our two data models, we would expect it to hold for a broader class of data models.

$O(N^{-\frac{2\nu}{2\nu+1}})$, i.e. KRR doesn't attain the minimax lower bound (Li et al., 2023). Such an effect is usually called *saturation* and is well-known in various non-parametric problems (Mathé, 2004; Bauer et al., 2007). However, the saturation effect can be removed with algorithms other than KRR, for example spectral cut-off (Bauer et al., 2007) or gradient descent (Pillaud-Vivien et al., 2018). In noiseless case, (Bordelon et al., 2020; Cui et al., 2021) show saturation at the same point $\frac{\kappa}{\nu} > 2$, with the rate changing to $O(N^{-2\nu})$. Whether noiseless saturation can be removed by algorithms other than KRR, to the best of our knowledge, was not studied in the literature.

*Our contribution.* In this work, we augment the above picture in several directions, as summarized in Table 1 and the following three blocks

**Loss functional.** In Section 3, we introduce a new kernel learning framework by specifying the learning algorithm with a spectral profile $h(\lambda)$ and expressing the generalization error as a quadratic functional of this profile. While specific choices of $h(\lambda)$ can give KRR, Gradient Flow (GF), or any iterative first-order algorithm, we go beyond these classical examples and consider the *optimal* learning algorithm as the minimizer of the loss functional for a given problem.

**The models.** As the loss functional is problem-specific, we consider two different models: a Wishart-type model with Gaussian features and a translational-invariant model on a circle. In Section 4, we derive loss functionals for these models. In addition, we introduce a simple *Naive Model for Noisy Observations* (NMNO). In the presence of observations noise, for reasonable learning algorithms, and at least for power-law spectrum, NMNO model gives asymptotically exact loss values for both Wishart and Circle models despite their differences. This suggests a possible universality property for a larger family of problems, including our Wishart and Circle models as special cases.

**Results for power-law spectrum.** In Section 5, we reach specific conclusions by restricting both kernel eigenvalues and the target function eigencoefficients to be described by power-laws (2).

- For both noisy and noiseless observations, we derive full loss asymptotics. While the resulting rates are indeed consistent with the prior works, precise characterization of leading order asymptotic term gives access to finer details, such as the shape $h^*(\lambda)$ of the optimal learning algorithm.

- We introduce the notion of *spectral localization* - the scale of kernel eigenvalues over which the loss is accumulated - and quantify it for algorithms under consideration. In particular, this perspective provides a simple explanation of KRR *saturation* phenomenon as the inability to sufficiently fit the main features of the target function.

- By characterizing the shape of the optimal algorithm, we point to the cases where it is optimal to *overlearn* the training data, akin to KRR with negative regularization. Moreover, we specify the values of power-law exponents at which overlearning is beneficial.

## 2 SETTING

**Generalization error.** We evaluate the estimator $\widehat{f}(\mathbf{x})$ with its squared prediction error $|\widehat{f}(\mathbf{x}) - f^*(\mathbf{x})|^2$, averaged over inputs $\mathbf{x}$ drawn from a population density $p(\mathbf{x})$, and then all the randomness in training dataset $\mathcal{D}_N$:

$$L_{\widehat{f}} = \frac{1}{2}\mathbb{E}_{\mathcal{D}_N, \boldsymbol{\varepsilon}}\big\|\widehat{f} - f^*\big\|^2 = \frac{1}{2}\int \mathbb{E}_{\mathcal{D}_N, \boldsymbol{\varepsilon}}\Big[\big(\widehat{f}(\mathbf{x}) - f^*(\mathbf{x})\big)^2\Big]p(\mathbf{x})d\mathbf{x}, \tag{3}$$

where $\boldsymbol{\varepsilon} = (\varepsilon_1, \ldots, \varepsilon_N) \sim \mathcal{N}(\mathbf{0}, I)$, $\|f\|^2 \equiv \langle f, f \rangle$, and angle brackets denote the scalar product $\langle f, g \rangle \equiv \int f(\mathbf{x})g(\mathbf{x})p(\mathbf{x})d\mathbf{x}$.

**Population spectrum.** Central to our approach is the connection between generalization error $L_{\widehat{f}}$ and spectral distributions $\lambda_l, c_l$ of the problem, defined by Mercer theorem as

$$K(\mathbf{x}, \mathbf{x}') = \sum \lambda_l \phi_l(\mathbf{x})\phi_l(\mathbf{x}'), \quad f^*(\mathbf{x}) = \sum c_l \phi_l(\mathbf{x}), \quad \langle \phi_l, \phi_{l'} \rangle = \delta_{ll'}. \tag{4}$$

Here, $\lambda_l$ are the kernel eigenvalues, $c_l$ are the target function coefficients, and $\phi_l(\mathbf{x})$ are the eigen-features of the kernel. In the most interesting scenarios, the number $P$ of features $\phi_l$ is infinite, and the respective eigenvalues $\lambda_l \to 0$ as $l \to \infty$.

In this work, we aim at two levels of results w.r.t. the population spectrum $\lambda_l, c_l$. First, we want to obtain a characterization of $L_{\widehat{f}}$ for a general $\lambda_l, c_l$, similar to what is done in the classic result (1). Second, we will assume the power-laws (2) to obtain a more detailed description of the generalization error $L_{\widehat{f}}$.

An important object is the empirical kernel matrix $\mathbf{K} \in \mathbb{R}^{N \times N}$ composed of evaluation of the kernel on training points $(\mathbf{K})_{ij} = K(\mathbf{x}_i, \mathbf{x}_j)$. Let us additionally denote by $\mathbf{y} = \mathbf{f}^* + \boldsymbol{\varepsilon} \in \mathbb{R}^N$ the observation vector with components $(\mathbf{y})_i = f^*(\mathbf{x}_i) + \varepsilon_i$; by $\boldsymbol{\Lambda} \in \mathbb{R}^{P \times P}$ the diagonal matrix with $(\boldsymbol{\Lambda})_{ll} = \lambda_l$; and by $\boldsymbol{\Phi} \in \mathbb{R}^{P \times N}$ the matrix of kernel features evaluated at training points, $(\boldsymbol{\Phi})_{li} = \phi_l(\mathbf{x}_i)$. Then, spectral decomposition (4) allows to write empirical kernel matrix as

$$\mathbf{K} = \boldsymbol{\Phi}^T \boldsymbol{\Lambda} \boldsymbol{\Phi}. \tag{5}$$

**Data models.** A standard approach to analyzing the generalization error consists in considering general families of kernels $K(\mathbf{x}, \mathbf{x}')$ and targets $f^*(\mathbf{x})$, typically defined by regularity assumptions, and deriving upper and lower generalization bounds (e.g., see (Caponnetto & De Vito, 2007)). We adopt a different approach that allows us to go beyond just the bounds and describe generalization error $L_{\widehat{f}}$ with more quantitative detail. To this end, we consider two particular models, *Circle* and *Wishart*, that represent extreme low- and high-dimensional cases of the kernel learning setting.

*Wishart model.* This model is a common choice for our setting: it was explicitly assumed in (Cui et al., 2021; Jacot et al., 2020b; Simon et al., 2023) and is closely related to settings of (Jacot et al., 2020a; Bordelon et al., 2020; Canatar et al., 2021; Wei et al., 2022). Specifically, assume that the kernel features $\phi_l(\mathbf{x})$ and data distribution $p(\mathbf{x})$ are such that feature matrix $\boldsymbol{\Phi}$ has i.i.d. Gaussian entries: $\phi_l(\mathbf{x}_i) \sim \mathcal{N}(0, 1)$. Then, the resulting empirical kernel matrix in (5) is called Wishart matrix in Random Matrix Theory (RMT) context. Intuitively, we can think about Wishart model as a model of some high-dimensional data with all the structural information about data distribution and kernel being wiped out by high-dimensional fluctuations.

*Circle model.* To describe the generalization of a kernel estimator trained by completely fitting (i.e. interpolating) training data, Spigler et al. (2020) and Beaglehole et al. (2023) used a model with translations-invariant kernel and training inputs forming a regular lattice in a hypercube. In this work, we consider, for transparency, a one-dimensional version of this model. Yet, the one-dimensional version will display all the important phenomena we plan to discuss. Specifically, let inputs come from the circle $x \in \mathcal{S} = \mathbb{R} \mod 2\pi$, and the training set $\mathcal{D}_N = \{u + \frac{2\pi i}{N}\}_{i=0}^{N-1}$. Here, $u$ is an overall shift of the training data lattice, which we sample uniformly $u \sim \mathcal{U}([0, 2\pi])$ to introduce some randomness in otherwise deterministic training set $\mathcal{D}_N$. Then, the kernel and the target are defined in the basis of Fourier harmonics as

$$K(x, x') = \sum_{l=-\infty}^{\infty} \lambda_l e^{\mathbf{i}l(x-x')}, \quad f^*(x) = \sum_{l=-\infty}^{\infty} c_l e^{\mathbf{i}lx}. \tag{6}$$

We assume $\lambda_l = \lambda_{-l} \in \mathbb{R}$ and $c_{-l} = \overline{c_l}$ to ensure that both the kernel and the target are real-valued.

## 3 SPECTRAL ALGORITHMS AND THEIR GENERALIZATION ERROR

Several authors, e.g. Bauer et al. (2007); Rastogi & Sampath (2017); Lin et al. (2020), considered *Spectral Algorithms* to generalize and extend the type of regularization performed by classical methods such as KRR and GD. Indeed, both KRR and GD fit observation vector $\mathbf{y}$ to a different extent in different spectral subspaces of the empirical kernel matrix $\mathbf{K}$. For spectral algorithms, this fitting degree is specified by a profile $h(\lambda)$ so that the estimator is given by

$$\widehat{f}(\mathbf{x}) = \mathbf{k}(\mathbf{x})^T \mathbf{K}^{-1} h\big(\tfrac{1}{N}\mathbf{K}\big)\mathbf{y}. \tag{7}$$

Here $\mathbf{k}(\mathbf{x}) \in \mathbb{R}^N$ has components $\big(\mathbf{k}(\mathbf{x})\big)_i = K(\mathbf{x}, \mathbf{x}_i)$. Function $h(\lambda)$ is applied to the kernel matrix $\frac{1}{N}\mathbf{K}$ in the operator sense: $h(\cdot)$ acts element-wise on diagonal matrices, and for an arbitrary positive semi-definite matrix with eigendecomposition $\mathbf{A} = \mathbf{U}^T \mathbf{D} \mathbf{U}$ we have $h(\mathbf{A}) = \mathbf{U}^T h(\mathbf{D})\mathbf{U}$.

Let us show how classical algorithms can be written in the form (7) with a specific choice of $h(\lambda)$:

1. Kernel Ridge Regression with regularization $\eta$ is obtained with $h_\eta(\lambda) = \frac{\lambda}{\lambda + \eta}$. Then, (7) transforms into the classical formula for KRR predictor: $\widehat{f}(\mathbf{x}) = \mathbf{k}(\mathbf{x})^T\big(\mathbf{K} + N\eta\mathbf{I}\big)^{-1}\mathbf{y}$.

2. Gradient Flow by time $t$ is obtained with $h_t(\lambda) = 1 - e^{-t\lambda}$. (For this and the next example, we provide the respective derivations in Section B.1.)

3. For an arbitrary general first-order iterative algorithm at iteration $t$, $h_t(\lambda)$ is given by the associated degree-$t$ polynomial with $h_t(\lambda = 0) = 0$ (see Section B.1). For example, GD with a learning rate $\alpha$ is given by $h_t(\lambda) = 1 - (1 - \alpha\lambda)^t$.

Now, we make a simple observation that is crucial to the current work. Note that generalization error (3) is quadratic in the estimator $\widehat{f}$, while $\widehat{f}$ is linear in $h$ according to (7). Thus, for any problem, the generalization error is quadratic in the profile $h$. This observation is formalized (see the proof in Section A) in

**Proposition 1.** *There exist signed measures $\rho^{(2)}(d\lambda_1, d\lambda_2)$, $\rho^{(1)}(d\lambda)$ and $\rho^{(\varepsilon)}(d\lambda)$ (given in equations (37)-(39)) such that the map $h \mapsto \widehat{f} \mapsto L_{\widehat{f}}$ given by (7), (3) is expressed as the quadratic functional*

$$\begin{aligned} L[h] =& \frac{1}{2}\left[\int\int h(\lambda_1)h(\lambda_2)\rho^{(2)}(d\lambda_1, d\lambda_2) - 2\int h(\lambda)\rho^{(1)}(d\lambda) + \|f^*(\mathbf{x})\|^2\right] \\ &+ \frac{1}{2}\frac{\sigma^2}{N}\int h^2(\lambda)\rho^{(\varepsilon)}(d\lambda). \end{aligned} \tag{8}$$

We will refer to the measures $\rho^{(1)}, \rho^{(2)}$ and $\rho^{(\varepsilon)}$ as the *learning measures*. Proposition 1 shows that the loss functional is completely specified by these measures. The first line in (8) describes the estimation of the target function from the signal part $\mathbf{f}^*$ of the observation vector $\mathbf{y}$, which is hindered by insufficiency of $N$ observations to capture fine details of $f^*(\mathbf{x})$. Similarly, the second line in (8) describes the effect of (unwanted) learning of the noise part $\varepsilon$ of observations $\mathbf{y}$.

The functional (8) makes the relation between the learning algorithm $h(\lambda)$ and the generalization error maximally explicit. However, properties of the underlying kernel and data are reflected in the learning measures $\rho^{(2)}(d\lambda_1, d\lambda_2)$, $\rho^{(1)}(d\lambda)$ and $\rho^{(\varepsilon)}(d\lambda)$ in a fairly complicated way. In Section A, we show some general connections between the kernel eigenvalues $\lambda_l$, the features $\phi_l(\mathbf{x})$, and the learning measures. Yet, the explicit characterization of learning measures is challenging even for our two data models, and constitutes the main technical step of our work.

**Optimal algorithm.** Consider a regression problem and its associated loss functional (8). Since the loss functional is positive semi-definite, under suitable regularity assumptions it has a (possibly non-unique) minimizer

$$h^*(\lambda) = \arg\min L[h] \tag{9}$$

that achieves the minimal possible generalization error in a given problem. We refer to the spectral algorithm with profile $h^*(\lambda)$ as *optimal*. In the context of models with power-law spectra, we will also speak of optimal algorithms in a broader sense, as those providing the optimal error scaling with $N$. We will analyze the conditions of optimality in the Circle and Wishart models and show that in the noisy setting they have the same structure, easily understood using a simplified loss model.

## 4 EXPLICIT FORMS OF THE LOSS FUNCTIONAL

**Circle model.** The main advantage of this model is that it admits an exact and explicit solution. Below, we describe its main properties, with derivations and proofs given in Section D.

Due to the fact that training inputs $\mathbf{x}_i$ form a regular lattice, the eigenvalues $\widehat{\lambda}_k$ of empirical kernel matrix $\mathbf{K}$ become deterministic: $\widehat{\lambda}_k = \sum_{n=-\infty}^{\infty} \lambda_{l+Nn}$. Behind this relation is the learning picture based on aliasing. For a given $k \in \overline{0, N-1}$, the information about the target function contained in all Fourier harmonics with frequencies $l = k + Nn$, $n \in \mathbb{Z}$ is compressed into the single $k$-th harmonic, and then projected back to the original $l = k + Nn$ harmonics of the estimator $\widehat{f}$. This leads to a transformation of population quantities $\lambda_l^a |c_l|^{2b}$ that we call $N$-*deformation*:

$$\left[ \lambda_l^a |c_l|^{2b} \right]_N \equiv \sum_{n=-\infty}^{\infty} \lambda_{l+Nn}^a |c_{l+Nn}|^{2b}. \tag{10}$$

It is periodic: $\left[ \lambda_l^a |c_l|^{2b} \right]_N = \left[ \lambda_{l+N}^a |c_{l+N}|^{2b} \right]_N$. Also, $\widehat{\lambda}_k = \left[ \lambda_k \right]_N$. Then, we have

**Theorem 1.** *Loss functional of the Circle model is given by*

$$L[h] = \frac{1}{2} \sum_{k=0}^{N-1} \left[ \left( \frac{\sigma^2}{N} + \left[ |c_k|^2 \right]_N \right) \frac{\left[ \lambda_k^2 \right]_N}{\left[ \lambda_k \right]_N^2} h^2(\widehat{\lambda}_k) - 2 \frac{\left[ \lambda_k |c_k|^2 \right]_N}{\left[ \lambda_k \right]_N} h(\widehat{\lambda}_k) + \left[ |c_k|^2 \right]_N \right]. \tag{11}$$

The special feature of the loss functional (11) compared to the general form (8) in that there are no off-diagonal $\lambda_1 \neq \lambda_2$ contributions to the loss. Then, the optimal algorithm is found by a simple point-wise minimization:

$$h^*(\widehat{\lambda}_k) = \frac{[\lambda_k |c_k|^2]_N [\lambda_k]_N}{\left( \frac{\sigma^2}{N} + \left[ |c_k|^2 \right]_N \right) [\lambda_k^2]_N}. \tag{12}$$

**Wishart model.** This model, although more common in the literature, does not enjoy an exact solution like the Circle model. However, using two approximations, in Section E we derive an explicit form of the measures $\rho^{(2)}, \rho^{(1)}, \rho^{(\varepsilon)}$ describing the loss by equation (8). We give now an outline of our derivation.

First, we point out that the learning measures from (8) can be reduced to the first and second moment of the imaginary part of the resolvent $\widehat{\mathbf{R}}(z) = (\frac{\mathbf{K}}{N} - z\mathbf{I})^{-1}$ computed at the points $z = \lambda + \mathbf{i}0_+$ near the real line. Then, we make standard RMT assumptions to describe the resolvent in terms of the Stieltjes transform $r(z)$ of spectral measure of $\mathbf{K}$, which satisfies the fixed-point equation

$$1 = -zr(z) + \frac{1}{N} \sum_l \frac{r(z)\lambda_l}{r(z)\lambda_l + 1}. \tag{13}$$

The first resolvent moment is computed straightforwardly and the second moment at the same point $z_1 = z_2$ can be computed with differentiation trick (Simon et al., 2023), but for the second moment at $z_1 \neq z_2$ a new tool is required. For that, we employ Wick's theorem of computing averages over Gaussian fields, where we take into account leading order pairings and neglect subleading $O(N^{-1})$ terms. The above procedure expresses the moments of $\widehat{R}(z)$ in terms of three auxiliary functions

$$v(z) = \sum_l \frac{c_l^2}{\lambda_l + r^{-1}(z)}, \quad u(z) = \sum_l \frac{\lambda_l c_l^2}{\lambda_l + r^{-1}(z)}, \quad w(z) = \sum_l \frac{\lambda_l^2}{\lambda_l + r^{-1}(z)}. \tag{14}$$

Finally, we obtain loss functional (8) by specifying each of the learning measures. Using the notation $\Im\{z\}$ for imaginary part of $z$, we find first moment of signal measure and the noise measure to be

$$\frac{\rho^{(1)}(d\lambda)}{d\lambda} = \frac{\Im u(\lambda)}{\pi \lambda}, \qquad \frac{\rho^{(\varepsilon)}(d\lambda)}{d\lambda} = \frac{\Im w(\lambda)}{\pi \lambda^2}. \tag{15}$$

The second moment of signal measure has diagonal $\lambda_1 = \lambda_2$ and off-diagonal $\lambda_1 \neq \lambda_2$ parts

$$\frac{\rho^{(2)}(d\lambda_1, d\lambda_2)}{d\lambda_1 d\lambda_2} = \frac{|r^{-1}(\lambda_1)|^2}{\pi \lambda_1^2} \Im\{v(\lambda_1)\} \delta(\lambda_1 - \lambda_2)$$

$$+ \frac{1}{\pi^2 \lambda_1 \lambda_2} \frac{\Im\{u(\lambda_2)\} \Im\{r^{-1}(\lambda_1)\} - \Im\{u(\lambda_1)\} \Im\{r^{-1}(\lambda_2)\}}{\lambda_1 - \lambda_2}. \tag{16}$$

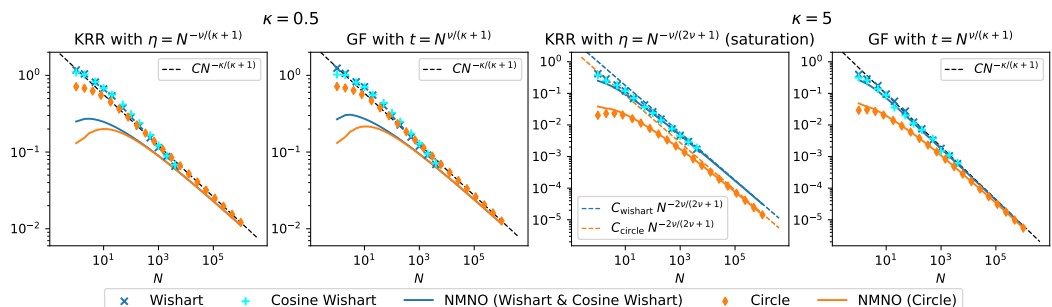

Figure 1: Generalization error of different data models in presence of observation noise converges to our NMNO model (*solid*) as $N \to \infty$, which in turn converges to its $O(N^{-\#})$ asymptotic (*dashed*). All plots have $\nu = 1.5$. *Cosine Wishart* is an additional data model not covered by our theory yet converging to NMNO. The difference between Circle and Wishart asymptotic on the plot 3 is due to localization of the error on scale $s = 0$ at saturation. For details and extended discussion see Sec. F.

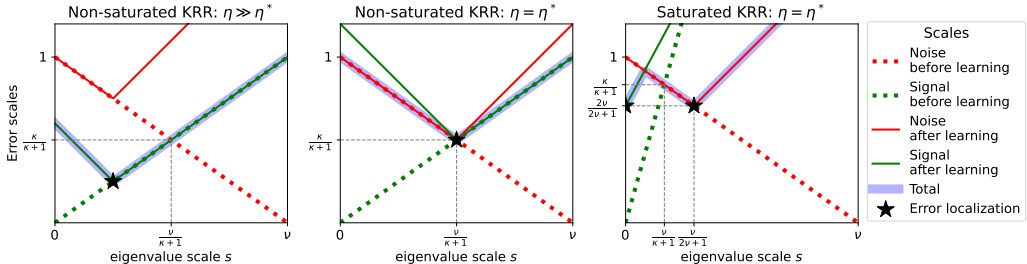

Figure 2: Scale diagrams of different KRR regimes for noisy observations. All plots have $\nu = 1.2$, while $\kappa = 1.0$ in the non-saturated case (left and center) and $\kappa = 5.0$ in the saturated case (right). The dotted lines represent the noise $1 - \frac{s}{\nu}$ and signal $\frac{\kappa}{\nu} s$ terms in equation (18). The solid lines show the same terms with added components $2S^{(h)}$ and $2S^{(1-h)}$. **Left:** the sub-optimal ($s_h > s_*$) non-saturated case. **Center:** the optimal ($s_h = s_*$) non-saturated case. **Right:** the saturated ($\kappa > 2\nu$) case with the choice $s_\eta = \frac{\nu}{2\nu+1}$ optimal for KRR, but sub-optimal among general algorithms $h$.

**Naive Model of Noisy Observations.** As we can see from our results for Circle and Wishart model above, the loss functional of a given model can be quite complicated. We will see, however, that in the presence of observation noise $\sigma^2 > 0$ the asymptotic ($N \to \infty$) generalization properties of both these models are well described by a simple artificial model (NMNO), introduced below. We conjecture this match to be a universal phenomenon, valid for a wide range of models.

For a large dataset sizes $N$, we expect all the complex details of the problem to be concentrated at eigendirections $l$ with small $\lambda_l$, whose features $\phi_l(\mathbf{x})$ can not be well captured by the empirical kernel matrix $\mathbf{K}$ of size $N$. On the contrary, $\mathbf{K}$ should succeed in capturing $\phi_l(\mathbf{x})$ with moderate $\lambda_l$, and empirical and population eigendecompositions should be close to each other.

Let us therefore assume that we can ignore the small eigenvalues and determine the generalization error only using components $l$ with moderate $\lambda_l$ (later, we will explain this by loss localization at moderate spectral scales). Then, the contribution of the $l$-th component to the generalization error can be approximated by $(1 - h(\lambda_l))^2 c_l^2$ for signal fitting part, and $\frac{\sigma^2}{N} h^2(\lambda_l)$ for the learned observation noise. This completely determines the associated loss functional. Let us describe the population spectral data $\lambda_l, c_l$ by the spectral eigenvalue measure $\mu_\lambda(d\lambda) = \sum_l \delta_{\lambda_l}(d\lambda)$ and the coefficient measure $\mu_c(d\lambda) = \sum_l c_l^2 \delta_{\lambda_l}(d\lambda)$. Then, we define the NMNO model by the functional

$$L^{(\text{nmno})}[h] = \frac{\sigma^2}{2N} \int_{\lambda_{\min}}^{1} h^2(\lambda)\mu_\lambda(d\lambda) + \frac{1}{2} \int_{\lambda_{\min}}^{1} \left(1 - h(\lambda)\right)^2 \mu_c(d\lambda). \tag{17}$$

Here, $\lambda_{\min}$ is a reference minimal population eigenvalue defined by the condition $\mu_\lambda([\lambda_{\min}, 1]) = N$ (i.e., such that the segment $[\lambda_{\min}, 1]$ contains exactly $N$ population eigenvalues).

## 5 RESULTS UNDER POWER-LAW SPECTRAL DISTRIBUTIONS

In this section we perform a deeper study of the Circle, Wishart and NMNO models of Section 4 in the setting of power-law distributions (2). Prioritizing transparency over generality, we assume $\lambda_l, c_l$ to be exact power-laws in the form convenient for a given model. Specifically, for Circle model we take $\lambda_l = (|l| + 1)^{-\nu}$ and $|c_l|^2 = (|l| + 1)^{-\kappa-1}$, while for Wishart model we assume continuous analogs of these spectral distributions, namely assume them to have smooth densities supported on $[0, 1]$: $\mu_\lambda(d\lambda) = \frac{1}{\nu}\lambda^{-1-\frac{1}{\nu}} d\lambda$ and $\mu_c(d\lambda) = \frac{1}{\nu}\lambda^{\frac{\kappa}{\nu}-1} d\lambda$.

### 5.1 SCALING ANALYSIS AND ITS APPLICATION TO THE NMNO MODEL

Under power-law spectral assumptions, it is key to observe that various important quantities scale as suitable powers of the training set size $N$. In general, given a sequence $a_N$, we will say that it *has scale* $s$ if for any $\epsilon > 0$ we have $|a_N| = o(N^{-s+\epsilon})$ and $|a_N| = \omega(N^{-s-\epsilon})$. If only the first or the second condition holds, we say that the scale of $a_N$ is not less or not greater than $s$ (respectively).

Suppose that $g_N(\lambda)$ is a sequence of functions on the spectral interval $(0, 1]$. We say that this sequence has a *scaling profile* $S^{(g)}(s), s \geq 0$, if for any sequence $\lambda_N \in (0, 1]$ that has scale $s$ the sequence $|g_N(\lambda_N)|$ has scale $S^{(g)}(s)$. It is easy to check that the scaling profile, if exists, is a continuous function of $s$ (see Lemma 1). The basic example of a sequence of functions having a scaling profile is the sequence $g_N(\lambda) = N^a \lambda^b$ with constant $a, b$; in this case $S^{(g)}(s) = bs - a$.

Integrals of functions with a scaling profile also have a specific scaling:

**Proposition 2** (see proof in Section C). *Let* $g_N(\lambda)$ *be a sequence of functions with a scaling profile* $S^{(g)}(s)$, *and let* $a_N > 0$ *be a sequence of scale* $a > 0$. *Then, the sequence of integrals* $\int_{a_N}^1 |g_N(\lambda)| d\lambda$ *has scale* $s_* = \min_{0 \leq s \leq a}(S^{(g)}(s) + s)$.

When prop. 2 can be applied to the functional (8), we call the set $\mathcal{S}_{\mathrm{loc}} = \arg\min_{0 \leq s \leq a}(S^{(g)}(s) + s)$ of scales which dominate the loss integral as *spectral localization scales* of the generalization error. In the rest of the paper, we reserve the letter $s$ to denote the scale of eigenvalues $\lambda$.

**Application to NMNO.** Now we apply the scaling arguments to the NMNO model (17), for which it will be easy to find explicit optimality conditions. Suppose that the problem has either discrete or continuous power-law spectrum with exponents $\nu, \kappa$ as described above for the Circle and Wishart model. Then, $\lambda_{\min}$ in equation (17) has finite scale $\nu$. Suppose that the functions $h$ and $1 - h$ have scaling profiles $S^{(h)}$ and $S^{(1-h)}$. Then, applying Proposition 2 in the continuous case or analogous Proposition 5 in the discrete case, we obtain

**Proposition 3.** *The NMNO loss* $L^{(\mathrm{nmno})}[h]$ *has scaling*

$$S^{\mathrm{nmno}}[h] = \min_{0 \leq s \leq \nu} \left[ \left(1 - \tfrac{s}{\nu} + 2S^{(h)}(s)\right) \wedge \left(\tfrac{\kappa}{\nu}s + 2S^{(1-h)}(s)\right) \right]. \tag{18}$$

Here, $\wedge = \min$; the first and second arguments in $\wedge$ come from the noise and signal terms in (17), respectively. In the continuous case we use the fact that $N^{-1}\lambda^{-1-1/\nu}h(\lambda)^2$ has scaling profile $1 - s(1 + \frac{1}{\nu}) + 2S^{(h)}$ while $\lambda^{\kappa/\nu-1}(1 - h(\lambda))^2$ has scaling profile $s(\frac{\kappa}{\nu} - 1) + 2S^{(1-h)}$.

Clearly, for any particular $s$ only one of the values $S^{(h)}(s)$ and $S^{(1-h)}(s)$ can be strictly positive. This implies a bound on feasible loss scalings:

$$S^{\mathrm{nmno}}[h] \leq \min_{0 \leq s \leq \nu} \left( \left(1 - \tfrac{s}{\nu}\right) \vee \tfrac{\kappa}{\nu}s \right) = \tfrac{\kappa}{\kappa+1}. \tag{19}$$

The minimum here is attained at the scale $s_* = \frac{\nu}{\kappa+1}$. Moreover, an algorithm $h$ attains the optimal scale $\frac{\kappa}{\kappa+1}$ exactly when for each $s \in [0, \nu]$ we have $\left(1 - \frac{s}{\nu} + 2S^{(h)}(s)\right) \wedge \left(\frac{\kappa}{\nu}s + 2S^{(1-h)}(s)\right) \geq \frac{\kappa}{\kappa+1}$:

**Proposition 4.** $S^{\mathrm{nmno}}[h] \leq \frac{\kappa}{\kappa+1}$, *and the equality occurs when 1)* $S^{(h)}(s) \geq \frac{1}{2}\left(\frac{s}{\nu} - \frac{1}{\kappa+1}\right)$ *for* $s \geq s_* = \frac{\nu}{\kappa+1}$ *and 2)* $S^{(1-h)}(s) \geq \frac{1}{2}\left(\frac{\kappa}{\kappa+1} - \frac{\kappa}{\nu}s\right)$ *for* $s \leq s_*$.

These results provide a simple picture of spectral algorithms close to optimality for the NMNO model: one should choose the spectral function $h$ so that $|h(\lambda)| \lesssim N^{\frac{1}{2(\kappa+1)}} \lambda^{\frac{1}{2\nu}}$ for $\lambda \lesssim N^{-\frac{\nu}{\kappa+1}}$

and so that $|1 - h(\lambda)| \leq N^{-\frac{\kappa}{2(\kappa+1)}} \lambda^{-\frac{\kappa}{2\nu}}$ for $\lambda \gtrsim N^{-\frac{\nu}{\kappa+1}}$. The value $s_* = \frac{\nu}{\kappa+1}$ can be referred to as the *loss localization scale* for the optimal algorithm.

Let us apply the obtained conditions to KRR and GF. KRR with regularization $\eta$ has $h_\eta(\lambda) = \frac{\lambda}{\lambda+\eta}$. Suppose that $\eta$ has scale $s_\eta$, then $h_\eta$ has the scaling profile $S^{(h)}(s) = (s - s_\eta) \vee 0$, while $1 - h_\eta$ equals $\frac{\eta}{\lambda+\eta}$ and has the scaling profile $S^{(1-h)}(s) = (s_\eta - s) \vee 0$. Recalling that $\nu > 1$, we see that condition 1) in Proposition 4 is satisfied iff $s_\eta \leq s_* = \frac{\nu}{\kappa+1}$. Condition 2) is more subtle: if $\frac{\kappa}{2\nu} \leq 1$, then it is satisfied iff $s_\eta \geq s_*$, but in the case $\frac{\kappa}{2\nu} > 1$ it is rather satisfied iff $s_\eta \geq \frac{\kappa}{2(\kappa+1)}$. We see, in particular, that in the case $\frac{\kappa}{2\nu} \leq 1$ conditions 1) and 2) are simultaneously satisfied iff $s_\eta = s_*$, while in the case $\frac{\kappa}{2\nu} > 1$ they cannot be simultaneously satisfied (and so KRR cannot achieve the optimal scaling $\frac{\kappa}{\kappa+1}$) – an effect called *saturation* (Mathé, 2004; Bauer et al., 2007). See Figure 2 for an illustration. In contrast, GF $h_t(\lambda) = 1 - e^{-t\lambda}$ has no saturation: choosing $t$ to be of scale $-s_*$ satisfies both conditions of Proposition 4 for all $\nu > 1$ and $\kappa > 0$.

## 5.2 NOISY OBSERVATIONS AND MODEL EQUIVALENCE

For our two data models, Circle and Wishart, the intuition behind the NMNO ansatz can be rigorously justified by showing that this ansatz represents the leading contribution to the true loss. For instance, consider the circle model with loss functional[(11) specified by (10). The empirical eigenvalues $\widehat{\lambda}_k = [\lambda_k]_N = \lambda_k + O(N^{-\nu})$ for $|k| < N/2$, so the scale $\nu$ of the correction $|\widehat{\lambda}_k - \lambda_k|$ is higher than the scale $s$ of the eigenvalues $\lambda_k$ except for the eigenvalues of the highest scale $s = \nu$. This shows that the empirical and population quantities are significantly different only on the highest spectral scale $s = \nu$. Continuing this line of arguments, we get

**Theorem 2.** *Assume that the learning algorithm $h(\lambda)$ is such that $h(\lambda)$ and $1 - h(\lambda)$ have scaling profiles $S^{(h)}(s)$ and $S^{(1-h)}(s)$. Assume that the maps $\log \lambda \mapsto \log |h(\lambda)|$ and $\log \lambda \mapsto \log |1-h(\lambda)|$ are globally Lipschitz, uniformly in $N$. Then, if $\nu = s$ is not a localization point of NMNO functional $L^{(\mathrm{nmno})}[h]$, Circle model specified by (11) and Wishart model specified by (16),(15) are equivalent to the NMNO model in the limit $N \to \infty$:*

$$L^{(\mathrm{nmno})}[h] = L^{(\mathrm{circle})}[h]\big(1 + o(1)\big) = L^{(\mathrm{wishart})}[h]\big(1 + o(1)\big). \tag{20}$$

We prove the theorem separately for Circle and Wishart models in Sections D.2 and E.3.3. Note that the condition of equivalence is specified using only NMNO model. Thus, if satisfied, it allows analyzing the simple functional (17) instead of the more complicated ones (16), (15) and (11). The requirement that $s = \nu$ is not a localization point is reasonable as the heuristic derivation of the NMNO model in Section 4 included an assumption that the loss is localized at moderate scales.

## 5.3 NOISELESS OBSERVATIONS

We focus on Circle model to describe main observations, with derivation deferred to Section D and Wishart model results deferred to Section E. Let us write the loss functional perturbatively

$$L[h] = \sum_{k=-\frac{N}{2}}^{\frac{N}{2}} \left[ \frac{|c_k|^2}{2} \big(1 + o(\tau)\big) \Big( h(\widehat{\lambda}_k) - h^*(\widehat{\lambda}_k) \Big)^2 + N^{-\kappa-1} \big(O(1) + O(\tau^{2\nu-\kappa-1})\big) \right] \tag{21}$$

with $\tau \equiv \frac{|k|+1}{N}$ as a small parameter. From this, we make several observations. First, take $h = h^*$. Then, if $\kappa < 2\nu$, the loss localizes on $s = \nu$ (i.e. the sum is accumulated at $|k| \sim N$) and has the rate $O(N^{-\kappa})$. This rate is natural and reflects that we are able to learn target function everywhere except at inaccessible scales $\lambda \ll N^{-\nu}$. Moreover, by examining the first term in (21), we see that this rate is not destroyed by learning algorithms sufficiently close to the optimal: $|h(\widehat{\lambda}_k) - h^*(\widehat{\lambda}_k)|^2 = o(\tau^\kappa)$.

The situation changes dramatically for $\kappa > 2\nu$: the optimal loss $L[h^*]$ becomes dominated by $O(\tau^{2\nu-\kappa-1})$ term in (21) and localizes at $s = 0$ (i.e. the sum is accumulated at $|k| \sim 1$), changing the rate to $O(N^{-2\nu})$. We call this behavior *saturation* since it has features similar to KRR saturation effect for noisy observations: transition at $\kappa = 2\nu$; change of error rate; localization at $s = 0$. However, noisy KRR saturation is algorithm-driven and can be removed by replacing KRR with GF, while saturation in (21) persists even for optimal algorithm $h^*(\lambda)$. Interestingly, the optimal

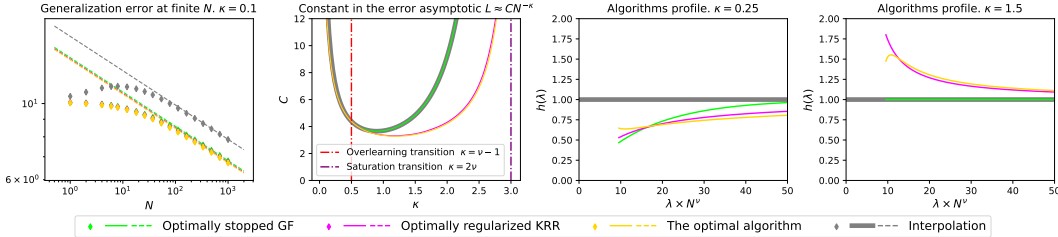

Figure 3: Generalization error (**left**) and profiles $h(\lambda)$ (**right**) of various algorithms applied to the Circle model with $\nu = 1.5$ and noiseless observations with different $\kappa$. Before overlearning transition $\kappa = \nu - 1$ optimal algorithms underlearn observations ($h(\lambda) < 1$) while starting to overlearn them ($h(\lambda) > 1$) after the transition. For details and extended discussion see Section F.

loss in the saturated phase can be achieved (asymptotically) by KRR with a negative regularization: $L[h^*] = L[h_{\eta^*}](1 + o(1))$, $\eta^* < 0$. Denoting Riemann zeta function as $\zeta(\alpha)$, we have

$$L[h_\eta] = \frac{1}{2}\Big((\eta N^\nu + 2\zeta(\nu))^2 + 2\zeta(2\nu)\Big)\left[\sum_{l=-\infty}^{\infty}\frac{|c_l|^2}{\lambda_l^2}\right]N^{-2\nu}\big(1 + o(1)\big), \tag{22}$$

with minimum at $\eta^* = -2\zeta(\nu)N^{-\nu}$. The benefit of negative regularization was empirically observed in (Kobak et al., 2020) and theoretically approached in (Tsigler & Bartlett, 2023). Within our framework, KRR with negative regularization can be thought of as a special case of *overlearning* the observations.

In fact, overlearning also occurs in the $\kappa < 2\nu$ phase. In Section D.3 we show that the loss functional can be asymptotically written (see (138)) using $\tau = \frac{|k|+1}{N}$ for (continuous) eigenspace indexing, for example $\widehat{\lambda}_\tau \leftarrow \widehat{\lambda}_k$. Then, denoting Hurwitz zeta function as $\zeta(\alpha, x)$, the optimal algorithm becomes

$$h^*(\widehat{\lambda}_\tau) = \frac{\zeta_\tau^{(\nu+\kappa+1)}\zeta_\tau^{(\nu)}}{\zeta_\tau^{(\kappa+1)}\zeta_\tau^{(2\nu)}}, \qquad \zeta_x^{(\alpha)} \equiv \zeta(\alpha, x) + \zeta(\alpha, 1-x). \tag{23}$$

The optimal algorithm (23) has an intriguing property (see also Figure 3): it interpolates the data $h^*(\lambda) = 1$ when $\kappa = \nu - 1$, and, otherwise, the sign of $1 - h^*(\lambda)$ coincides with the sign of $\nu - 1 - \kappa$. In other words, for hard targets with $\kappa < \nu - 1$, classical regularization by underfitting the observation is required for optimal performance. But, for easier targets with $\kappa > \nu - 1$ it becomes optimal to overlearn the training data in contrast to conventional wisdom. The same holds for Wishart model (see Sec. E.3.4). Thus, we identify the point $\kappa = \nu - 1$ as *overlearning transition*.

## 6 DISCUSSION

We have extended results of type (1) to general spectral algorithms, as given by our loss functional (11) for Circle model and (15),(16) for Wishart model. It allows to address questions that require going beyond specific (KRR,GF) algorithms. For example, we show that the nature of saturation at $\kappa = 2\nu$ is different for noisy and noiseless observations, with the latter being an intrinsic property of the given kernel and data model, and can not be removed by any choice of the learning algorithm.

Our formalism of spectral localization and scaling, while being compact, provides a simple and transparent picture of the variety of convergence rates under power-law spectra distributions. Also, the equivalence result between our two data models and naive model of noisy observations (17) relies on the straightforward estimation of the scale of perturbation of population quantities by finite size $N$ of the training dataset. Thus, an interesting direction for future research would be to check whether the equivalence holds for other data and kernel models.

Finally, let us mention the advantage of full loss asymptotic $L = CN^{-\#}(1 + o(1))$ compared to the rates $L = O(N^{-\#})$. In this work, we used the full asymptotic to obtain the shape of optimal algorithm $h^*(\lambda)$. In the noiseless case, the knowledge of $h^*(\lambda)$ allowed us to characterize the overlearning transition at $\kappa = \nu - 1$, which otherwise would be invisible on the level of the rate $O(N^{-\kappa})$. Investigating whether $\kappa = \nu - 1$ remains the point of overlearning phase transition for more general data models is an interesting direction for future research.

ACKNOWLEDGMENTS

We thank Eric Moulines for insightful discussions during the initial stage of the work.

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

## CONTENTS

## A   Spectral perspectives

In this section, we show general relations between generalization error (3), population spectra $\lambda_l, c_l$, and learning algorithms $h(\lambda)$. Recall that profile $h(\lambda)$ is applied to the eigenvalues of the kernel matrix $\mathbf{K}$ (i.e. values of the kernel function evaluated on the training dataset $\mathcal{D}_N$). Therefore, relating generalization error and $h(\lambda)$ involves the properties of the empirical spectrum: eigenvalues of $\mathbf{K}$ and the related eigendecomposition of the observation vector $\mathbf{y}$.

With these remarks in mind, we may say that we are dealing with *population* and *empirical* perspectives on generalization error (3). We start with population perspective in Section A.1, which is behind the classical result (1). Then, we proceed with the empirical perspective in Section A.2, which basically amounts to proving Proposition 1 and introducing learning measures $\rho^{(2)}, \rho^{(1)}, \rho^{(\varepsilon)}$. Finally, In Section A.3, we combine population and empirical perspectives. While probably less conceptual than the first two perspectives, the joint population-empirical perspective is an essential step in our derivation of the loss functional for the Wishart model.

### A.1   Population perspective: transfer matrix

A central object for the population perspective is the *transfer matrix* $\widehat{T}_{ll'}$ introduced explicitly, for example, in (Simon et al., 2023) in the context of KRR. Specifically, let us decompose the prediction (7) over kernel eigenfunctions $\phi_l(\mathbf{x})$ as $\widehat{f}(\mathbf{x}) = \sum_l \widehat{c}_l \phi_l(\mathbf{x})$. Then, the prediction coefficients $\widehat{c}_l$ can be written as

$$\widehat{c}_l = \sum_{l'} \widehat{T}_{ll'} c_{l'} + \sigma \widehat{\varepsilon}_l, \quad \widehat{T}_{ll'} = \lambda_l \phi_l^T \mathbf{K}^{-1} h\big(\tfrac{1}{N}\mathbf{K}\big)\phi_{l'}, \quad \widehat{\varepsilon}_l = \lambda_l \phi_l^T \mathbf{K}^{-1} h\big(\tfrac{1}{N}\mathbf{K}\big)\varepsilon, \quad (24)$$

where $\phi_l$ is the vector of eigenfunctions computed at the dataset inputs $(\phi_l)_i = \phi_l(\mathbf{x}_i)$, and $(\varepsilon)_i = \varepsilon_i$ is the vector of observation noise. Note that the transfer matrix $\widehat{T}_{ll'}$ has a clear interpretation of the rate at which the information $c_{l'}$ contained in spectral component $l'$ is transferred to spectral component $l$. The population noise component $\widehat{\varepsilon}_l$ describes how much of the of the observation noise $\varepsilon$ was learned in the $l$-th population spectral component.

The population loss (3) (sometimes we use this term as a synonym to generalization error) is straightforwardly expressed through the first and second moments of the transfer matrix, and the variance of population noise components

$$L_{\widehat{f}} = \frac{1}{2}\mathbb{E}_{\mathcal{D}_N,\varepsilon}\sum_l \big(\widehat{c}_l - c_l\big)^2 = \frac{1}{2}\sum_{l_1,l_2} c_{l_1}\Big(\sum_{l'} T^{(2)}_{l_1 l' l' l_2} - 2T^{(1)}_{l_1 l_2} + \delta_{l_1 l_2}\Big)c_{l_2} + \frac{\sigma^2}{2N}\sum_l \varepsilon_l^{(2)}, \quad (25)$$

where

$$T_{ll'}^{(1)} = \mathbb{E}_{\mathcal{D}_N}\left[\widehat{T}_{ll'}\right], \tag{26}$$

$$T_{l_1 l_1' l_2' l_2}^{(2)} = \mathbb{E}_{\mathcal{D}_N}\left[\widehat{T}_{l_1' l_1}\widehat{T}_{l_2' l_2}\right], \tag{27}$$

$$\varepsilon_l^{(2)} = N\mathbb{E}_{\mathcal{D}_N,\varepsilon}[(\widehat{\varepsilon}_l)^2] = \mathbb{E}_{\mathcal{D}_N}\left[\lambda_l^2 \frac{1}{N}\boldsymbol{\phi}_l^T\left(\left(\tfrac{1}{N}\mathbf{K}\right)^{-1}h\left(\tfrac{1}{N}\mathbf{K}\right)\right)^2\boldsymbol{\phi}_l\right]. \tag{28}$$

The representation (25) makes the most explicit dependence of the loss on population coefficients $c_l$, while the dependence on the learning algorithm $h(\lambda)$ and population eigenvalues $\lambda_l$ is hidden inside moments $T_{ll'}^{(1)}$, $T_{l_1 l_1' l_2' l_2}^{(2)}$ of the transfer matrix and noise variance $\varepsilon_l^{(2)}$. Yet, for the case of KRR the results (1) shows that the dependence on $\lambda_l$ can be made fairly explicit.

## A.2 Empirical perspective: learning measure

As this perspective focuses on empirical spectrum of kernel matrix and observation vector, we start with writing eigendecomposition of $\mathbf{K}, \mathbf{f}^* = \mathbf{y} - \sigma\boldsymbol{\varepsilon}$ and $\boldsymbol{\varepsilon}$ as

$$\frac{1}{N}\mathbf{K} = \sum_{k=1}^{N}\widehat{\lambda}_k\mathbf{u}_k\mathbf{u}_k^T, \quad \mathbf{u}_k^T\mathbf{u}_{k'} = \delta_{kk'}, \tag{29}$$

$$\frac{1}{\sqrt{N}}\mathbf{f}^* = \sum_{k=1}^{N}\widehat{c}_k\mathbf{u}_k, \tag{30}$$

$$\boldsymbol{\varepsilon} = \sum_{k=1}^{N}\widehat{\varepsilon}_k\mathbf{u}_k, \tag{31}$$

where in the last line $\widehat{\varepsilon}_k$ are i.i.d. normal Gaussian because orthogonal transformation to empirical eigenbasis $\{\mathbf{u}_k\}_{k=1}^{N}$ leaves the distribution of isotropic Gaussian vectors $\boldsymbol{\varepsilon} \sim \mathcal{N}(0, \mathbf{I})$ unchanged.

Then, inserting spectral decomposition (29) of the empirical kernel matrix into the prediction (7) gives

$$\widehat{f}_h(\mathbf{x}) = \left(\mathbf{k}(\mathbf{x})\right)^T\left[\frac{1}{N}\sum_{k=1}^{N}\mathbf{u}_k\mathbf{u}_k^T\frac{h(\widehat{\lambda}_k)}{\widehat{\lambda}_k}\right]\mathbf{f}^* + \frac{\sigma}{N}\sum_{k=1}^{N}\widehat{\varepsilon}_k\frac{h(\widehat{\lambda}_k)}{\widehat{\lambda}_k}\left(\mathbf{k}(\mathbf{x})\right)^T\mathbf{u}_k$$
$$= \int h(\lambda)\widehat{\rho}^{(f)}(\mathbf{x}, d\lambda) + \frac{\sigma}{\sqrt{N}}\int h(\lambda)\widehat{\rho}^{(\varepsilon)}(\mathbf{x}, d\lambda), \tag{32}$$

where $\widehat{\rho}_N^{(f)}(\mathbf{x}, d\lambda)$ and $\widehat{\rho}_N^{(\varepsilon)}(\mathbf{x}, d\lambda)$ are target and noise *learning measures*:

$$\widehat{\rho}^{(f)}(\mathbf{x}, d\lambda) = \left(\mathbf{k}(\mathbf{x})\right)^T\left[\frac{1}{N}\sum_{k=1}^{N}\frac{\mathbf{u}_k\mathbf{u}_k^T}{\lambda}\delta_{\widehat{\lambda}_k}\right]\mathbf{f}^*, \tag{33}$$

$$\widehat{\rho}^{(\varepsilon)}(\mathbf{x}, d\lambda) = \frac{1}{\sqrt{N}}\sum_{k=1}^{N}\widehat{\varepsilon}_k\frac{\left(\mathbf{k}(\mathbf{x})\right)^T\mathbf{u}_k}{\lambda}\delta_{\widehat{\lambda}_k}. \tag{34}$$

The target learning measure defines what pattern is learned from the target function at the neighborhood $d\lambda$ of the empirical spectral position $\lambda$. Similarly, the noise learning measure defines the patterns of the noise learned in the neighborhood of $\lambda$.

As for the population perspective, we substitute the expression of prediction in terms of learning measures (32) into the population loss (3)

$$
\begin{aligned}
L_{\widehat{f}} = &\frac{1}{2}\mathbb{E}_{\mathcal{D}_N,\varepsilon}\left[\left\|\left\|\int h(\lambda)\widehat{\rho}^{(f)}(\mathbf{x},d\lambda) + \frac{\sigma}{\sqrt{N}}\int h(\lambda)\widehat{\rho}^{(\varepsilon)}(\mathbf{x},d\lambda) - f^*(\mathbf{x})\right\|\right\|^2\right] \\
= &\frac{1}{2}\left[\int\int h(\lambda_1)h(\lambda_2)\mathbb{E}_{\mathcal{D}_N}\big\langle\widehat{\rho}^{(f)}(\mathbf{x},d\lambda_1),\widehat{\rho}^{(f)}(\mathbf{x},d\lambda_2)\big\rangle\right. \\
&- 2\int h(\lambda)\mathbb{E}_{\mathcal{D}_N}\big\langle f^*(\mathbf{x}),\widehat{\rho}^{(f)}(\mathbf{x},d\lambda)\big\rangle + \langle f^*(\mathbf{x}),f^*(\mathbf{x})\rangle \\
&+ 2\frac{\sigma}{\sqrt{N}}\mathbb{E}_{\mathcal{D}_N}\left\langle f^*(\mathbf{x}) - \int h(\lambda)\rho^{(f)}(\mathbf{x},d\lambda),\int h(\lambda)\mathbb{E}_\varepsilon\rho^{(\varepsilon)}(\mathbf{x},d\lambda)\right\rangle \\
&+ \left.\frac{\sigma^2}{N}\int\int h(\lambda_1)h(\lambda_2)\mathbb{E}_{\mathcal{D}_N,\varepsilon}\big\langle\widehat{\rho}^{(\varepsilon)}(\mathbf{x},d\lambda_1),\widehat{\rho}^{(\varepsilon)}(\mathbf{x},d\lambda_2)\big\rangle\right].
\end{aligned}
\tag{35}
$$

Now, observe that the term in the second-to-last line in (35) is linear in noise learning measure averaged over $\varepsilon$ which is zero: $\mathbb{E}_\varepsilon\rho^{(\varepsilon)}(\mathbf{x},d\lambda) = \frac{1}{\sqrt{N}}\sum_{k=1}^N\frac{\left(\mathbf{k}(\mathbf{x})\right)^T\mathbf{u}_k}{\lambda}\delta_{\widehat{\lambda}_k}\mathbb{E}_\varepsilon\widehat{\varepsilon}_k = 0$ since $\mathbb{E}_\varepsilon\widehat{\varepsilon}_k = 0$. Similarly, taking the expectation over observation noise $\varepsilon$ helps to simplify the last term

$$
\begin{aligned}
\mathbb{E}_\varepsilon\big\langle\widehat{\rho}^{(\varepsilon)}(\mathbf{x},d\lambda_1),\widehat{\rho}^{(\varepsilon)}(\mathbf{x},d\lambda_2)\big\rangle &= \sum_{k_1,k_2}\delta_{\widehat{\lambda}_{k_1}}(d\lambda_1)\delta_{\widehat{\lambda}_{k_2}}(d\lambda_2)\frac{\langle\mathbf{u}_{k_1}^T\mathbf{k}(\mathbf{x}),\mathbf{u}_{k_2}^T\mathbf{k}(\mathbf{x})\rangle}{N\lambda_1\lambda_2}\mathbb{E}_\varepsilon\widehat{\varepsilon}_{k_1}\widehat{\varepsilon}_{k_2} \\
&= \delta_{\lambda_1}(d\lambda_2)\frac{1}{N}\sum_k\delta_{\widehat{\lambda}_k}(d\lambda_2)\frac{\|\mathbf{u}_k^T\mathbf{k}(\mathbf{x})\|^2}{\lambda^2},
\end{aligned}
\tag{36}
$$

where we have used $\mathbb{E}\widehat{\varepsilon}_{k_1}\widehat{\varepsilon}_{k_2} = \delta_{k_1 k_2}$. Now, one can recognize the loss functional stated in Proposition 1: the first 3 terms and the last term of (35) correspond to the respective terms of (8). In other words, the learning measures announced in Proposition 1 are given by

$$
\rho^{(1)}(d\lambda) = \mathbb{E}_{\mathcal{D}_N}\left[\big\langle f^*(\mathbf{x}),\widehat{\rho}^{(f)}(\mathbf{x},d\lambda)\big\rangle\right],
\tag{37}
$$

$$
\rho^{(2)}(d\lambda_1,d\lambda_2) = \mathbb{E}_{\mathcal{D}_N}\left[\big\langle\widehat{\rho}^{(f)}(\mathbf{x},d\lambda_1),\widehat{\rho}^{(f)}(\mathbf{x},d\lambda_2)\big\rangle\right],
\tag{38}
$$

$$
\rho^{(\varepsilon)}(d\lambda) = \mathbb{E}_{\mathcal{D}_N}\left[\frac{1}{N}\sum_{k=1}^N\frac{\|\mathbf{u}_k^T\mathbf{k}(\mathbf{x})\|^2}{\lambda^2}\delta_{\widehat{\lambda}_k}\right],
\tag{39}
$$

Again, the loss functional (8) represents the empirical perspective on the generalization error, making the dependence on the learning algorithm $h(\lambda)$ very explicit. But, the dependence on the problem's kernel structure and target function is hidden inside measures $\rho^{(1)}(d\lambda)$ and $\rho^{(2)}(d\lambda_1,d\lambda_2)$.

### A.3 JOINT POPULATION-EMPIRICAL PERSPECTIVE: TRANSFER MEASURE

To combine to perspective described above, consider a $l$-th spectral component of learning measure $\widehat{\rho}_l^{(f)}(d\lambda) \equiv \langle\phi_l(\mathbf{x}),\widehat{\rho}^{(f)}(\mathbf{x},d\lambda)\rangle$. Then, inserting decomposition (4) of the target function into target learning measure (33) allows to write

$$
\widehat{\rho}_l^{(f)}(d\lambda) \equiv \langle\phi_l(\mathbf{x}),\widehat{\rho}^{(f)}(\mathbf{x},d\lambda)\rangle = \sum_{l'}c_{l'}\widehat{\rho}_{ll'}^{(f)}(d\lambda),
\tag{40}
$$

where

$$
\widehat{\rho}_{ll'}^{(f)}(d\lambda) = \frac{\lambda_l}{\lambda}\frac{1}{N}\sum_{k=1}^N\left(\phi_l^T\mathbf{u}_k\right)\left(\mathbf{u}_k^T\phi_{l'}\right)\delta_{\widehat{\lambda}_k}
\tag{41}
$$

can be naturally called a *transfer measure*. Now, we insert decomposition (40) into (37) and (38), as well as population eigendecomposition (4) into (39). The scalar products in (37) and (38) become

$$
\big\langle f^*(\mathbf{x}),\widehat{\rho}^{(f)}(\mathbf{x},d\lambda)\big\rangle = \sum_l c_l\widehat{\rho}_l^{(f)}(d\lambda) = \sum_{l,l'}c_l c_{l'}\widehat{\rho}_{ll'}^{(f)}(d\lambda),
$$

$$
\big\langle\widehat{\rho}^{(f)}(\mathbf{x},d\lambda_1),\widehat{\rho}^{(f)}(\mathbf{x},d\lambda_2)\big\rangle = \sum_l\widehat{\rho}_l^{(f)}(d\lambda_1)\widehat{\rho}_l^{(f)}(d\lambda_2) = \sum_{l,l_1,l_2}c_{l_1}c_{l_2}\widehat{\rho}_{ll_1}^{(f)}(d\lambda_1)\widehat{\rho}_{ll_2}^{(f)}(d\lambda_2).
$$

$$\tag{42}$$

As for the norm $\left\|\mathbf{u}_k^T \mathbf{k}(\mathbf{x})\right\|^2$ in (39), we can use

$$\left\langle \mathbf{k}(\mathbf{x}), \left(\mathbf{k}(\mathbf{x})\right)^T \right\rangle = \sum_l \lambda_l^2 \boldsymbol{\phi}_l \boldsymbol{\phi}_l^T. \tag{43}$$

Combining the expressions above and noting that $\|f^*(\mathbf{x})\|^2 = \sum_l c_l^2$ gives yet another representation of the population loss in terms of the first and second moment of the transfer measure and population decomposition of noise variance measure.

$$L[h] = \frac{1}{2} \sum_{l_1, l_2} c_{l_1} \Big( \int h(\lambda_1) h(\lambda_2) \sum_{l'} \rho^{(2)}_{l_1 l' l' l_2}(d\lambda_1, d\lambda_2) - 2 \int h(\lambda) \rho^{(1)}_{l_1 l_2}(d\lambda) + \delta_{l_1 l_2} \Big) c_{l_2} \tag{44}$$

$$+ \frac{\sigma^2}{2N} \sum_l \int h(\lambda)^2 \rho^{(\varepsilon)}_l(d\lambda), \tag{45}$$

where

$$\rho^{(1)}_{ll'}(d\lambda) = \mathbb{E}_{\mathcal{D}_N}\left[ \widehat{\rho}^{(f)}_{ll'}(d\lambda) \right], \tag{46}$$

$$\rho^{(2)}_{l_1 l_1' l_2' l_2}(d\lambda_1, d\lambda_2) = \mathbb{E}_{\mathcal{D}_N}\left[ \widehat{\rho}^{(f)}_{l_1' l_1}(d\lambda_1) \widehat{\rho}^{(f)}_{l_2' l_2}(d\lambda_2) \right], \tag{47}$$

$$\rho^{(\varepsilon)}_l(d\lambda) = \mathbb{E}_{\mathcal{D}_N}\left[ \frac{\lambda_l^2}{\lambda^2} \frac{1}{N} \sum_{k=1}^N \left(\boldsymbol{\phi}_l^T \mathbf{u}_k\right)^2 \delta_{\widehat{\lambda}_k} \right]. \tag{48}$$

If the moments of transfer measure $\rho^{(1)}_{ll'}(d\lambda)$, $\rho^{(2)}_{l_1 l_1' l_2' l_2}(d\lambda_1, d\lambda_2)$ and noise variance measure $\rho^{(\varepsilon)}_l(d\lambda)$ are known, the representation (44) connects spectral distribution $\lambda_l, c_l$ of the problem and learning algorithm $h(\lambda)$ with the population loss, thus justifying the name *joint population-empirical spectral perspective*.

## B  GRADIENT-BASED ALGORITHMS

The purpose of this section is two-fold. First, in Section B.1, we support our examples of $h(\lambda)$ provided in Section 3 with the respective derivations. This amounts to show that for linear models trained with a gradient-based algorithm, the predictor during optimization can be written in the form (7) with a specific choice of $h(\lambda)$. Second, in Section B.2 try to connect general spectral algorithms specified by some profile $h(\lambda)$ with gradient-based optimization, which was not included in the main paper due to the space constraints. For that, we provide a simple construction based on a pair of GF processes.

### B.1  KERNEL FORM OF PREDICTORS

To consider gradient-based optimization for the kernel method setting discussed in the main paper, we need to introduce a linear parametric model $\widehat{f}(\mathbf{w}, \mathbf{x})$ whose parameters $\mathbf{w}$ will be updated during the optimization process. Starting with a kernel $K(\mathbf{x}, \mathbf{x}')$ with population decomposition (4), let us define the model features $\psi_l(\mathbf{x}) = \sqrt{\lambda_l} \phi_l(\mathbf{x})$. Then, combining the features in a vector $\boldsymbol{\psi}(\mathbf{x}) \in \mathbb{R}^P$, $\left(\boldsymbol{\psi}(\mathbf{x})\right)_l = \psi_l(\mathbf{x})$, the linear model is defined as

$$\widehat{f}(\mathbf{w}, \mathbf{x}) = \langle \mathbf{w}, \boldsymbol{\psi}(\mathbf{x}) \rangle, \quad \mathbf{w} \in \mathbb{R}^P. \tag{49}$$

For positive definite kernels $P = \infty$, and both model's features and parameters belong to RKHS $\mathcal{H}_K$ of the kernel $K$: $\mathbf{w}, \boldsymbol{\psi}(\mathbf{x}) \in \mathcal{H}_K$.

The (neural) tangent kernel (Jacot et al., 2018) of the model (49) is given by $\mathrm{NTK}_{\widehat{f}}(\mathbf{x}, \mathbf{x}') = \langle \boldsymbol{\psi}(\mathbf{x}), \boldsymbol{\psi}(\mathbf{x}') \rangle = \sum_l \lambda_l \phi_l(\mathbf{x}) \phi_l(\mathbf{x}') = K(\mathbf{x}, \mathbf{x}')$, thus reproducing our original kernel we have started with. Note that one can go in the opposite direction: start from the linear (49) and then define a kernel method specified by the tangent kernel (NTK) of the linear model. An especially interesting example of the latter direction is a (non-linear) neural network $f(\boldsymbol{\theta}, \mathbf{x})$ linearized at $\boldsymbol{\theta}_0$ resulting in $f_{\mathrm{lin}}(\boldsymbol{\theta}, \mathbf{x}) = f(\boldsymbol{\theta}_0, \mathbf{x}) + \langle \boldsymbol{\theta} - \boldsymbol{\theta}_0, \nabla_{\boldsymbol{\theta}} f(\boldsymbol{\theta}_0, \mathbf{x}) \rangle$. If constant prediction $f(\boldsymbol{\theta}_0, \mathbf{x})$ is ignored, the linearized neural network is also described by (49) with gradients as the model features $\boldsymbol{\psi}(\mathbf{x}) = \nabla_{\boldsymbol{\theta}} f(\boldsymbol{\theta}_0, \mathbf{x})$ and the displacement from $\boldsymbol{\theta}_0$ as model parameters $\mathbf{w} = \boldsymbol{\theta} - \boldsymbol{\theta}_0$.

To finalize the connection between the parameter-based setting and kernel-based setting from the main paper, linear model (49) needs to be trained by minimizing quadratic loss on train dataset $\mathcal{D}_N$

$$
\begin{aligned}
L_{\mathcal{D}_N}(\mathbf{w}) &\equiv \frac{1}{2N}\sum_{i=1}^{N}\big(\widehat{f}(\mathbf{w},\mathbf{x}_i)-y_i\big)^2 \\
&= \frac{1}{2N}\sum_{i=1}^{N}\big(\langle\mathbf{w},\boldsymbol{\psi}(\mathbf{x}_i)\rangle-\langle\mathbf{w}^*,\boldsymbol{\psi}(\mathbf{x}_i)\rangle\big)^2 = \frac{1}{2}(\mathbf{w}-\mathbf{w}^*)^T\mathbf{H}(\mathbf{w}-\mathbf{w}^*),
\end{aligned}
\tag{50}
$$

where $\mathbf{H}=\frac{1}{N}\sum_{i=1}^{N}\boldsymbol{\psi}(\mathbf{x}_i)\otimes\boldsymbol{\psi}(\mathbf{x}_i)$ is the Hessian of the train loss. In the following, it will be convenient to denote $\boldsymbol{\Psi}$ the matrix of features calculated on the training dataset $(\boldsymbol{\Psi})_{li}=\psi_l(\mathbf{x}_i)$, and use finite-dimensional notation for inner and outer product in the parameter space: i.e. the Hessian $\mathbf{H}=\frac{1}{N}\boldsymbol{\Psi}\boldsymbol{\Psi}^T$ and empirical kernel matrix $\mathbf{K}=\boldsymbol{\Psi}^T\boldsymbol{\Psi}$. In (50), we have assumed that there exists a parameter value $\mathbf{w}^*$ so that the model (49) completely fits the observations $\boldsymbol{\Psi}^T\mathbf{w}^*=\mathbf{y}$. Considering the typical case $P>N$, this amounts to a feature matrix having full rank $\operatorname{rank}(\boldsymbol{\Psi})=N$[1].

Now, let us proceed with showing how gradient-based optimization fits into the family of spectral algorithms given by (7). We start with the basic example of vanilla Gradient Descent with learning rate $\alpha$, having parameter update rule $\mathbf{w}_{t+1}=\mathbf{w}_t-\alpha\nabla_{\mathbf{w}}L_{\mathcal{D}_N}(\mathbf{w}_t)$. For the quadratic loss (50), this reduces to $\mathbf{w}_{t+1}=\mathbf{w}_t-\alpha\mathbf{H}(\mathbf{w}_t-\mathbf{w}^*)=\mathbf{w}^*+(\mathbf{I}-\alpha\mathbf{H})(\mathbf{w}_t-\mathbf{w}^*)$, or, equivalently,

$$
\mathbf{w}_{t+1}-\mathbf{w}^*=(\mathbf{I}-\alpha\mathbf{H})^t(\mathbf{w}_0-\mathbf{w}^*)=p_t(\mathbf{H})(\mathbf{w}_0-\mathbf{w}^*).
\tag{51}
$$

Here we introduced the polynomial $p_t(\lambda)=(1-\alpha\lambda)^t$ that will prove a useful notation in the following and is related to the profile $h_t(\lambda)$ as $p_t(\lambda)=1-h_t(\lambda)$. To obtain the representation (7) for the learned prediction $\widehat{f}_t(\mathbf{x})=\langle\mathbf{w}_t,\boldsymbol{\psi}(\mathbf{x})\rangle$ we additionally need to set $\mathbf{w}_0=0$. Then,

$$
\widehat{f}_t(\mathbf{x})=\langle\mathbf{w}^*+p_t(\mathbf{H})(\mathbf{w}_0-\mathbf{w}^*),\boldsymbol{\psi}(\mathbf{x})\rangle=\big\langle\big(\mathbf{I}-p_t(\mathbf{H})\big)\mathbf{w}^*,\boldsymbol{\psi}(\mathbf{x})\big\rangle=\langle h_t(\mathbf{H})\mathbf{w}^*,\boldsymbol{\psi}(\mathbf{x})\rangle
\tag{52}
$$

Next, note that polynomial $p_t(\lambda)$ is often called *residual polynomial* due to its normalization at $\lambda=0$ as $p_t(0)=1$, or equivalently $h_t(0)=0$. The latter implies that we can write $h_t(\lambda)=\lambda q_t(\lambda)$ with some polynomial $q_t(\lambda)$ of degree $t-1$. Using an algebraic identity $\mathbf{J}^T\mathbf{J}q(\mathbf{J}^T\mathbf{J})=\mathbf{J}^T q(\mathbf{J}\mathbf{J}^T)\mathbf{J}$ for arbitrary matrix $\mathbf{J}$ and polynomial $q$ allows us to finally obtain (7)

$$
\begin{aligned}
\widehat{f}_t(\mathbf{x}) &= \langle\mathbf{H}q_t(\mathbf{H})\mathbf{w}^*,\boldsymbol{\psi}(\mathbf{x})\rangle=\frac{1}{N}\langle\boldsymbol{\Psi}\boldsymbol{\Psi}^T q_t(\tfrac{1}{N}\boldsymbol{\Psi}\boldsymbol{\Psi}^T)\mathbf{w}^*,\boldsymbol{\psi}(\mathbf{x})\rangle \\
&= \frac{1}{N}\langle\boldsymbol{\Psi}q_t(\tfrac{1}{N}\boldsymbol{\Psi}^T\boldsymbol{\Psi})\boldsymbol{\Psi}^T\mathbf{w}^*,\boldsymbol{\psi}(\mathbf{x})\rangle \\
&\overset{(1)}{=} \mathbf{k}(\mathbf{x})^T\frac{1}{N}q_t(\tfrac{\mathbf{K}}{N})\mathbf{y}=\mathbf{k}(\mathbf{x})^T\mathbf{K}^{-1}\tfrac{1}{N}\mathbf{K}q_t(\tfrac{1}{N}\mathbf{K})\mathbf{y}=\mathbf{k}(\mathbf{x})^T\mathbf{K}^{-1}h_t(\tfrac{1}{N}\mathbf{K})\mathbf{y},
\end{aligned}
\tag{53}
$$

where in (1) we have used that $\boldsymbol{\Psi}^T\mathbf{w}^*=\mathbf{y}$ and $\langle\boldsymbol{\Psi},\boldsymbol{\psi}(\mathbf{x})\rangle=\mathbf{k}(\mathbf{x})^T$. Thus, we have shown that for GD with learning rate $\alpha$ representation (7) holds with $h(\lambda)=1-(1-\alpha\lambda)^t$.

The argument above can be easily extended to the case of Gradient Flow (GF). First note that under GF dynamics $\frac{d}{dt}\mathbf{w}_t=-\nabla L_{\mathcal{D}_N}(\mathbf{w}_t)$ the parameters are $\mathbf{w}_t-\mathbf{w}^*=e^{-\mathbf{H}t}(\mathbf{w}_0-\mathbf{w}^*)$, thus implying $p_t(\lambda)=e^{-\lambda t}$ and $h_t(\lambda)=1-e^{-\lambda t}$. Then, for $q_t(\lambda)=\frac{h_t(\lambda)}{\lambda}=\frac{1-e^{-\lambda t}}{\lambda}$ we also have $\mathbf{J}^T\mathbf{J}q(\mathbf{J}^T\mathbf{J})=\mathbf{J}^T q(\mathbf{J}\mathbf{J}^T)\mathbf{J}$, which can be seen, for example, by from the Taylor expansion $q_t(\lambda)=t\sum_{n=0}^{\infty}\frac{(-t\lambda)^n}{(n+1)!}$. The rest of the argument is unchanged.

Gradient descent is also easily extended to arbitrary first-order iterative optimization algorithms. For all such algorithms, the parameter change $\mathbf{w}_t-\mathbf{w}_0$ on iteration $t$ belongs to an order $t$ Krylov subspace: $\mathbf{w}_t-\mathbf{w}_0\in\operatorname{span}\{\mathbf{H}(\mathbf{w}_0-\mathbf{w}^*),\mathbf{H}^2(\mathbf{w}_0-\mathbf{w}^*),\dots,\mathbf{H}^t(\mathbf{w}_0-\mathbf{w}^*)\}$ (see, e.g. (Nesterov, 2003), page 42). This is equivalent to saying that $\mathbf{w}_{t+1}-\mathbf{w}^*=p_t(\mathbf{H})(\mathbf{w}_0-\mathbf{w}^*)$ with $p_t(\lambda)$ being an arbitrary residual polynomial (i.e. normalized as $p_t(0)=1$). Since in our GD argument we did not use any property of its $p_t(\lambda)$ except for residual normalization, the argument continues to hold, making the representation (7) for all first-order iterative optimization algorithms.

---

[1]A similar assumption was implicitly made in the main paper in the definition of the spectral algorithm (7). Indeed, the existence of inverse $\mathbf{K}^{-1}$ also requires $\operatorname{rank}(\boldsymbol{\Psi})=N$. This is a natural assumption for positive definite kernels $K$: empirical kernel matrix has full rank if evaluated on a set of distinct inputs $\mathbf{x}_i$, which in turn happens almost surely for typical generation processes of $\mathcal{D}_N$ such as i.i.d. drawn $\mathbf{x}_i$. In principle, one can also consider the case of non-full rank of $\mathbf{K}$, or alternatively non-existence of $\mathbf{w}^*$ completely fitting the observations, and replace $\mathbf{K}^{-1}$ in (7) with pseudoinverse. For simplicity, we leave such cases to future work.

## B.2 IMPLEMENTING SPECTRAL ALGORITHMS WITH A PAIR OF GRADIENT FLOWS

Recall that in the section above we get GD residual $p_t(\lambda) = 1 - h_t(\lambda) = e^{-\lambda t}$. An alternative way to get this would be to declare $\mathbf{w}_t - \mathbf{w}^* = p_t(\mathbf{H})(\mathbf{w}_0 - \mathbf{w}^*)$ and then rewrite original Gradient Flow ODE $\frac{d}{dt}\mathbf{w}_t = -\nabla_{\mathbf{w}} L_{\mathcal{D}_N}(\mathbf{w}_t)$ in terms of the residual $p_t(\lambda)$ as

$$\partial_t p_t(\lambda) = -\lambda p_t(\lambda), \quad p_0(\lambda) = 1, \tag{54}$$

with an immediate solution $p_t(\lambda) = e^{-\lambda t}$.

Now, suppose we are given some target profile $\widetilde{h}(\lambda)$ with respective residual $\widetilde{p}(\lambda) = 1 - \widetilde{h}(\lambda) \neq 0$ that needs to be implemented. Our strategy is to design such GF process with residual $q_t(\lambda)$ that converge to the desired profile at long training times $q_\infty(\lambda) \equiv \lim_{t\to\infty} q_t(\lambda) = \widetilde{p}(\lambda)$. The basic GF process given by (54) always converges to full interpolation of the training data: $\lim_{t\to\infty} p_t(\lambda) = \lim_{t\to\infty} e^{-\lambda t} = 0$. We propose to overcome this interpolation property by using two optimization processes – the first process is standard GF (54) converging to 0, and the second GF process will use gradients of the first GF to converge to $\widetilde{p}(\lambda) \neq 0$. These two process are defined by a pair of ODEs

$$\begin{cases} \frac{d}{dt}\mathbf{w}_t &= -\nabla_{\mathbf{w}} L_{\mathcal{D}_N}(\mathbf{w}_t), \\ \frac{d}{dt}\mathbf{u}_t &= -\big(1 + g(t)\big)\nabla_{\mathbf{w}} L_{\mathcal{D}_N}(\mathbf{w}_t), \end{cases} \tag{55}$$

where $\mathbf{w}_t$ and $\mathbf{u}_t$ are the parameters of the first and the second process respectively, and the initial conditions are assumed to be identical $\mathbf{w}_0 = \mathbf{u}_0$. Associating residuals $p_t(\lambda), q_t(\lambda)$ to parameters $\mathbf{w}, \mathbf{u}$, ODE system (55) is rewritten as

$$\begin{cases} \partial_t p_t(\lambda) &= -\lambda p_t(\lambda), \\ \partial_t q_t(\lambda) &= -\big(1 + g(t)\big)\lambda p_t(\lambda), \end{cases} \quad p_0(\lambda) = q_0(\lambda) = 1. \tag{56}$$

Here $g(t)$ is a function controlling the final solution $q_\infty(\lambda)$, and therefore needs to be chosen based on the desired solution $\widetilde{p}(\lambda)$. At a given control function $g(t)$ the final solution $q_\infty(\lambda)$ can be easily found by integrating the second equation as $\int_0^\infty \partial_t q_t(\lambda) = q_\infty(\lambda) - q_0(\lambda)$ and substituting the solution of the basic GF $p_t(\lambda) = e^{-\lambda t}$ into $\partial_t q_t(\lambda)$. Then, setting the final solution to the desired value $q_\infty(\lambda) = \widetilde{p}(\lambda)$ leads to an integral equation on the control $g(t)$

$$\int_0^\infty \big(1 + g(t)\big)\lambda e^{-\lambda t} dt = 1 - \widetilde{p}(\lambda). \tag{57}$$

Since $\int_0^\infty \lambda e^{-\lambda t} dt = 1$, we cancel 1 from both sides, arriving at Laplace transform of $g(t)$ on the left-hand side

$$\int_0^\infty g(t) e^{-\lambda t} dt = -\frac{\widetilde{p}(\lambda)}{\lambda}. \tag{58}$$

Thus, choosing $g(t)$ as an inverse Laplace transform of $-\frac{\widetilde{p}(\lambda)}{\lambda}$ implements the desired spectral algorithm $\widetilde{h}(\lambda)$.

## C  SCALING STATEMENTS

In the section, we give rigorous versions of the scaling statements outlined in Section 5.1. In the end of the section we also provide Proposition 5 as a discrete analog of Proposition 2, which will be required for the Circle model.

**Intuitive derivation.**  Before proceeding with rigorous proofs, let us give a simple intuition behind the scale of sums and integrals stated in Propositions 2 and 5.

We start with the integral case, following notations and assumptions from Proposition 2. To find the scale of the integral $\int_{a_N}^1 |g_N(\lambda)| d\lambda$, let us divide the range of scales $s \in [0, a]$ into many small segments and look at the contribution from a single segment $[s_0, s_0 + \varepsilon]$, corresponding to the interval of eigenvalues $\Lambda_{s_0} = [N^{-\varepsilon}\lambda_0, \lambda_0]$, $\lambda_0 = N^{-s_0}$. Due to the continuity of scaling profile $S^g(s)$ (see Lemma 1 below), we can neglect the change of the $g_N(\lambda)$ on $\Lambda_{s_0}$. Approximating the length of

eigenvalues interval as $|\Lambda_{s_0}| \approx \lambda_0$, we can estimate the contribution to the integral from $[s_0, s_0 + \varepsilon]$ as

$$|g_N(\lambda)| \times |\Lambda_{s_0}| \sim |g_N(\lambda)| \times \lambda_0 \sim N^{-S^g(s_0)-s_0}, \tag{59}$$

which is exactly the expression under the minimum in Proposition 2. To see that only the scales which minimize $G(s) = S^g(s) + s$ give a non-vanishing contribution to the final result, take two scales $s_1, s_2$ such that $G(s_2) - G(s_1) = \delta > 0$. Then, according to (59), the contribution from scale segment $[s_2, s_2 + \varepsilon]$ will be $N^\delta$ times smaller than from the scale segment $[s_1, s_1 + \varepsilon]$, and therefore will vanish in the limit $N \to \infty$.

We can summarize the above with the following simple heuristic: replace $d\lambda$ with $\lambda$ under the integral and maximize the resulting expression to get an estimation of the integral scale. The case of a discrete sum goes along the same lines, leading to the heuristic of replacing the sum $\sum_k$ with current index $k$: $\sum_k |g_N(\lambda_k^{(N)})| \to k \times |g_N(\lambda_k^{(N)})|$, and then maximizing the resulting expression.

**Continuity of scaling profiles.**

**Lemma 1.** *The scaling profile $S^g(s)$, if exists, is a continuous function of $s$.*

*Proof.* Suppose that $S^g$ exists but is not continuous. Then there exists $s_* \geq 0$ and a sequence $s_m \to s_*$ such that $|S^g(s_m) - S^g(s_*)| > c$ for all $m$ and some positive constant $c$. Suppose w.l.o.g. that $S^g(s_m) < S^g(s_*) - c$ for all $m$. Consider some fixed $m$ and choose some sequence $\lambda_N^{(m)}$ of scale $s_m$. In particular, we then have

$$\lambda_N^{(m)} < N^{-s_m+1/m} \text{ and } \lambda_N^{(m)} > N^{-s_m-1/m} \tag{60}$$

for $N > N_m$ with some $N_m$. By definition of the scaling profile, we also have

$$|g_N(\lambda_N^{(m)})| > N^{-S^{(g)}(s_m)-c/2} > N^{-S^{(g)}(s_*)+c/2} \tag{61}$$

for $N$ large enough; say for $N > N_m$ with the same $N_m$ as before. We can assume w.l.o.g. that $N_m$ is monotone increasing. Now define the sequence $\lambda_N$ by

$$\lambda_N = \lambda_N^{(m)}, \quad N_m < N \leq N_{m+1}. \tag{62}$$

This sequence has scale $s_*$, but $|g_N(\lambda_N)| > N^{-S^{(g)}(s_*)+c/2}$ for all sufficiently large $N$, contradicting the fact that that $g_N(\lambda_N)$ must have scale $S^{(g)}(s_*)$. □

**Proof of Proposition 2.** *Part 1.* Let us first show the part of the statement that says that for any $\epsilon > 0$

$$\int_{a_N}^1 |g_N(\lambda)| d\lambda = o(N^{-s_*+\epsilon}). \tag{63}$$

Suppose that this is not the case, and there is $\epsilon > 0$ and a subsequence $N_m$ such that

$$\int_{a_N}^1 |g_N(\lambda)| d\lambda > N_m^{-s_*+\epsilon}. \tag{64}$$

Divide the interval $[0, a]$ into finitely many subintervals $I_r = [b_r, b_{r+1}]$ of length less than $\epsilon/2$. For each subinterval $I_r$, define

$$\lambda_{m,r} = \underset{\lambda: -\log_{N_m} \lambda \in I_r}{\arg\max} |g_{N_m}(\lambda)|. \tag{65}$$

Note that for each $r$ the sequence $m \mapsto -\log_{N_m} \lambda_{m,r}$ takes values in the compact interval $I_r$, so it has a limit point $s_r^* \in I_r$. By going to a subsequence, we can assume w.l.o.g. that the limit point is unique, i.e. is the limit. Then we can define for each $r$ the sequence $\lambda_N^{(r)}$ by setting $\lambda_N^{(r)} = \lambda_{m,r}$ if $N = N_m$ and somehow complementing it for $N \neq N_m$ so that the sequence $\lambda_N^{(r)}$ has scale $s_r^*$. By scaling assumption, we then have

$$g_N(\lambda_N^{(r)}) = o(N^{-S^{(g)}(s_r^*)+\epsilon/2}) \tag{66}$$

and in particular

$$g_{N_m}(\lambda_{m,r}) = o(N_m^{-S^{(g)}(s_r^*)+\epsilon/2}). \tag{67}$$

By definition of $\lambda_{m,r}$,

$$\int_{a_{N_m}}^1 |g_{N_m}(\lambda)|d\lambda = \int_{-\log_{N_m} a_{N_m}}^0 |g_{N_m}(N_m^{-q})|\frac{dN_m^{-q}}{dq}dq \le \sum_r g_{N_m}(\lambda_{m,r})N_m^{-b_r}. \tag{68}$$

It follows that

$$\int_{a_N}^1 |g_{N_m}(\lambda)|d\lambda = o\Big(\sum_r N_m^{-S^{(g)}(s_r^*)+\epsilon/2}N_m^{-b_r}\Big) \tag{69}$$

$$= o\Big(\sum_r N_m^{-(S^{(g)}(s_r^*)+s_r^*)+\epsilon}\Big) \tag{70}$$

$$= o(N_m^{-s_*+\epsilon}) \tag{71}$$

contradicting assumption (64).

*Part 2.* Now we prove the opposite inequality:

$$\int_{a_N}^1 |g_N(\lambda)|d\lambda = \omega(N^{-s_*-\epsilon}). \tag{72}$$

Let $q_* = \arg\min_{0 \le s \le a}(S^{(g)}(s) + s)$. By continuity of $S^{(g)}$, there exists an interval $I = [q_* - \delta, q_* + \delta]$ where $S^{(g)}(s) < S^{(g)}(q_*) + \epsilon/2$. Arguing as in Part 1, we then deduce from the scaling assumption on $S^{(g)}$ that

$$\min_{\lambda:-\log_N \lambda \in I} |g_N(\lambda)| = \omega(N^{-S^{(g)}(q_*)-\epsilon}). \tag{73}$$

It follows that

$$\int_{a_N}^1 |g_N(\lambda)|d\lambda \ge (N^{-q_*+\delta} - N^{-q_*-\delta})\min_{\lambda:-\log_{N_m} \lambda \in I} |g_N(\lambda)| \tag{74}$$

$$= \omega(N^{-S^{(g)}(q_*)-\epsilon}N^{-(q_*-\delta)}) \tag{75}$$

$$= \omega(N^{-S^{(g)}(q_*)-q*-\epsilon}) \tag{76}$$

$$= \omega(N^{-s_*-\epsilon}) \tag{77}$$

as desired. This completes the proof of Proposition 2.

**Discrete spectrum.** Suppose that $\{\lambda_k^{(N)} \in (0,1]\}_{k=1}^N$ is a $N$-dependent, size-$N$ multiset (with possibly repeated elements) such that the sequence of the respective distribution functions $F_N(\lambda) = |\{k \in \overline{1,N} : \lambda < \lambda_k \le 1\}|$ has a scaling profile $S^{(F)}(s)$. Observe, in particular, that in the Circle model with the population eigenvalues $\lambda_l = (|l|+1)^{-\nu}$ the distribution of the empirical eigenvalues $\widehat{\lambda}_k$ (as well as of the population eigenvalues $\lambda_k$ for $k$ restricted to the interval $(-N/2, N/2]$) has the scaling profile $S^{(F)}(s) = -\frac{s}{\nu}$.

**Proposition 5.** *Assuming that $\min\{(\lambda_k^{(N)})_{k=1}^N\} = \omega(N^{-a})$ with some $a > 0$ and $S^{(F)}(s)$ is strictly monotone decreasing, the sequence of sums $\sum_{k=1}^N |g_N(\lambda_k^{(N)})|$ has scale $s_* = \min_{0 \le s \le a}(S^{(g)}(s) + S^{(F)}(s))$.*

The proof of this proposition is analogous to the proof of Proposition 2.

# D   CIRCLE MODEL

Here, we give all our derivations related to Circle model.

## D.1   LOSS FUNCTIONAL

In this section we provide the proof of Theorem 1.

The main technical motivation behind our Circle model is to simplify the empirical kernel matrix $\mathbf{K}$. Indeed, $\mathbf{K}$ becomes a symmetric circulant matrix

$$(\mathbf{K})_{ij} = \sum_{l=-\infty}^{\infty} \lambda_l e^{\mathbf{i}\frac{2\pi l(i-j)}{N}}. \tag{78}$$

To establish the relationship between the (complex) eigendecomposition of empirical kernel matrix (29) and observation vector $(\mathbf{y})_i = y_i$ on one side, and the population spectra $\lambda_l, c_{+,l}, c_{-,l}$ on the other side, we write

$$(\mathbf{K})_{ij} = \sum_{k=0}^{N-1} \sum_{n=-\infty}^{\infty} \lambda_{k+Nn} e^{\mathbf{i}\frac{2\pi(k+Nn)(i-j)}{N}} = \sum_{k=0}^{N-1} \widehat{\lambda}_k e^{\mathbf{i}\frac{2\pi k(i-j)}{N}}. \tag{79}$$

This leads to empirical eigenvalues

$$\widehat{\lambda}_k = \sum_{n=-\infty}^{\infty} \lambda_{k+Nn}, \qquad (\mathbf{u}_k)_i = \frac{1}{\sqrt{N}} e^{\mathbf{i}\frac{2\pi ki}{N}}. \tag{80}$$

Note that empirical eigenvalues $\widehat{\lambda}_k$ turned out to be non-random, which is a consequence of the regularity of training dataset $\mathcal{D}_N$. Observe that each empirical eigenvalue except $\widehat{\lambda}_0$ is twice degenerate: $\widehat{\lambda}_k = \widehat{\lambda}_{N-k}$ for $0 < k < N/2$. This is the consequence of the fact that we took the kernel function $K(x - x')$ to be even.

Turning to the target function, we have

$$y_i = \sigma\varepsilon_i + \sum_l c_l e^{\mathbf{i}l(u+\frac{2\pi i}{N})}. \tag{81}$$

The respective empirical coefficients in the decomposition $\frac{1}{\sqrt{N}}\mathbf{y} = \sum_k \widehat{y}_k \mathbf{u}_k$ are

$$\widehat{y}_k = \frac{1}{N} \sum_{i=0}^{N-1} y_i e^{-\mathbf{i}\frac{2\pi ki}{N}} = \frac{\sigma\varepsilon_k}{\sqrt{N}} + \sum_{n=-\infty}^{\infty} c_{k+Ns} e^{\mathbf{i}(k+Nn)u}. \tag{82}$$

Here $\varepsilon_k = \frac{1}{\sqrt{N}} \sum_i \varepsilon_i e^{-\mathbf{i}\frac{2\pi ki}{N}} \in \mathbb{C}$ are complex Gaussian random variables. They are i.i.d. up to a few dependencies, for example, $\varepsilon_k = \overline{\varepsilon_{N-k}}$ (overline denotes complex conjugation here). Therefore, later we will use the definition of $\varepsilon_k$ in terms of $\varepsilon_i$ to avoid accurate formulation of its statistics.

Finally, let us use the obtained eigendecomposition of the empirical kernel and target to write an expression for the prediction components $\widehat{c}_l$

$$\widehat{c}_l = \frac{1}{2\pi} \int_0^{2\pi} \widehat{f}(x) e^{-\mathbf{i}lx} dx = \lambda_l \sum_{k=0}^{N-1} \frac{h(\widehat{\lambda}_k)}{\widehat{\lambda}_k} \widehat{y}_k \left[ \frac{1}{N} \sum_{i=0}^{N-1} e^{\mathbf{i}k\frac{2\pi i}{N}} e^{-\mathbf{i}l(u+\frac{2\pi i}{N})} \right]$$
$$= \frac{\lambda_l}{\widehat{\lambda}_{k_l}} h(\widehat{\lambda}_{k_l}) e^{-\mathbf{i}lu} \widehat{y}_{k_l}, \tag{83}$$

where $k_l = l \mod N$. Note that representations (83) and (82) define transfer measure $\widehat{\rho}_{ll'}^{(f)}(d\lambda)$ introduced in Section A.3. Due to the regular structure of the training dataset in our setting, the transfer measure also has a specific regular structure.

1. In the basis of Fourier harmonics $\{e^{\mathbf{i}lx}\}_{l=-\infty}^{\infty}$, the information is transferred from $l'$ to $l$ only if $l - l'$ is divisible by $N$ (we can say that such $l, l'$ are compatible).

2. If $l, l'$ are compatible, the information is transferred only through a single empirical eigenvalue $\lambda_{N,k_l}$.

Now, we start to derive the specific form of (44) for our translation-invariant setting. It will be convenient to divide the contribution to the final loss $L_N[h]$ into a bias term and two variance terms responsible for randomness w.r.t. to $u$ and $\varepsilon_i$:

$$L[h] = L^{\text{Bias}}[h] + L^{\text{Var},u}[h] + L^{\text{Var},\varepsilon}[h]. \tag{84}$$

**Bias term.** This term is the loss of the mean prediction with population coefficients

$$\mathbb{E}_{u,\varepsilon}[\widehat{c}_l] = \frac{\lambda_l}{\widehat{\lambda}_{k_l}} h(\widehat{\lambda}_{k_l}) \int_0^{2\pi} e^{-\mathbf{i}lu} \sum_{n=-\infty}^{\infty} c_{l+Nn} e^{\mathbf{i}(l+Nn)u} \frac{du}{2\pi} = \frac{\lambda_l}{\widehat{\lambda}_{k_l}} h(\widehat{\lambda}_{k_l}) c_l, \tag{85}$$

whose substitution into $\|f^*(x) - \mathbb{E}_{u,\varepsilon}[\widehat{f}(x)]\|^2$ gives

$$L_N^{\text{Bias}}[h] = \frac{1}{2} \sum_{l=-\infty}^{\infty} \left(1 - \frac{\lambda_l}{\widehat{\lambda}_{k_l}} h(\widehat{\lambda}_{k_l})\right)^2 |c_l|^2. \tag{86}$$

Here we see that how well the component $l$ can be learned depends on the closeness between $\lambda_l$ and $\widehat{\lambda}_{k_l}$: for $l \ll N$ we have $k_l = l$ and typically $\lambda_l \approx \widehat{\lambda}_l$ (assuming that $\lambda_l$ decay fast with $l$). Thus, the target function can be learned well by setting $h(\lambda) = 1$.

**Noise variance term.** Since the prediction is linear in the noise $\varepsilon_i$, its contribution to the loss comes only from the terms which are quadratic in $\varepsilon_i$. Then, we define the noise variance by the contribution of such terms to the loss. Denoting the contribution of the noise to the prediction as $\widehat{f}^{(\varepsilon)}$, we calculate its second moment

$$\mathbb{E}_{u,\varepsilon}\left[\widehat{f}^{(\varepsilon)}\overline{\widehat{f}^{(\varepsilon)}}\right] = \left(\frac{\lambda_l h(\widehat{\lambda}_{k_l})}{\widehat{\lambda}_{k_l}}\right)^2 \frac{1}{N^2} \sum_{i_1,i_2} e^{-\mathbf{i}k_l \frac{2\pi(i_1-i_2)}{N}} \mathbb{E}_\varepsilon[\varepsilon_{i_1}\varepsilon_{i_2}] = \left(\frac{\lambda_l h(\widehat{\lambda}_{k_l})}{\widehat{\lambda}_{k_l}}\right)^2 \frac{\sigma^2}{N}, \tag{87}$$

where we used that $|\widehat{f}^{(\varepsilon)}|^2$ is independent of $u$ making expectation $\mathbb{E}_u$ trivial. The respective contribution to the loss is

$$L^{\text{Var},\varepsilon}[h] = \frac{1}{2} \sum_{l=-\infty}^{\infty} \left(\frac{\lambda_l h(\widehat{\lambda}_{k_l})}{\widehat{\lambda}_{k_l}}\right)^2 \frac{\sigma^2}{N}. \tag{88}$$

**Dataset variance term.** This part is simply the contribution of the rest of the variance prediction. The respective second moment is

$$\mathbb{E}_{u,\varepsilon}\left[(\widehat{c}_l - \widehat{c}_l^{(\varepsilon)})\overline{(\widehat{c}_l - \widehat{c}_l^{(\varepsilon)})}\right] = \left(\frac{\lambda_l h(\widehat{\lambda}_{k_l})}{\widehat{\lambda}_{k_l}}\right)^2 \int \sum_{n_1,n_2=-\infty}^{\infty} c_{l+Nn_1}\overline{c_{l+Nn_2}} e^{\mathbf{i}N(n_1-n_2)u} \frac{du}{2\pi}$$
$$= \left(\frac{\lambda_l h(\widehat{\lambda}_{k_l})}{\widehat{\lambda}_{k_l}}\right)^2 \sum_{n=-\infty}^{\infty} |c_{l+Nn}|^2. \tag{89}$$

Subtracting the mean (85) from the second moment, we get the dataset variance loss term

$$L^{\text{Var},u}[h] = \frac{1}{2} \sum_{l=-\infty}^{\infty} \left(\frac{\lambda_l h(\widehat{\lambda}_{k_l})}{\widehat{\lambda}_{k_l}}\right)^2 \sum_{n\neq 0} |c_{l+Nn}|^2. \tag{90}$$

**Final expression.** Let us First combine bias and dataset variance terms.

$$L^{\text{Var},u}[h] + L_N^{\text{Bias}}[h] = \frac{1}{2} \sum_{l=-\infty}^{\infty} \left[|c_l|^2 - 2|c_l|^2 \frac{h(\widehat{\lambda}_{k_l})}{\widehat{\lambda}_{k_l}}\lambda_l + \left(\lambda_l \frac{h(\widehat{\lambda}_{k_l})}{\widehat{\lambda}_{k_l}}\right)^2 \sum_{n=-\infty}^{\infty} |c_{l+Nn}|^2\right]$$
$$= \frac{1}{2} \sum_{l=-\infty}^{\infty} \left[(|c_l|^2 - 2|c_l|^2 \frac{h(\widehat{\lambda}_{k_l})}{\widehat{\lambda}_{k_l}}\lambda_l + |c_l|^2 \left(\frac{h(\widehat{\lambda}_{k_l})}{\widehat{\lambda}_{k_l}}\right)^2 \sum_{n=-\infty}^{\infty} \lambda_{l+Nn}^2\right], \tag{91}$$

where we have rearranged the sum over $l$ with fixed $k_l$ in the quadratic term.

Adding the noise variance term, we are now able to write the final expression for the generalization error

$$L[h] = \frac{1}{2} \sum_{l=-\infty}^{\infty} \left[|c_l|^2 \left(1 - 2\frac{h(\widehat{\lambda}_{k_l})}{\widehat{\lambda}_{k_l}}\lambda_l + \left(\frac{h(\widehat{\lambda}_{k_l})}{\widehat{\lambda}_{k_l}}\right)^2 [\lambda_l^2]_N\right) + \frac{\sigma^2}{N}\left(\frac{\lambda_l h(\widehat{\lambda}_{k_l})}{\widehat{\lambda}_{k_l}}\right)^2\right]. \tag{92}$$

where we have used N-deformations (10) notation for the sum $\sum_{n=-\infty}^{\infty} \lambda_{l+Nn}^2 [\lambda_l^2]_N$. As a last step, we observe that the sum over indices $l$ the same fixed $k_l = k \mod N$ again leads to N-deformations (10), allowing to rewrite the full sum over $l \in \mathbb{Z}$ into the sum over $k_l \in 0, 1, \ldots, N-1$. Then, denoting $k_l$ simply as $k$, we have

$$L[h] = \frac{1}{2} \sum_{k=0}^{N-1} \left[ \left( \frac{\sigma^2}{N} + [|c_k|^2]_N \right) \frac{[\lambda_k^2]_N}{[\lambda_k]_N^2} h^2(\widehat{\lambda}_k) - 2 \frac{[\lambda_k |c_k|^2]_N}{[\lambda_k]_N} h(\widehat{\lambda}_k) + [|c_k|^2]_N \right], \qquad (93)$$

which proves Theorem 1.

Let us now describe the optimal learning algorithm (9). Note that the functional (93) is fully local, and therefore the optimal algorithm is well defined and is given by pointwise minimization at each $\widehat{\lambda}_k$. The resulting optimal algorithm $h^*(\widehat{\lambda}_k)$ is

$$h^*(\widehat{\lambda}_k) = \frac{[\lambda_k |c_k|^2]_N [\lambda_k]_N}{\left( \frac{\sigma^2}{N} + [|c_k|^2]_N \right) [\lambda_k^2]_N}. \qquad (94)$$

It may also be convenient to give a "completed square" form of the loss functional $L[h] = \delta L[h - h^*] + L[h^*]$, $\delta L[h'] \geq 0$ that separates the minimal possible error $L[h^*]$ with an excess positive error if the algorithm is non-optimal $h - h^* \neq 0$

$$L[h] = \frac{1}{2} \sum_{k=0}^{N-1} \left[ \left( \frac{\sigma^2}{N} + [|c_k|^2]_N \right) \frac{[\lambda_k^2]_N}{[\lambda_k]_N^2} \left( h(\widehat{\lambda}_k) - h^*(\widehat{\lambda}_k) \right)^2 \right.$$
$$\left. + [|c_k|^2]_N - \frac{[\lambda_k |c_k|^2]_N^2}{\left( \frac{\sigma^2}{N} + [|c_k|^2]_N \right) [\lambda_k^2]_N} \right]. \qquad (95)$$

Finally, we note that one can use translation symmetry $k \to k + N$ of N-deformations (10) in order to shift the summation in (93) to values $-\frac{N}{2} \leq k \leq \frac{N}{2}$, for example, $k \in \{-\lfloor \frac{N}{2} \rfloor, \ldots, \lceil \frac{N}{2} \rceil - 1\}$. Like in (21), we denote such summation range simply as $\sum_{k=-\frac{N}{2}}^{\frac{N}{2}}$ . The purpose of this shift is to put all high empirical spectral quantities $[\lambda_k^a |c_k|^{2b}]_N$ in the region $|k| \ll N$, allowing to write, for example, $\widehat{\lambda}_k = O\big((|k|+1)^{-\nu}\big)$.

## D.2 POWER-LAW ANSATZ: NOISY OBSERVATIONS

Now, we turn to a more detailed analysis of the Circle model. As in the main paper, we separately consider the noisy $\sigma^2 > 0$ case in this section and the noiseless $\sigma^2 = 0$ case in the next Section D.3. Recall that we adapt basic power-law spectrum $\lambda_l = l^{-\nu}$, $c_l^2 = l^{-\kappa-1}$, $l \geq 1$ since for the circle model the population is naturally indexed by the whole integer set $\mathbb{Z}$, leading to

$$\lambda_l = (|l|+1)^{-\nu}, \quad c_l^2 = (|l|+1)^{-\kappa-1}, \quad l \in \mathbb{Z}. \qquad (96)$$

The purpose of the current section is to show the equivalence of circle model and NMNO models, thus proving the respective part of Theorem 2. The intuition behind NMNO relies on the closeness of population and empirical spectral distributions in the eigenspaces distant from the spectrum edge $|l| \ll N$. Thus, we need to compare N-deformations $[\lambda_k^a |c_k|^{2b}]_N$ with their population counterparts $\lambda_k^a |c_k|^{2b}$, considering the values $|k| \leq \frac{N}{2}$ relevant for the loss functional (11). From the definition (10) we get

$$[\lambda_k^a |c_k|^{2b}]_N = \lambda_k^a |c_k|^{2b} + \sum_{n \neq 0} \lambda_{k+Nn}^a |c_{k+Nn}|^{2b}$$
$$= \lambda_k^a |c_k|^{2b} + \sum_{n_1=1}^{\infty} \frac{N^{-a\nu - b(\kappa+1)}}{\left( n_1 + \frac{1+k}{N} \right)^{a\nu + b(\kappa+1)}} + \sum_{n_2=1}^{\infty} \frac{N^{-a\nu - b(\kappa+1)}}{\left( n_2 + \frac{1-k}{N} \right)^{a\nu + b(\kappa+1)}} \qquad (97)$$
$$= \lambda_k^a |c_k|^{2b} + O(N^{-a\nu - b(\kappa+1)}) = \lambda_k^a |c_k|^{2b} \left( 1 + O(\tau^{a\nu + b(\kappa+1)}) \right),$$

where, in the last line, we have assumed that $a\nu + b(\kappa+1) > 1$ so that both $n_1$ and $n_2$ series are converging. Also, we have recalled the notation $\tau = \frac{|k|+1}{N}$ introduced in Section 5.3.

It turns out that the relation $\left[\lambda_k^a |c_k|^{2b}\right]_N = \lambda_k^a |c_k|^{2b} + O(N^{-a\nu - b(\kappa+1)})$ is sufficient to establish equivalence to NMNO model. For that, write the Circle model loss functional (11) as NMNO functional $L^{(\mathrm{nmno})}[h]$, defined in (17), plus two corrections terms

$$L[h] = L^{(\mathrm{nmno})}[h] + \delta L_{\mathrm{alg}}[h] + \delta L_{\mathrm{coeff}}[h], \tag{98}$$

where the correction due to the displacements between the population and empirical eigenvalues in the argument of the learning algorithm is

$$\delta L_{\mathrm{alg}}[h] = \frac{1}{2} \sum_{k=-\frac{N}{2}}^{\frac{N}{2}} \left[ \frac{\sigma^2}{N}\left(h^2(\widehat{\lambda}_k) - h^2(\lambda_k)\right) + c_l^2\left(\left(1 - h(\widehat{\lambda}_k)\right)^2 - \left(1 - h(\lambda_k)\right)^2\right)\right], \tag{99}$$

and the correction due to difference between population $\lambda_k^a |c_k|^{2b}$ and empirical $\left[\lambda_k^a |c_k|^{2b}\right]_N$ coefficients in the loss functional

$$\delta L_{\mathrm{coeff}}[h] = \frac{1}{2} \sum_{k=-\frac{N}{2}}^{\frac{N}{2}} \left[ \frac{\sigma^2}{N} \frac{\left[\lambda_k^2\right]_N - \left[\lambda_k\right]_N^2}{\left[\lambda_k\right]_N^2} h^2(\widehat{\lambda}_k) + \left(\left[|c_k|^2\right]_N \frac{\left[\lambda_k^2\right]_N}{\left[\lambda_k\right]_N^2} - |c_k|^2\right) h^2(\widehat{\lambda}_k)\right.$$
$$\left. - 2\left(\frac{\left[\lambda_k |c_k|^2\right]_N}{\left[\lambda_k\right]_N} - |c_k|^2\right) h(\widehat{\lambda}_k) + \left[|c_k|^2\right]_N - |c_k|^2 \right]. \tag{100}$$

From this point, our strategy is to specify the scales of all the terms of NMNO functional $L^{(\mathrm{nmno})}[h]$, and of the two corrections $\delta L_{\mathrm{alg}}[h], \delta L_{\mathrm{coeff}}[h]$. Then, we can invoke the scaling argument of Proposition 2 (actually its discrete version in Proposition 5) to show that all the correction terms give negligible contribution to the loss.

First, recall from Section 5.1 that the scaling of NMNO terms is given by

$$S\left[\frac{\sigma^2}{N} h^2(\lambda_k)\right] = 1 + 2S^{(h)}(s), \tag{101}$$

$$S\left[c_k^2\left(1 - h(\lambda_k)\right)^2\right] = \frac{\kappa+1}{\nu} s + 2S^{(1-h)}(s). \tag{102}$$

Next, we proceed to the $\delta L_{\mathrm{alg}}[h]$ correction. To bound its terms one needs a certain smoothness assumption on $h(\lambda)$. Currently, in Theorem 2, we require that the maps $\log \lambda \mapsto \log|h(\lambda)|$ and $\log \lambda \mapsto \log|1 - h(\lambda)|$ are globally Lipschitz, but maybe a weaker smoothness condition is possible. To understand the application of this condition, take some function $g(x)$ such that a mapping $\log x \to \log g(x)$ is Lipschitz with constant $C$, i.e. $\left|\log \frac{g(x+\Delta x)}{g(x)}\right| \leq C\left|\log \frac{x+\Delta x}{x}\right|$. Then, taking any constant $C' > C$, there is $\delta > 0$ such that for all $\left|\frac{\Delta x}{x}\right| < \delta$ we have $\left|g(x+\Delta x) - g(x)\right| < C'g(x)\left|\frac{\Delta x}{x}\right|$. Coming back to the difference between empirical and population eigenvalues and using (97) gives $\frac{\widehat{\lambda}_k - \lambda_k}{\lambda_k} = O(\tau^\nu)$, and therefore $|h^2(\widehat{\lambda}_k) - h^2(\lambda_k)| = O\left(h^2(\lambda_k)\tau^\nu\right)$ (and similar estimate for $(1 - h(\lambda))^2$). Recalling the scale $S[\tau] = S\left[\frac{|k|+1}{N}\right] = 1 - \frac{s}{\nu}$, we get a bound on the scale of the respective correction terms

$$S\left[\frac{\sigma^2}{N}\left(h^2(\widehat{\lambda}_k) - h^2(\lambda_k)\right)\right] \geq 1 + \nu - s + 2S^{(h)}(s), \tag{103}$$

$$S\left[c_k^2\left(\left(1 - h(\widehat{\lambda}_k)\right)^2 - \left(1 - h(\lambda_k)\right)^2\right)\right] \geq \nu + \frac{\kappa+1-\nu}{\nu} s + 2S^{(1-h)}(s). \tag{104}$$

Finally, repetitive application of (97) to all the terms of $\delta L_{\mathrm{coeff}}[h]$ gives the remaining scales

$$S\left[\frac{\sigma^2}{N} \frac{\left[\lambda_k^2\right]_N - \left[\lambda_k\right]_N^2}{\left[\lambda_k\right]_N^2} h^2(\widehat{\lambda}_k)\right] \geq 1 + (\nu - s) + 2S^{(h)}(s), \tag{105}$$

$$S\left[\left(\left[|c_k|^2\right]_N \frac{\left[\lambda_k^2\right]_N}{\left[\lambda_k\right]_N^2} - |c_k|^2\right) h^2(\widehat{\lambda}_k)\right] \geq (\kappa+1) \wedge \left(\nu + \frac{\kappa+1-\nu}{\nu} s\right) + 2S^{(h)}(s), \tag{106}$$

$$S\left[\left(\frac{\left[\lambda_k |c_k|^2\right]_N}{\left[\lambda_k\right]_N} - |c_k|^2\right) h(\widehat{\lambda}_k)\right] \geq \nu + \frac{\kappa+1-\nu}{\nu} s + S^{(h)}(s), \tag{107}$$

$$S\left[\left[|c_k|^2\right]_N - |c_k|^2\right] \geq \kappa + 1. \tag{108}$$

Here, all the estimations are relatively straightforward, except for, possibly, the bound (106) that involves two values depending on whether $\kappa + 1 > \nu$ or the opposite (a reminiscent of overlearning transition of Section 5.3 here!). We demonstrate the bounding of this term in detail

$$
\begin{aligned}
\left[|c_k|^2\right]_N \frac{[\lambda_k^2]_N}{[\lambda_k]_N^2} &= |c_k|^2 \left(1 + O(\tau^{\kappa+1})\right) \frac{\lambda_k^2\left(1 + O(\tau^{2\nu})\right)}{\left(\lambda_k\left(1 + O(\tau^\nu)\right)\right)^2} \\
&= |c_k|^2 \left(1 + O(\tau^{\kappa+1}) + O(\tau^\nu)\right) = |c_k|^2 \left(1 + O\left(\tau^{(\kappa+1)\wedge\nu}\right)\right) \\
&= |c_k|^2 + O\left(\frac{(|k|+1)^{0\wedge(\nu-\kappa-1)}}{N^{(\kappa+1)\wedge\nu}}\right).
\end{aligned}
\tag{109}
$$

This immediately implies the bound on the scale of $\left[|c_k|^2\right]_N \frac{[\lambda_k^2]_N}{[\lambda_k]_N^2} - |c_k|^2$ used in (106).

Now, having specified the scale of all the corrections, our task is to show that they contribute negligibly to the loss. In other words, the scale of the total contribution of all the corrections has to be strictly lower than that of NMNO loss. Using Proposition 5 this is written as

$$
\min_{0\le s\le\nu}\left[\left(1 - \tfrac{s}{\nu} + 2S^{(h)}(s)\right) \wedge \left(\tfrac{\kappa}{\nu}s + 2S^{(1-h)}(s)\right)\right]
\tag{110}
$$

$$
< \min_{0\le s\le\nu}\left[\left(1 + (\nu - s) + 2S^{(h)}(s)\right)\right.
\tag{111}
$$

$$
\wedge \left((\kappa+1) \wedge \left(\nu + \tfrac{\kappa+1-\nu}{\nu}s\right) + 2S^{(h)}(s)\right)
\tag{112}
$$

$$
\wedge \left(\nu + \tfrac{\kappa+1-\nu}{\nu}s + S^{(h)}(s)\right)
\tag{113}
$$

$$
\wedge \left(\nu + \tfrac{\kappa+1-\nu}{\nu}s + 2S^{(h)}(s)\right)
\tag{114}
$$

$$
\left.\wedge (\kappa+1) - \tfrac{s}{\nu}\right].
\tag{115}
$$

It is easy to see that this strict inequality can only be violated (by becoming equality) when the mins are attained at $s = \nu$ so that the r.h.s. of (105) is equal to the r.h.s. of (101)). However, this possibility is excluded by hypothesis of the theorem. This completes the proof of Theorem 2 for the Circle model.

We remark that the Lipschitz condition in Theorem 2, used to compare algorithm $h(\lambda)$ evaluated at the population and empirical eigenvalues, is required only for the discrete (Circle) problem. It is easy to check that this condition holds for the KRR algorithm with $h_\eta(\lambda) = \frac{\lambda}{\lambda+\eta}$. However, it is violated for GF with $h_t(\lambda) = 1 - e^{-\lambda t}$: for time $t \sim N^{s_t}$ the mapping $\log\lambda \to \log(1 - h_t(\lambda)) = -\lambda t$ is not Lipschitz on the scales $s < s_t$. Fortunately, $(1 - h_t(\lambda))^2$ on such scales is exponentially small, and the contribution to the loss from corresponding terms, both NMNO and its corrections, can be ignored. Thus, the equivalence between Circle and NMNO models still holds for GF algorithm. We expect that the Lipschitz condition of Theorem 2 can be weakened to take into account GF algorithm but leave it for future work.

## D.3 POWER-LAW ANSATZ: NOISELESS OBSERVATIONS

In this section, we derive the results presented in Section 5.3, while also giving full versions of the results that were discussed only partially in the main paper. For the convenience of exposition, the order in this section repeats that of Section 5.3.

### D.3.1 PERTURBATIVE EXPANSION OF THE LOSS FUNCTIONAL

In the main paper, we relied on equation (21) as a starting point to quite easily conclude that the loss localizes on the smallest spectral scale $s = 0$ in the saturated phase $\kappa > 2\nu$ while localizing on the highest scale $s = \nu$ in the non-saturated phase $\kappa < 2\nu$. Essentially, equation (21) ignores the details of the problem at subspaces corresponding to small eigenvalues $\widehat{\lambda}_k \sim \widehat{\lambda}_{N/2}$ (and $|k| \sim N$), while providing a simple estimation of different loss functional terms for large eigenvalue subspaces with $\widehat{\lambda}_k \gg \widehat{\lambda}_{N/2}$ (and $|k| \ll N$). Alternatively, if one starts from exact loss functional (11), there seems to be no clear path to deducing the existence of saturated and non-saturated phases, and obtaining their convergence rates.

The above discussion illuminates the importance of perturbative expansion of the loss functional in the small parameter $\tau = \frac{|k|+1}{N}$, or, in other words, perturbative corrections $\widehat{\lambda}_k - \lambda_k$ and $\widehat{c}_k - c_k$ of population spectral distributions in the presence of finite dataset size $N$. For the circle model, the effect of finite dataset size $N$ is captured by deviation of N-deformations $[\lambda_k^a|c_k|^{2b}]_N$ from their population counterparts $\lambda_k^a|c_k|^{2b}$. The simplest form of this deviation was already obtained in equation (97), which we will use below to derive equation (21).

Let us start by writing down the loss functional in "completed square" from (95) and in the absence of observation noise

$$L[h] = \frac{1}{2} \sum_{k=-\frac{N}{2}}^{\frac{N}{2}} \left[ \frac{[|c_k|^2]_N [\lambda_k^2]_N}{([\lambda_k]_N)^2} \left(h(\widehat{\lambda}_k) - h^*(\widehat{\lambda}_k)\right)^2 + [|c_k|^2]_N - \frac{([\lambda_k|c_k|^2]_N)^2}{[|c_k|^2]_N [\lambda_k^2]_N} \right]. \quad (116)$$

Note that the first term here, i.e. the factor in front of $\left(h(\widehat{\lambda}_k) - h^*(\widehat{\lambda}_k)\right)^2$, was already estimated in the second line of (109): it agrees with the factor $\frac{|c_k|^2}{2}(1+o(\tau))$ appearing in (21) up to replacement $O(\tau^{(\kappa+1)\wedge\nu}) = o(\tau)$ valid due to $(\kappa+1) \wedge \nu > 1$. We use this simplification in (21) because we do not need a more accurate estimation of the correction term at this moment.

The second term of (21) corresponds to the generalization error of the optimal algorithm, which allows to denote it as $L_k[h^*]$ since $L[h^*] = \sum_{k=-N/2}^{N/2} L_k[h^*]$. This term is estimated as follows

$$\begin{aligned}
L_k[h^*] &= [|c_k|^2]_N - \frac{([\lambda_k|c_k|^2]_N)^2}{[|c_k|^2]_N [\lambda_k^2]_N} \\
&= |c_k|^2 \left(1 + O(\tau^{\kappa+1})\right) - \frac{|c_k|^4 \lambda_k^2 \left(1 + O(\tau^{\kappa+1+\nu})\right)}{|c_k|^2 \lambda_k^2 \left(1 + O(\tau^{\kappa+1}) + O(\tau^{2\nu})\right)} \\
&= |c_k|^2 \left( O(\tau^{\kappa+1}) + O(\tau^{\kappa+1+\nu}) + O(\tau^{2\nu}) \right) \\
&= (\tau N)^{-\kappa-1} \left( O(\tau^{\kappa+1}) + O(\tau^{2\nu}) \right) \\
&= N^{-\kappa-1} \left( O(1) + O(\tau^{2\nu-\kappa-1}) \right).
\end{aligned} \quad (117)$$

### D.3.2 Optimally scaled learning algorithms

In Section 5.3 we mentioned that the rate $O(N^{-\kappa})$ of the optimal algorithm $h^*(\lambda)$ in the non-saturated phase also holds for learning algorithms $h(\lambda)$ within a suitable neighbourhood of $h^*(\lambda)$, characterized by a condition $|h(\widehat{\lambda}_k) - h^*(\widehat{\lambda}_k)|^2 = o(\tau^\kappa)$. In this section, we derive this result while also providing a more systematic discussion of learning algorithms that do not destroy the rate of the optimal algorithm.

Let us call an algorithm $h(\lambda)$ *optimally scaled* if the scale $S[L[h]]$ of associated generalization error $L[h]$ is the same as that of the optimal algorithm $h^*(\lambda)$

$$S[L[h]] = S[L[h^*]]. \quad (118)$$

While one might try to find all algorithms satisfying (118), here we take a less ambitious approach by considering a simple family of conditions on $|h(\lambda) - h^*(\lambda)|$ and then choosing the weakest condition within the family. Specifically, for any two constants $a, b$, consider the following bound on the scale of deviation of an algorithm $h(\lambda)$ from the optimal one:

$$S[|h(\lambda) - h^*(\lambda)|^2] \geq as + b, \quad s \in [0, \nu], \quad (119)$$

which is a slightly weaker (e.g. up to a $\log N$ factors) version of $|h(\lambda) - h^*(\lambda)|^2 = O(\lambda^{-a}N^{-b})$. Here we limited the scale of $\lambda$ to $s \in [0, \nu]$ because this is precisely the interval of scales occupied by the set of empirical eigenvalues $\{\widehat{\lambda}_k\}_{k=0}^{N-1}$ passed as an input to algorithms $h, h^*$.

It is easy to check whether all algorithms satisfying condition (119) are optimally scaled. First, we can equivalently rewrite (118) as $S[L[h] - L[h^*]] \geq S[L[h^*]]$. Then, applying Proposition 5 to (21) yields

$$S\big[L[h] - L[h^*]\big] = \min_{0 \le s \le \nu} \Big( S\big[|c_k|^2\big] + S\big[|h(\widehat{\lambda}_k) - h^*(\widehat{\lambda}_k)|^2\big] - \frac{s}{\nu} \Big)$$

$$\ge \min_{0 \le s \le \nu} \Big( (\kappa + 1)\frac{s}{\nu} + as + b - \frac{s}{\nu} \Big) = \begin{cases} b, & a \ge -\frac{\kappa}{\nu} \\ b + \kappa + a\nu, & a < -\frac{\kappa}{\nu}. \end{cases} \tag{120}$$

According to the above calculation, the set of all pairs $a, b$ that guarantee algorithms under condition (119) to be optimally scaled is given by $A = \{(a,b) \in \mathbb{R}^2 \mid b + 0 \wedge (\kappa + a\nu) \ge S\big[L[h^*]\big]\}$. Now, if we wish to pick a pair $(a,b) \in A$ such that condition (119) is the weakest at a spectral scale $s$, we have to minimize $as + b$. Fortunately, there is a unique pair $(a^*, b^*)$ that provides the weakest condition across all relevant spectral scales

$$(a^*, b^*) = \Big( -\frac{\kappa}{\nu}, S\big[L[h^*]\big] \Big) = \arg\min_{(a,b) \in A} as + b \quad \forall s \in [0, \nu]. \tag{121}$$

Applying this result to saturated and non-saturated phases, with respective convergence rates $O(N^{-2\nu})$ and $O(N^{-\kappa})$, gives the desired conditions for optimally scaled algorithms

$$S\big[|h(\lambda) - h^*(\lambda)|^2\big] \ge -\frac{\kappa}{\nu}s + \kappa \quad \text{in the non-saturated phase } \kappa < 2\nu, \tag{122}$$

$$S\big[|h(\lambda) - h^*(\lambda)|^2\big] \ge -\frac{\kappa}{\nu}s + 2\nu \quad \text{in the saturated phase } \kappa > 2\nu. \tag{123}$$

**A stronger version of optimally scaled algorithm condition.** Recall that the loss of the optimal algorithm $h^*$ in saturated and non-saturated phases localize at $s = 0$ and $s = \nu$ respectively. However, conditions (122), (123) do not provide the same localization property for the excess error $L[h] - L[h^*]$. To see this, take $|h(\lambda) - h^*(\lambda)|^2 = O(\lambda^{-\frac{\kappa}{\nu}} N^{-\kappa \wedge 2\nu})$, which corresponds to values $(a^*, b^*)$ given in (121). Substituting these values in excess error scale calculation (120) makes the function under the minimum $s$-independent, implying that the excess loss localizes on all spectral scales $s \in [0, \nu]$. Such spread of localization scales introduces a logarithmic factor in the error rate. Taking, for simplicity, $|h(\widehat{\lambda}_k) - h^*(\widehat{\lambda}_k)|^2 = (|k| + 1)^{-\kappa} N^{-\kappa \wedge 2\nu}$, and using (21) gives

$$L[h] - L[h^*] = \sum_{k=-\frac{N}{2}}^{\frac{N}{2}} c_k^2 (1 + o(\tau))|h(\widehat{\lambda}_k) - h^*(\widehat{\lambda}_k)|^2$$

$$= N^{-\kappa \wedge 2\nu} \sum_{k=-\frac{N}{2}}^{\frac{N}{2}} \frac{1 + o(\tau)}{|k| + 1} = O(N^{-\kappa \wedge 2\nu} \log N). \tag{124}$$

To avoid these issues, let us introduce a slightly stronger version of conditions (122),(123), specified by picking a small parameter $\varepsilon > 0$:

$$|h(\lambda) - h^*(\lambda)|^2 = O\big((\lambda^{\frac{1}{\nu}} N)^{-\kappa - \varepsilon}\big) \quad \text{in the non-saturated phase } \kappa < 2\nu, \tag{125}$$

$$|h(\lambda) - h^*(\lambda)|^2 = O\big(\lambda^{-\frac{\kappa}{\nu} + \varepsilon} N^{-2\nu}\big) \quad \text{in the saturated phase } \kappa > 2\nu. \tag{126}$$

For any $\varepsilon > 0$, the above conditions guarantee localization of the excess error either at $s = 0$ (saturated phase) or $s = \nu$ (non-saturated phase), as can be seen from (120). Also, using these conditions in computation like (124) produces the rate of the optimal algorithm without $\log N$ factor $L[h] - L[h^*] = O(N^{-\kappa \wedge 2\nu})$.

Finally, let us comment on $|h(\widehat{\lambda}_k) - h^*(\widehat{\lambda}_k)|^2 = o(\tau^\kappa)$ condition for non-saturated phase, mentioned in the main paper. Since $\tau = (\lambda_k^{\frac{1}{\nu}} N)^{-1}$, in many cases one can replace $o(\tau^\kappa)$ with $O(\tau^{\kappa + \varepsilon})$ with some $\varepsilon$ and thus satisfy condition (125). When no $\varepsilon > 0$ can provide such replacement, it is possible to show that the rate of the excess error remains $L[h] - L[h^*] = O(N^{-\kappa})$, although with a more technically involved version of computation (124).

### D.3.3 SATURATED PHASE $\kappa > 2\nu$

When discussing the saturated phase in Section 5.3, we first deduced from $O(\tau^{\#})$ terms in equation (21) that the loss will localize on the scale $s = 0$ (corresponding to $|k| \sim 1$), and then stated that the loss of the optimal algorithm can be achieved by optimally regularized KRR. In this section, we derive an asymptotic expression of the loss functional in the saturated phase, which both explains the statements made in the main paper and gives a more systematic picture of the loss-algorithm relations in the saturated phase.

**Generalization error of optimal algorithm** $L[h^*]$. The asymptotic of $L[h^*]$ originates from $O(\tau^{2\nu-\kappa-1})$ term in equation (21), which dominates the whole loss functional for $\kappa > 2\nu$. To verify that this is indeed the case, we need a more accurate characterization of N-deformation correction terms than in (97). For that, we recognize that the sums over $n_1, n_2$ in the second line of (97) are Hurwitz zeta functions $\zeta(\alpha, x) \equiv \sum_{n=0}^{\infty} (n + x)^{-\alpha}$ evaluated at $\alpha = a\nu + b(\kappa + 1)$ and two specific values of $x$:

$$\left[\lambda_k^a |c_k|^{2b}\right]_N \overset{\alpha = a\nu + b(\kappa+1)}{=} \lambda_k^a |c_k|^{2b} + N^{-\alpha}\left(\zeta\left(\alpha, 1 + \tfrac{1+k}{N}\right) + \zeta\left(\alpha, 1 + \tfrac{1-k}{N}\right)\right). \tag{127}$$

Substituting Taylor expansion $\zeta(\alpha, 1 + \epsilon) = \zeta(\alpha) - \alpha\zeta(\alpha + 1)\epsilon + O(\epsilon^2)$ of Hurwitz zeta function at $x = 1$ into (127) we get a more detailed version of (97)

$$\left[\lambda_k^a |c_k|^{2b}\right]_N = \lambda_k^a |c_k|^{2b} + 2\zeta(\alpha)N^{-\alpha} + O(N^{-\alpha-1}) + O(\tau^2 N^{-\alpha}). \tag{128}$$

Equipped with (128), we turn to derivation of the leading asymptotic of $O(\tau^{2\nu-\kappa-1})$ term dominating the loss. By looking at (117), we can see that the term of interest comes from finite $N$ corrections of $\left[\lambda_k^2\right]_N$. Then, mimicking the derivation (117) and using (128) for $\left[\lambda_k^2\right]_N$, we have

$$\begin{aligned} L_k[h^*] &= |c_k|^2\left(1 + O(\tau^{\kappa+1})\right) - \frac{|c_k|^2\lambda_k^2\left(1 + O(\tau^{\kappa+1})\right)}{\lambda_k^2 + 2\zeta(2\nu)N^{-2\nu}\left(1 + O(N^{-1}) + O(\tau^2)\right)} \\ &= 2\zeta(2\nu)\frac{|c_k|^2}{\lambda_k^2}N^{-2\nu}\left(1 + O(N^{-1}) + O(\tau^2)\right) + O(N^{-\kappa-1}). \end{aligned} \tag{129}$$

Substituting the above in the loss functional sum results gives

$$\begin{aligned} L[h^*] &= \frac{1}{2}\sum_{k=-\frac{N}{2}}^{\frac{N}{2}}\left[2\zeta(2\nu)\frac{|c_k|^2}{\lambda_k^2}N^{-2\nu}\left(1 + O(N^{-1}) + O(\tau^2)\right) + O(N^{-\kappa-1})\right] \\ &= \frac{1}{2}2\zeta(2\nu)N^{-2\nu}\left(1 + o(1)\right)\sum_{k=-\frac{N}{2}}^{\frac{N}{2}}\frac{|c_l|^2}{\lambda_l^2} \\ &= \frac{1}{2}2\zeta(2\nu)N^{-2\nu}\left(1 + o(1)\right)\sum_{l=-\infty}^{\infty}\frac{|c_l|^2}{\lambda_l^2}. \end{aligned} \tag{130}$$

Here in the last equality we extended the summation to all population eigenspaces due to convergence of the series $\sum_{l=-\infty}^{\infty}\frac{|c_l|^2}{\lambda_l^2}$ at $\kappa > 2\nu$, which reflects localization of the error on scale $s = 0$, corresponding to $|l| \sim 1$.

Note that in (130) we could have evaluated the sum over the population spectra as

$$\sum_{l=-\infty}^{\infty}\frac{|c_l|^2}{\lambda_l^2} = \sum_{l=-\infty}^{\infty}(|l| + 1)^{2\nu-\kappa-1} = 2\zeta(\kappa + 1 - 2\nu) - 1. \tag{131}$$

However, we view the version with the sum as being more general. Intuitively, it stays valid in the case of population spectra having power-law form only asymptotically: $\lambda_l = (|l| + 1)^{-\nu}(1 + o(1))$ and $|c_l|^2 = (|l| + 1)^{-\kappa-1}(1 + o(1))$. In that case, the value of the sum is not fixed by the asymptotics of $\lambda_l, |c_l|^2$ and may vary substantially. In contrast, we can see from computation (129) and corrections to N-deformations (97) and (128) that the factor $2\zeta(2\nu)$ is determined by population spectrum $\lambda_l$ with indices $l$ near the values $l = N, -N, 2N, -2N, \ldots$. Therefore, the factor $2\zeta(2\nu)$

in the loss is determined purely by asymptotic behavior of $\lambda_l$. Overall, this creates an interesting situation where the loss (130) is localized on the smallest scale $s = 0$, but the asymptotic shape of the population spectrum almost fully specifies the generalization error asymptotic $L[h^*] = CN^{-2\nu}(1 + o(1))$, with only the constant factor $\sum_l \frac{|c_l|^2}{\lambda_l^2}$ determined by the details living on the localization scale $s = 0$.

**Full generalization error $L[h]$.** The full generalization error is obtained by combining $L[h^*]$ with the part of (116) associated with deviation from the optimal algorithm $h(\lambda) - h^*(\lambda)$. We will consider only the algorithms $h(\lambda)$ with mild deviation from the optimal, as given by condition (125). As demonstrated in Section D.3.2, this condition guarantees the rate not worse than the rate $O(N^{-2\nu})$ of the optimal algorithm, and also localization of the excess error on the same scale $s = 0$.

Having ensured localization of the excess error on the scale $s = 0$, let us first look at the perturbative expression of the optimal algorithm $h^*(\lambda)$ at this scale. Substituting (128) into (12), we get

$$h^*(\widehat{\lambda}_k) = 1 + 2\zeta(\nu)\frac{N^{-\nu}}{\lambda_k}\big(1 + O(N^{-1}) + O(\tau)\big). \tag{132}$$

Using the above expression of the optimal algorithm, we compute the full loss functional as follows

$$
\begin{aligned}
L[h] &= \frac{1}{2}\sum_{k=-\frac{N}{2}}^{\frac{N}{2}} |c_k|^2\big(1 + o(\tau)\big)\Big(h(\widehat{\lambda}_k) - 1 - 2\zeta(\nu)\frac{N^{-\nu}}{\widehat{\lambda}_k}\big(1 + O(\tfrac{1}{N}) + O(\tau)\big)\Big)^2 \\
&\quad + \frac{1}{2}\sum_{k=-\frac{N}{2}}^{\frac{N}{2}} \frac{|c_k|^2}{\lambda_k^2} 2\zeta(2\nu)N^{-2\nu}(1 + o(1)) \\
&= \frac{1}{2}N^{-2\nu}(1 + o(1))\sum_{k=-\frac{N}{2}}^{\frac{N}{2}} \frac{|c_k|^2}{\lambda_k^2}\left[\Big(\lambda_k N^\nu\big(h(\lambda_k) - 1\big) - 2\zeta(\nu)\Big)^2 + 2\zeta(2\nu)\right]
\end{aligned}
\tag{133}
$$

While the above expression for $L[h]$ is already a valid asymptotic for the loss in the saturated phase, we make a few further refinements.

First, note that (133) relies on condition (125) for the algorithm $h(\lambda)$. This condition might be quite difficult to verify in practice, as it requires the knowledge of the optimal algorithm $h(\lambda)$. However, perturbative expansion (132) shows that optimal algorithm satisfies $|h^*(\lambda) - 1|^2 = O(\lambda^{-2}N^{-2\nu})$. Since in saturated phase $\frac{\kappa}{\nu} > 2$, we can always make $\lambda^{-2}N^{-2\nu}$ smaller than $\lambda^{-\frac{\kappa}{\nu}+\varepsilon}N^{-2\nu}$ by choosing $\varepsilon < \frac{\kappa}{\nu} - 2$. Thus, employing triangular inequality shows that the original condition (125) is equivalent to

$$|h(\lambda) - 1|^2 = O(\lambda^{-\frac{\kappa}{\nu}+\varepsilon}N^{-2\nu}), \quad 0 < \varepsilon < \frac{\kappa}{\nu} - 2. \tag{134}$$

The main advantage of condition (134) compared to (126) is that it is easily verifiable. For instance, KRR has $1 - h_\eta(\lambda) = \frac{\eta}{\lambda+\eta}$ which satisfies (134) for $|\eta| = O(N^{-\nu})$. Similarly, for gradient flow we have $1 - h_t(\lambda) = e^{-t\lambda}$ which satisfies (134) with $t = \Omega(N^{\frac{2\nu^2}{\kappa}} + \varepsilon')$ and some $\varepsilon' > 0$ depending on $\varepsilon$.

Note that KRR and GF examples of $h(\lambda)$ considered above satisfy (134) not only on the scales $s \in [0, \nu]$ but for all $s \geq 0$, or equivalently $\lambda \leq 1$. Based on this observation, let us impose condition (134) on all $\lambda \leq 1$. Then, the summation indices in (133) to all population values $l \in \mathbb{Z}$, similarly to how it was done for optimal algorithm loss in (130).

To summarize, through the derivations and discussions above we have proved

**Theorem 3.** *Consider the Circle model in saturated phase $\kappa > 2\nu$. Then, if an algorithm $h(\lambda)$ satisfies (134) for all $\lambda \leq 1$, the loss functional is given by*

$$L[h] = \frac{1}{2}N^{-2\nu}(1 + o(1))\sum_{l=-\infty}^{\infty} \frac{|c_l|^2}{\lambda_l^2}\left[\Big(\lambda_l N^\nu\big(h(\lambda_l) - 1\big) - 2\zeta(\nu)\Big)^2 + 2\zeta(2\nu)\right]. \tag{135}$$

Finally, let us comment on the KRR result (22) presented in the main paper. Observe that the combination $\lambda(h(\lambda) - 1)$ entering the loss functional (135) is given, in case of KRR, by

$$\lambda(h_\eta(\lambda) - 1) = -\frac{\eta}{1 + \frac{\eta}{\lambda}} = -\eta\Big(1 + O(\frac{\eta}{\lambda})\Big). \tag{136}$$

Substitution of the above into loss functional (135) leads to the KRR expression (22). According to condition (134), this expression is applicable only for small enough regularization strengths $|\eta| = O(N^{-\nu})$.

### D.3.4 NON-SATURATED PHASE $\kappa < 2\nu$

In this section, we obtain a limiting form of loss functional in the non-saturated phase. While this form was not discussed explicitly in the main text, it was used to produce the first two plots in Figure 3.

An important feature of the non-saturated phase is the localization of the loss on the highest spectral scale $s = \nu$, which is ensured for algorithms $h(\lambda)$ satisfying (125). Such localization corresponds to values $\tau \sim 1$, so we can not rely on perturbative expansion in small parameter $\tau \to 0$, as we did in the sections above. Instead, for all terms of the loss functional (116) we need to take into account their limiting $N \to \infty$ form at fixed $\tau$. For N-deformations, such limit leads to symmetrized Hurwitz zeta function $\zeta_\tau^{(\alpha)} = \zeta(\alpha, \tau) + \zeta(\alpha, 1 - \tau)$ introduced in equation (23) of the main text. Indeed, slightly rearranging (127), we get

$$\big[\lambda_k^a |c_k|^{2b}\big]_N = N^{-\alpha}\Big(\zeta\big(\alpha, \tau\big) + \zeta\big(\alpha, 1 - \tau + \frac{2}{N}\big)\Big) \xrightarrow[\tau=const]{N\to\infty,} N^{-\alpha}\zeta_\tau^{(\alpha)}. \tag{137}$$

Next, observe that on the scale $s = \nu$ the density of eigenvalues $\lambda_k$ is very high, in a sense that $\frac{|\lambda_{k+1} - \lambda_k|}{\lambda_k} = O(\tau^{-1}N^{-1}) \xrightarrow[\tau=const]{N\to\infty,} 0$. Thus, in the limit $N \to \infty$ the summation over spectral index $k$ in the loss functional (116) translates into an integral: $\sum_{k=-\frac{N}{2}}^{\frac{N}{2}} \to \int_{-\frac{N}{2}}^{\frac{N}{2}} dk \to 2N^{-1}\int_0^{\frac{1}{2}} d\tau$, where for the last transition recall that $\tau = \frac{|k|+1}{N}$ and N-deformations $[\lambda_k^a|c_k|^{2b}]_N$ are even functions of $k$. This leads to the following continuous form of the loss functional (116)

$$
\begin{aligned}
L^{(\mathrm{cont})}[h] &= N^{-\kappa} \int_0^{\frac{1}{2}} \left[ \frac{\zeta_\tau^{(\kappa+1)}\zeta_\tau^{(2\nu)}}{\left(\zeta_\tau^{(\nu)}\right)^2} \Big(h(\widehat{\lambda}_\tau) - h^*(\widehat{\lambda}_\tau)\Big)^2 + \zeta_\tau^{(\kappa+1)} - \frac{\left(\zeta_\tau^{(\nu+\kappa+1)}\right)^2}{\zeta_\tau^{(\kappa+1)}\zeta_\tau^{(2\nu)}} \right] d\tau \\
&= N^{-\kappa} \int_0^{\frac{1}{2}} \left[ \frac{\zeta_\tau^{(\kappa+1)}\zeta_\tau^{(2\nu)}}{\left(\zeta_\tau^{(\nu)}\right)^2} h^2(\widehat{\lambda}_\tau) - 2\frac{\zeta_\tau^{(\nu+\kappa+1)}}{\zeta_\tau^{(\nu)}} h(\widehat{\lambda}_\tau) + \zeta_\tau^{(\kappa+1)} \right] d\tau,
\end{aligned}
\tag{138}
$$

where $\widehat{\lambda}_\tau = N^{-\nu}\zeta_\tau^{(\nu)}$ and the optimal algorithm $h^*(\widehat{\lambda}_\tau)$ is given by (23).

### D.3.5 THE OVERLEARNING TRANSITION POINT

Observe that if $\kappa = \nu - 1$, then in the case of zero noise $\sigma^2 = 0$ the optimal algorithm (12) becomes $h^*(\widehat{\lambda}_k) \equiv 1$, and the same holds for the limiting $(N \to \infty)$ version (23). We prove now that for larger (smaller) $\kappa$ the optimal algorithm becomes overlearning (underlearning). We give the proof for the limiting $(N \to \infty)$ version, but it is easy to see that the proof extends without change to the original discrete version as well.

**Lemma 2.** *Consider the $N \to \infty$ limit of the optimal algorithm given by* (23):

$$h^*(\widehat{\lambda}_\tau) = \frac{\zeta_\tau^{(\nu+\kappa+1)}\zeta_\tau^{(\nu)}}{\zeta_\tau^{(\kappa+1)}\zeta_\tau^{(2\nu)}}, \qquad \zeta_x^{(\alpha)} \equiv \zeta(\alpha, x) + \zeta(\alpha, 1 - x). \tag{139}$$

*Then for any $\tau \in (0, 1)$*

$$h^*(\widehat{\lambda}_\tau) \begin{cases} < 1, & \kappa + 1 < \nu, \\ = 1, & \kappa + 1 = \nu, \\ > 1, & \kappa + 1 > \nu. \end{cases} \tag{140}$$

*Proof.* We have

$$\ln h^*(\widehat{\lambda}_\tau) = (\ln \zeta_\tau^{(\nu+\kappa+1)} - \ln \zeta_\tau^{(\kappa+1)}) - (\ln \zeta_\tau^{(2\nu)} - \ln \zeta_\tau^{(\nu)}) = \int_\nu^{\kappa+1} d\alpha \int_\alpha^{\alpha+\nu} d\beta \; \tfrac{d^2}{d\beta^2} \ln \zeta_\tau^{(\beta)}.$$
(141)

To prove the lemma, it suffices to show that the function $f(\alpha) = \zeta_\tau^{(\alpha)}$ is strictly log-convex for $\alpha > 1$, i.e. $f f'' > (f')^2$. Note that

$$\frac{d^m}{d\alpha^m} f(\alpha) = \sum_{n=0}^\infty (-\ln(n+\tau))^m (n+\tau)^{-\alpha} + \sum_{n=0}^\infty (\ln(n+1-\tau))^m (n+1-\tau)^{-\alpha}.$$
(142)

Then, the inequality $f f'' > (f')^2$ is just the Cauchy inequality for the vectors

$$(\ldots, (n+\tau)^{-\alpha/2}, \ldots, (n+1-\tau)^{-\alpha/2}, \ldots)$$
(143)

and

$$(\ldots, -\ln(n+\tau)(n+\tau)^{-\alpha/2}, \ldots, \ln(n+1-\tau)(n+1-\tau)^{-\alpha/2}, \ldots).$$
(144)

The inequality is strict because the vectors are not collinear. $\square$

The over/under-learning property $h^*(\lambda_\tau) \gtrless 1$ is clearly visible in the right two subfigures of Figure 3. At $\kappa > \nu - 1$, the optimal regularized KRR is also overlearning (with a negative regularization). The optimally stopped GF in this regime is GF continued to infinity, i.e. $h(\lambda) = \lim_{t\to\infty}(1 - e^{-t\lambda}) = 1$.

## E  WISHART MODEL

As for the circle model, in this section we collect all our derivations for the Wishart model.

Our strategy in computing loss functional (8) relies on the resolvent $\widehat{\mathbf{R}}(z)$ of empirical kernel matrix $\mathbf{K} = \boldsymbol{\Phi}^T \boldsymbol{\Lambda} \boldsymbol{\Phi}$:

$$\widehat{\mathbf{R}}(z) = \left( \frac{1}{N} \boldsymbol{\Phi}^T \boldsymbol{\Lambda} \boldsymbol{\Phi} - z\mathbf{I} \right)^{-1} = \left( \frac{1}{N} \sum_l \lambda_l \phi_l \phi_l^T - z\mathbf{I} \right)^{-1}, \qquad z \in \mathbb{C},$$
(145)

where $\phi_l$ are the columns of feature matrix $\boldsymbol{\Phi}$.

We note that the projections on empirical eigenvalues and eigenvectors, appearing in empirical transfer measure (41) are related to the resolvent via the limit in the complex plane

$$\sum_{k=1}^N \delta_{\widehat{\lambda}_k}(d\lambda)\mathbf{u}_k\mathbf{u}_k^T = \frac{1}{\pi} \lim_{y\to 0_+} \Im\left( \sum_k \widehat{\lambda}_k \mathbf{u}_k\mathbf{u}_k^T - (\lambda + \mathbf{i}y)\mathbf{I} \right)^{-1} d\lambda$$
$$= \frac{1}{\pi} \lim_{y\to 0_+} \Im\widehat{\mathbf{R}}(\lambda + \mathbf{i}y)d\lambda,$$
(146)

where $\Im z$ denotes the imaginary part of $z \in \mathbb{C}$.

Now, denote the resolvent projected on features $\phi_l$ as

$$\widehat{R}_{ll'}(z) = \frac{1}{N} \phi_l^T \widehat{\mathbf{R}}(z) \phi_{l'},$$
(147)

and the first two moments of the latter

$$R_{ll'}(z) = \mathbb{E}_{\boldsymbol{\Phi}}\left[ \widehat{R}_{ll'}(z) \right],$$
(148)

$$R_{l_1 l_1' l_2' l_2}(z_1, z_2) = \mathbb{E}_{\boldsymbol{\Phi}}\left[ \widehat{R}_{l_1 l_1'}(z_1) \widehat{R}_{l_2' l_2}(z_2) \right].$$
(149)

Then, substituting (146) into the transfer measure (41) immediately connects it to the projected resolvent

$$\widehat{\rho}_{ll'}^{(f)}(d\lambda) = \frac{\lambda_l}{\pi\lambda}\lim_{y\to 0_+}\Im\left\{\widehat{R}_{ll'}(\lambda+\mathbf{i}y)\right\}d\lambda. \tag{150}$$

Carrying the relation (150) over to the first and second moments of the transfer measure gives

$$\rho_{ll'}^{(1)}(d\lambda) = \frac{\lambda_l}{\pi\lambda}\lim_{y\to 0_+}R_{ll'}^{(\text{im})}(\lambda+\mathbf{i}y)d\lambda, \tag{151}$$

$$\rho_{l_1 l_1' l_2' l_2}^{(2)}(d\lambda_1, d\lambda_2) = \frac{\lambda_{l_1'}\lambda_{l_2'}}{\pi^2\lambda_1\lambda_2}\lim_{\substack{y_1\to 0_+ \\ y_2\to 0_+}}R_{l_1 l_1' l_2' l_2}^{(\text{im})}(\lambda_1+\mathbf{i}y_1, \lambda_2+\mathbf{i}y_2)d\lambda_1 d\lambda_2, \tag{152}$$

$$\rho_l^{(\varepsilon)}(d\lambda) = \frac{\lambda_l^2}{\pi\lambda^2}\lim_{y\to 0_+}R_{ll}^{(\text{im})}(\lambda+\mathbf{i}y)d\lambda, \tag{153}$$

where $R_{ll'}^{(\text{im}))}$ and $R_{l_1 l_1' l_2' l_2}^{(\text{im})}$ are the moments of the imaginary part of the resolvent projections

$$R_{ll'}^{(\text{im})}(z) = \mathbb{E}_{\mathcal{D}_N}\left[\Im\left\{\widehat{R}_{ll'}(z)\right\}\right], \tag{154}$$

$$R_{l_1 l_1' l_2' l_2}^{(\text{im})}(z_1, z_2) = \mathbb{E}_{\mathcal{D}_N}\left[\Im\left\{\widehat{R}_{l_1 l_1'}(z_1)\right\}\Im\left\{\widehat{R}_{l_2' l_2}(z_2)\right\}\right]. \tag{155}$$

Then, the learning measures defining the loss functional (8) are obtained by summation over population indices as in (44).

## E.1 RESOLVENT

Given the connection between resolvent and learning measures $\rho^{(2)}, \rho^{(1)}, \rho^{(\varepsilon)}$ described above, we see that the resolvent moments (148) and (149) are fundamental building blocks for the loss functional (8). In this section, we try to calculate these moments and simplify the result as much as possible.

To start, recall that the $\frac{1}{N}\text{Tr}[\widehat{\mathbf{R}}(z)]$ gives the Stieltjes transform of the empirical spectral measure $\widehat{\mu} = \frac{1}{N}\sum_{k=1}^N\delta_{\widehat{\lambda}_k}$. A central quantity in our calculations will be Stieltjes transform of the average spectral measure $\mu = \mathbb{E}[\widehat{\mu}]$

$$r(z) \equiv \int_{-\infty}^{\infty}\frac{\mu(dt)}{z-t} = \frac{1}{N}\mathbb{E}\left[\text{Tr}[\widehat{\mathbf{R}}(z)]\right]. \tag{156}$$

Next, denote resolvents of kernel matrices with one or two spectral components removed as

$$\widehat{\mathbf{R}}_{-l}(z) = \left(\widehat{\mathbf{R}}^{-1}(z) - \frac{1}{N}\lambda_l\phi_l\phi_l^T\right)^{-1} = \left(\frac{1}{N}\sum_{m\neq l}\lambda_m\phi_m\phi_m^T - z\mathbf{I}\right)^{-1}, \tag{157}$$

$$\widehat{\mathbf{R}}_{-l-l'}(z) = \left(\widehat{\mathbf{R}}_{-l}^{-1}(z) - \frac{1}{N}\lambda_{l'}\phi_{l'}\phi_{l'}^T\right)^{-1} = \left(\frac{1}{N}\sum_{m\neq l,l'}\lambda_m\phi_m\phi_m^T - z\mathbf{I}\right)^{-1}, \tag{158}$$

and the respective Stieltjes transforms as $r_{-l}(z),\ r_{-l-l'}(z)$.

In our calculations, we will use two assumptions

**Assumption 1.** *(Self-averaging property)*. *For a random Gaussian vector $\phi \sim \mathcal{N}(0, \mathbf{I})$ independent from $\mathbf{\Phi}$*

$$\frac{1}{N}\phi^T\widehat{\mathbf{R}}(z)\phi \approx \frac{1}{N}\mathbb{E}\left[\phi^T\widehat{\mathbf{R}}(z)\phi\right] = r(z).$$

**Assumption 2.** *(Stability under eigenvalue removal)*

$$r_{-l-l'}(z) \approx r_{-l}(z) \approx r(z).$$

In classical RMT settings, e.g. that of Marchenko-Pastur law, both of these assumptions can be rigorously shown. Specifically, both the statistical fluctuations of $\frac{1}{N}\phi^T\widehat{\mathbf{R}}(z)\phi$ and the change of

$r(z)$ after eigenvalue removal give no contribution to the limiting spectral measure as $N \to \infty$. We leave for the future work the validity of these assumptions in our settings.

As a final preparation step, let us write down a quick way of deriving the self-consistent (fixed-point) equation for $r(z)$ using the above assumptions. Starting with the trivial relation $1 = \frac{1}{N} \operatorname{Tr}[\mathbf{I}]$, we get

$$1 = \frac{1}{N} \mathbb{E} \left[ \operatorname{Tr} \left[ \widehat{\mathbf{R}}(z) \left( \frac{1}{N} \sum_l \lambda_l \boldsymbol{\phi}_l \boldsymbol{\phi}_l^T - z\mathbf{I} \right) \right] \right] = \frac{1}{N} \sum_l \lambda_l \mathbb{E} \left[ \frac{1}{N} \boldsymbol{\phi}_l^T \widehat{\mathbf{R}}(z) \boldsymbol{\phi}_l \right] - zr(z). \quad (159)$$

Next, we relate $\boldsymbol{\phi}_l^T \widehat{\mathbf{R}}(z) \boldsymbol{\phi}_l$ to $\boldsymbol{\phi}_l^T \widehat{\mathbf{R}}_{-l}(z) \boldsymbol{\phi}_l$ via Sherman-Morrison formula in order to break the dependence between $\boldsymbol{\phi}_l$ and the matrix inside.

$$\frac{1}{N} \boldsymbol{\phi}_l^T \widehat{\mathbf{R}}(z) \boldsymbol{\phi}_l = \frac{1}{N} \boldsymbol{\phi}_l^T \widehat{\mathbf{R}}_{-l}(z) \boldsymbol{\phi}_l - \frac{\lambda_l \left( \frac{1}{N} \boldsymbol{\phi}_l^T \widehat{\mathbf{R}}_{-l}(z) \boldsymbol{\phi}_l \right)^2}{1 + \lambda_l \frac{1}{N} \boldsymbol{\phi}_l^T \widehat{\mathbf{R}}_{-l}(z) \boldsymbol{\phi}_l} = \frac{\frac{1}{N} \boldsymbol{\phi}_l^T \widehat{\mathbf{R}}_{-l}(z) \boldsymbol{\phi}_l}{1 + \lambda_l \frac{1}{N} \boldsymbol{\phi}_l^T \widehat{\mathbf{R}}_{-l}(z) \boldsymbol{\phi}_l} \quad (160)$$

$$\overset{(1)}{=} \frac{r_{-l}(z)}{1 + \lambda_l r_{-l}(z)} \overset{(2)}{=} \frac{r(z)}{1 + \lambda_l r(z)},$$

where in $(1)$ and $(2)$ we used the assumptions 1 and 2 respectively. Substituting this into (159) gives the self-consistent equation

$$1 = -zr(z) + \frac{1}{N} \sum_l \frac{r(z)\lambda_l}{1 + r(z)\lambda_l}, \quad (161)$$

which is often written in a fixed-point form as

$$r(z) = 1 \Big/ \left( -z + \frac{1}{N} \sum_l \frac{\lambda_l}{1 + r(z)\lambda_l} \right). \quad (162)$$

For $z = -\eta < 0$, we have $r(z) = \eta_{\text{eff},N}^{-1}$ for effective regularization $\eta_{\text{eff},N}$ defined in (1).

### E.1.1 COMPUTING THE RESOLVENT MOMENTS

**Reflection symmetry.** Here, we mainly repeat Simon et al. (2023) to establish useful exact relations for resolvent expectations. Recall that distribution of a Gaussian random vector $\mathbf{z} \sim \mathcal{N}(0, \mathbf{I})$ remains invariant under orthogonal transformations: $\mathbf{Uz} \sim \mathcal{N}(0, \mathbf{I})$, where $\mathbf{U}$ is an arbitrary orthogonal matrix. Now, we notice that our averages of interest (154) and (155) have a form $\mathbb{E}_{\boldsymbol{\Phi}} \big[ f(\mathbf{K}, \boldsymbol{\Phi}) \big]$ for some function $f(\cdot, \cdot)$ of empirical kernel matrix $\mathbf{K} = \boldsymbol{\Phi} \boldsymbol{\Lambda} \boldsymbol{\Phi}^T$ and "features" $\boldsymbol{\Phi}$. Applying transformation $\boldsymbol{\Phi} \to \mathbf{U}\boldsymbol{\Phi}$ to perturbed kernel matrix $\boldsymbol{\Phi}^T \mathbf{U} \boldsymbol{\Lambda} \mathbf{U}^T \boldsymbol{\Phi}$ gives

$$\mathbb{E} \left[ f(\boldsymbol{\Phi}^T \mathbf{U} \boldsymbol{\Lambda} \mathbf{U}^T \boldsymbol{\Phi}, \boldsymbol{\Phi}) \right] = \mathbb{E} \left[ f(\boldsymbol{\Phi}^T \boldsymbol{\Lambda} \boldsymbol{\Phi}, \mathbf{U}\boldsymbol{\Phi}) \right]. \quad (163)$$

If we take $\mathbf{U}$ to be a reflection along one of the basis axes, e.g. $(\mathbf{U}^{(m)})_{ll'} = \delta_{ll'}(1 - 2\delta_{lm})$ for reflection along axis $m$, then $\mathbf{U}^{(m)} \boldsymbol{\Lambda} (\mathbf{U}^{(m)})^T = \boldsymbol{\Lambda}$. This implies

$$\mathbb{E} \left[ f(\boldsymbol{\Phi}^T \boldsymbol{\Lambda} \boldsymbol{\Phi}, \boldsymbol{\Phi}) \right] = \mathbb{E} \left[ f(\boldsymbol{\Phi}^T \boldsymbol{\Lambda} \boldsymbol{\Phi}, \mathbf{U}^{(m)} \boldsymbol{\Phi}) \right]. \quad (164)$$

Applying (164) to (154) and (155) gives their non-zero elements are only those with paired indices: $R_{ll}(z)$ for the first moment and $R_{ll'l'}(z_1, z_2)$, $R_{ll'll'}(z_1, z_2)$, $R_{ll'l'l}(z_1, z_2)$ for the second moment. Moreover, note that we are interested only in those components of the resolvent second moment that contribute to the loss (44). This leaves $R_{ll'l'l}(z_1, z_2)$ to be the only relevant components.

**First moment.** Here, all the necessary computations were already done in (160), leading to

$$R_{ll}(z) = \frac{r(z)}{1 + \lambda_l r(z)}. \quad (165)$$

**Second moment at $z_1 = z_2$.** In this case, the second moment is connected to the derivative of the first moment, which was utilized by Simon et al. (2023) and Bordelon et al. (2020) to obtain the generalization error of KRR. Indeed,

$$
\begin{aligned}
R_{ll'l'l}(z, z) &= \mathbb{E}\left[\frac{1}{N}\phi_l^T \widehat{\mathbf{R}}(z)\frac{\phi_{l'}\phi_{l'}^T}{N}\widehat{\mathbf{R}}(z)\phi_l\right] \\
&= -\mathbb{E}\left[\frac{1}{N}\phi_l^T\frac{\partial\left(\sum_m \lambda_m\frac{\phi_m\phi_m^T}{N} - z\mathbf{I}\right)^{-1}}{\partial\lambda_{l'}}\phi_l\right] = -\partial_{\lambda_{l'}}R_{ll}(z).
\end{aligned}
\tag{166}
$$

As prerequisite for computing the derivative $\partial_{\lambda_{l'}}R_{ll}(z)$, we compute similar derivative of inverse Stieltjes transform $r^{-1}(z)$. Differentiating the fixed point equation (161) in the form $r^{-1} + z = \frac{1}{N}\sum_l \frac{\lambda_l r^{-1}}{\lambda_l + r^{-1}}$ w.r.t. to either $z$ or $\lambda_{l'}$ gives

$$
\frac{\partial r^{-1}}{\partial z}\left[1 - \frac{1}{N}\sum_l \frac{\lambda_l}{\lambda_l + r^{-1}} + \frac{1}{N}\sum_l \frac{\lambda_l r^{-1}}{(\lambda_l + r^{-1})^2}\right] = -1,
\tag{167}
$$

$$
\frac{\partial r^{-1}}{\partial \lambda_{l'}}\left[1 - \frac{1}{N}\sum_l \frac{\lambda_l}{\lambda_l + r^{-1}} + \frac{1}{N}\sum_l \frac{\lambda_l r^{-1}}{(\lambda_l + r^{-1})^2}\right] = \frac{1}{N}\frac{r^{-2}}{(\lambda_{l'} + r^{-1})^2},
\tag{168}
$$

leading to

$$
\frac{\partial r^{-1}}{\partial \lambda_{l'}} = -\frac{\partial r^{-1}}{\partial z}\frac{1}{N}\frac{r^{-2}}{(\lambda_{l'} + r^{-1})^2},
\tag{169}
$$

$$
\frac{\partial r^{-1}}{\partial z} = \frac{r^{-1}}{z - \frac{1}{N}\sum_l \frac{\lambda_l r^{-2}}{(\lambda_l + r^{-1})^2}}.
\tag{170}
$$

Using this, we second moment becomes

$$
R_{ll'l'l}(z, z) = -\partial_{\lambda_{l'}}R_{ll}(z) = \frac{\delta_{ll'}}{(\lambda_l + r^{-1})^2} - \frac{1}{N}\frac{r^{-2}\partial_z r^{-1}}{(\lambda_l + r^{-1})^2(\lambda_{l'} + r^{-1})^2}.
\tag{171}
$$

**Second moment at $z_1 \neq z_2$ with $l = l'$.** Here and in the next case $l \neq l'$ we will again apply Sherman-Morrison formula, including two subsequent applications to break dependence with $\phi_l, \phi_{l'}$. Denoting $\widehat{\mathbf{R}}_\#(z)$ the resolvent with, possibly, removed eigenvalue, removing an extra eigenvalue can be written in a simplified form under assumptions 1 and 2

$$
\widehat{\mathbf{R}}_\#(z) = \widehat{\mathbf{R}}_{\#-l}(z) - \frac{\lambda_l}{1 + \lambda_l r(z)}\widehat{\mathbf{R}}_{\#-l}(z)\frac{\phi_l\phi_l^T}{N}\widehat{\mathbf{R}}_{\#-l}(z) = q\left(\widehat{\mathbf{R}}_{\#-l}(z), \frac{\phi_l\phi_l^T}{N}, a_l(z)\right),
\tag{172}
$$

where $q(\mathbf{X}, \mathbf{Y}, a) = \mathbf{X} - a\mathbf{XYX}$ is a polynomial in two matrices $\mathbf{X}, \mathbf{Y}$ and a scalar variable $a$, and $a_l(z)$ is a shorthand notation for $\lambda_l/(1 + \lambda_l r(z))$.

Using representation (172) we can write our second moment element as

$$
R_{llll}(z_1, z_2) = \mathbb{E}\left[\text{Tr}\left[\mathbf{Y}q(\mathbf{X}_1, \mathbf{Y}, a_1)\mathbf{Y}q(\mathbf{X}_2, \mathbf{Y}, a_2)\right]\right],
\tag{173}
$$

where we have denoted

$$
\mathbf{X}_1 = \widehat{\mathbf{R}}_{-l}(z_1), \quad \mathbf{X}_2 = \widehat{\mathbf{R}}_{-l}(z_2), \quad \mathbf{Y} = \frac{\phi_l\phi_l^T}{N}, \quad a_1 = a_l(z_1), \quad a_2 = a_l(z_2).
\tag{174}
$$

Now we exploit the independence between $\mathbf{X}_1, \mathbf{X}_2$ and $\mathbf{Y}$ by first taking the expectations w.r.t. $\mathbf{Y}$. Since $\mathbf{Y}$ is product of two Gaussian random variables and (173) is polynomial in $\mathbf{Y}$ containing monomials of degree from 2 to 4, we need to compute Gaussian moments of the order from 4 to 8. This can be conveniently done using Wick's theorem for computing moments of Gaussian random variables, which equates $2m$-th moment to the sum over all pairings of $2m$ variables of the products of $m$ intra-pair covariances. Specifically, for normal vector $\mathbf{x} \sim \mathcal{N}(0, \mathbf{I})$, it reduces to the products of Dirac deltas

$$
\mathbb{E}\left[x_{i_1}x_{i_2}\ldots x_{i_{2m-1}}x_{i_{2m}}\right] = \frac{1}{2^m}\sum_{\sigma\in S_{2m}}\prod_{j=1}^m \mathbb{E}\left[x_{i_{\sigma(2j-1)}}x_{i_{\sigma(2j)}}\right] = \frac{1}{2^m}\sum_{\sigma\in S_{2m}}\prod_{j=1}^m \delta_{i_{\sigma(2j-1)}i_{\sigma(2j)}},
\tag{175}
$$

where $S_{2m}$ denotes the group of permutations of the index set $\{1, 2, \ldots, 2m-1, 2m\}$. Now we apply Wick's theorem separately to each monomial from (173). For the simplest order-two monomial, we have

$$
\begin{aligned}
\mathbb{E}_{\mathbf{Y}}\left[\mathrm{Tr}[\mathbf{Y}\mathbf{X}_1\mathbf{Y}\mathbf{X}_2]\right] &= N^{-2} \sum_{i_+ i_- j_+ j_-} \mathbb{E}_{\boldsymbol{\phi} \sim \mathcal{N}(0,\mathbf{I})}\left[\phi_{i_+}\phi_{i_-}(\mathbf{X}_1)_{i_- j_+}\phi_{j_+}\phi_{j_-}(\mathbf{X}_2)_{j_- i_+}\right] \\
&= N^{-2}\left(Tr[\mathbf{X}_1]\,\mathrm{Tr}[\mathbf{X}_2] + 2\,\mathrm{Tr}[\mathbf{X}_1\mathbf{X}_2]\right) \\
&= N^{-2}\left(Tr[\mathbf{X}_1]\,\mathrm{Tr}[\mathbf{X}_2] + 2\frac{Tr[\mathbf{X}_1] - Tr[\mathbf{X}_2]}{z_1 - z_2}\right) \\
&= r(z_1)r(z_2) + \frac{2}{N}\frac{r(z_1) - r(z_2)}{z_1 - z_2} = r_1 r_2 + O(N^{-1}),
\end{aligned}
\tag{176}
$$

where $r_1, r_2$ is a shorthand for $r(z_1), r(z_2)$. Here we first applied Wick's theorem, then transformed the product $\mathbf{X}_1\mathbf{X}_2 = (\mathbf{X}_1 - \mathbf{X}_2)/(z_1 - z_2)$. Then, we wrote the resolvent traces in terms of Stieltjes transform $\mathrm{Tr}[\mathbf{X}_{1,2}] = r(z_{1,2})$ using assumption (1), thus removing the need to take expectation with respect $\mathbf{X}_1, \mathbf{X}_2$ in the remaining calculation of the second moment $R_{llll}(z_1, z_2)$. An important observation is that when computing the moment of order $n$, the leading term in $\frac{1}{N}$ has to be the product of traces $Tr[\mathbf{X}_1], Tr[\mathbf{X}_2]$, while all the other terms (containing at least one trace of a product) will give $O(N^{-1})$ contribution. Using the above observation, we can easily compute leading terms for all the other averages:

$$
\begin{aligned}
\mathbb{E}_{\mathbf{Y}}\left[\mathrm{Tr}[\mathbf{Y}\mathbf{X}_1\mathbf{Y}\mathbf{X}_1\mathbf{Y}\mathbf{X}_2]\right] &= N^{-3}\,\mathrm{Tr}[\mathbf{X}_1]^2\,\mathrm{Tr}[\mathbf{X}_2] + O(N^{-1}) = r_1^2 r_2 + O(N^{-1}), \\
\mathbb{E}_{\mathbf{Y}}\left[\mathrm{Tr}[\mathbf{Y}\mathbf{X}_1\mathbf{Y}\mathbf{X}_2\mathbf{Y}\mathbf{X}_2]\right] &= N^{-3}\,\mathrm{Tr}[\mathbf{X}_1]\,\mathrm{Tr}[\mathbf{X}_2]^2 + O(N^{-1}) = r_1 r_2^2 + O(N^{-1}), \\
\mathbb{E}_{\mathbf{Y}}[\mathrm{Tr}[\mathbf{Y}\mathbf{X}_1\mathbf{Y}\mathbf{X}_1\mathbf{Y}\mathbf{X}_2\mathbf{Y}\mathbf{X}_2]] &= N^{-4}\,\mathrm{Tr}[\mathbf{X}_1]^2\,\mathrm{Tr}[\mathbf{X}_2]^2 + O(N^{-1}) = r_1^2 r_2^2 + O(N^{-1}).
\end{aligned}
\tag{177}
$$

Combining all the monomials, we can summarize the whole computation as follows

$$
R_{llll}(z_1, z_2) = q(r_1, 1, a_1)q(r_2, 1, a_2) + O(N^{-1}) = \frac{1}{(\lambda_l + r_1^{-1})(\lambda_l + r_2^{-1})} + O(N^{-1}), \tag{178}
$$

where in the last equality we used $q(r(z), 1, a_l(z)) = \frac{1}{\lambda_l + r^{-1}(z)}$. Note that in the limit $z_1 \to z_2$, the expressions above coincide with previously derived (171) up to subleading $O(N^{-1})$ terms.

**Second moment at $z_1 \neq z_2$ with $l \neq l'$.** Here we need to remove two eigenvalues $\lambda_l$ and $\lambda_{l'}$. The respective resolvent expression is

$$
\begin{aligned}
\widehat{\mathbf{R}}(z) &= q(\widehat{\mathbf{R}}_{-l}(z), \tfrac{\phi_l \phi_l^T}{N}, a_l(z)) = q(q(\widehat{\mathbf{R}}_{-l-l'}(z), \tfrac{\phi_{l'}\phi_{l'}^T}{N}, a_{l'}(z)), \tfrac{\phi_l\phi_l^T}{N}, a_l(z))) \\
&= \widetilde{q}(\widehat{\mathbf{R}}_{-l-l'}(z), \tfrac{\phi_l\phi_l^T}{N}, \tfrac{\phi_{l'}\phi_{l'}^T}{N}, a_l(z), a_{l'}(z)),
\end{aligned}
\tag{179}
$$

where

$$
\begin{aligned}
\widetilde{q}(\mathbf{X}, \mathbf{Y}, \mathbf{Y}', a, a') &= q(q(\mathbf{X}, \mathbf{Y}', a'), \mathbf{Y}, a) \\
&= \mathbf{X} - a\mathbf{X}\mathbf{Y}\mathbf{X} - a'\mathbf{X}\mathbf{Y}'\mathbf{X} + aa'\left(\mathbf{X}\mathbf{Y}\mathbf{X}\mathbf{Y}'\mathbf{X} + \mathbf{X}\mathbf{Y}'\mathbf{X}\mathbf{Y}\mathbf{X}\right).
\end{aligned}
\tag{180}
$$

Substituting (179) into (149) will produce an expression of the form

$$
R_{ll'l'l}(z_1, z_2) = \mathbb{E}\left[\mathrm{Tr}\left[\mathbf{Y}'\widetilde{q}(\mathbf{X}_1, \mathbf{Y}, \mathbf{Y}', a_1, a_1')\mathbf{Y}\widetilde{q}(\mathbf{X}_2, \mathbf{Y}, \mathbf{Y}', a_2, a_2')\right]\right], \tag{181}
$$

where in addition to (174) we have denoted $\mathbf{Y}' = \frac{\phi_{l'}\phi_{l'}^T}{N}$, $a_1' = a_{l'}(z_1)$, and $a_2' = a_{l'}(z_2)$.

Now, let us again calculate the expectations over $\mathbf{Y}, \mathbf{Y}'$ using Wick's theorem. The difference with the $l = l'$ case is that the pairings between $\phi_l$ and $\phi_{l'}$ produce zeros due to their independence. For example, the expectation for the simplest monomial is

$$
\begin{aligned}
\mathbb{E}_{\mathbf{Y},\mathbf{Y}'}\left[\mathrm{Tr}[\mathbf{Y}'\mathbf{X}_1\mathbf{Y}\mathbf{X}_2]\right] &= \frac{1}{N^2}\sum_{i_+ i_- j_+ j_-}\mathbb{E}_{\boldsymbol{\phi},\boldsymbol{\phi}' \sim \mathcal{N}(0,\mathbf{I})}\left[\phi'_{i_+}\phi'_{i_-}(\mathbf{X}_1)_{i_- j_+}\phi_{j_+}\phi_{j_-}(\mathbf{X}_2)_{j_- i_+}\right] \\
&= N^{-2}\,\mathrm{Tr}[\mathbf{X}_1\mathbf{X}_2] = -\frac{1}{N}\frac{r(z_1) - r(z_2)}{z_1 - z_2}.
\end{aligned}
\tag{182}
$$

Observe that the monomial above has $\frac{1}{N}$ magnitude. This is the consequence that $N^0$ terms from the $l = l'$ case arose from pairings between $\mathbf{Y}'$ and $\mathbf{Y}$, which are zero in the $l \neq l'$ case. Taking into account the symmetry $R_{ll'll'}(z_1, z_2) = R_{ll'll'}(z_2, z_1)$, we proceed similarly and calculate the leading terms for all the other independent monomials.

Terms with $\mathbf{X}_2$:

$$\mathbb{E}_{\mathbf{Y},\mathbf{Y}'}\left[\mathrm{Tr}[\mathbf{Y}'\mathbf{X}_1\mathbf{Y}'\mathbf{X}_1\mathbf{Y}\mathbf{X}_2]\right] = N^{-3}\,\mathrm{Tr}[\mathbf{X}_1\mathbf{X}_2]\,\mathrm{Tr}[\mathbf{X}_1] + O(N^{-2}),$$
$$\mathbb{E}_{\mathbf{Y},\mathbf{Y}'}\left[\mathrm{Tr}[\mathbf{Y}'\mathbf{X}_1\mathbf{Y}\mathbf{X}_1\mathbf{Y}\mathbf{X}_2]\right] = N^{-3}\,\mathrm{Tr}[\mathbf{X}_1\mathbf{X}_2]\,\mathrm{Tr}[\mathbf{X}_1] + O(N^{-2}),$$
$$\mathbb{E}_{\mathbf{Y},\mathbf{Y}'}\left[\mathrm{Tr}[\mathbf{Y}'\mathbf{X}_1\mathbf{Y}'\mathbf{X}_1\mathbf{Y}\mathbf{X}_1\mathbf{Y}\mathbf{X}_2]\right] = N^{-4}\,\mathrm{Tr}[\mathbf{X}_1\mathbf{X}_2]\,\mathrm{Tr}[\mathbf{X}_1]\,\mathrm{Tr}[\mathbf{X}_2] + O(N^{-2}),$$
$$\mathbb{E}_{\mathbf{Y},\mathbf{Y}'}\left[\mathrm{Tr}[\mathbf{Y}'\mathbf{X}_1\mathbf{Y}\mathbf{X}_1\mathbf{Y}'\mathbf{X}_1\mathbf{Y}\mathbf{X}_2]\right] = 3N^{-4}3\,\mathrm{Tr}[\mathbf{X}_1\mathbf{X}_2]\,\mathrm{Tr}[\mathbf{X}_1^2] + O(N^{-3}) = O(N^{-2}).$$
(183)

New terms with $\mathbf{X}_2\mathbf{Y}\mathbf{X}_2$:

$$\mathbb{E}_{\mathbf{Y},\mathbf{Y}'}[\mathrm{Tr}[\mathbf{Y}'\mathbf{X}_1\mathbf{Y}'\mathbf{X}_1\mathbf{Y}\mathbf{X}_2\mathbf{Y}\mathbf{X}_2]] = N^{-4}\,\mathrm{Tr}[\mathbf{X}_1\mathbf{X}_2]\,\mathrm{Tr}[\mathbf{X}_1]\,\mathrm{Tr}[\mathbf{X}_2] + O(N^{-2}),$$
$$\mathbb{E}_{\mathbf{Y},\mathbf{Y}'}[\mathrm{Tr}[\mathbf{Y}'\mathbf{X}_1\mathbf{Y}\mathbf{X}_1\mathbf{Y}\mathbf{X}_2\mathbf{Y}\mathbf{X}_2]] = N^{-4}\,\mathrm{Tr}[\mathbf{X}_1\mathbf{X}_2]\,\mathrm{Tr}[\mathbf{X}_1]\,\mathrm{Tr}[\mathbf{X}_2] + O(N^{-2}),$$
$$\mathbb{E}_{\mathbf{Y},\mathbf{Y}'}[\mathrm{Tr}[\mathbf{Y}'\mathbf{X}_1\mathbf{Y}'\mathbf{X}_1\mathbf{Y}\mathbf{X}_1\mathbf{Y}\mathbf{X}_2\mathbf{Y}\mathbf{X}_2]] = N^{-5}\,\mathrm{Tr}[\mathbf{X}_1\mathbf{X}_2]\,\mathrm{Tr}[\mathbf{X}_1]^2\,\mathrm{Tr}[\mathbf{X}_2] + O(N^{-2}),$$
$$\mathbb{E}_{\mathbf{Y},\mathbf{Y}'}[\mathrm{Tr}[\mathbf{Y}'\mathbf{X}_1\mathbf{Y}\mathbf{X}_1\mathbf{Y}'\mathbf{X}_1\mathbf{Y}\mathbf{X}_2\mathbf{Y}\mathbf{X}_2]] =$$
$$= 3N^{-5}\,\mathrm{Tr}[\mathbf{X}_1\mathbf{X}_2]\,\mathrm{Tr}[\mathbf{X}_1^2]\,\mathrm{Tr}[\mathbf{X}_2] + O(N^{-3}) = O(N^{-2}).$$
(184)

New terms with $\mathbf{X}_2\mathbf{Y}'\mathbf{X}_2$:

$$\mathbb{E}_{\mathbf{Y},\mathbf{Y}'}[\mathrm{Tr}[\mathbf{Y}'\mathbf{X}_1\mathbf{Y}\mathbf{X}_1\mathbf{Y}\mathbf{X}_2\mathbf{Y}'\mathbf{X}_2]] = N^{-4}\,\mathrm{Tr}[\mathbf{X}_1\mathbf{X}_2]\,\mathrm{Tr}[\mathbf{X}_1]\,\mathrm{Tr}[\mathbf{X}_2] + O(N^{-2}),$$
$$\mathbb{E}_{\mathbf{Y},\mathbf{Y}'}[\mathrm{Tr}[\mathbf{Y}'\mathbf{X}_1\mathbf{Y}'\mathbf{X}_1\mathbf{Y}\mathbf{X}_1\mathbf{Y}\mathbf{X}_2\mathbf{Y}'\mathbf{X}_2]] = N^{-5}\,\mathrm{Tr}[\mathbf{X}_1\mathbf{X}_2]\,\mathrm{Tr}[\mathbf{X}_1]^2\,\mathrm{Tr}[\mathbf{X}_2] + O(N^{-2}),$$
$$\mathbb{E}_{\mathbf{Y},\mathbf{Y}'}[\mathrm{Tr}[\mathbf{Y}'\mathbf{X}_1\mathbf{Y}\mathbf{X}_1\mathbf{Y}'\mathbf{X}_1\mathbf{Y}\mathbf{X}_2\mathbf{Y}'\mathbf{X}_2]] =$$
$$= 3N^{-5}\,\mathrm{Tr}[\mathbf{X}_1\mathbf{X}_2]\,\mathrm{Tr}[\mathbf{X}_1^2]\,\mathrm{Tr}[\mathbf{X}_2] + O(N^{-3}) = O(N^{-2}).$$
(185)

New terms with $\mathbf{X}_2\mathbf{Y}\mathbf{X}_2\mathbf{Y}'\mathbf{X}_2$:

$$\mathbb{E}_{\mathbf{Y},\mathbf{Y}'}[\mathrm{Tr}[\mathbf{Y}'\mathbf{X}_1\mathbf{Y}'\mathbf{X}_1\mathbf{Y}\mathbf{X}_1\mathbf{Y}\mathbf{X}_2\mathbf{Y}\mathbf{X}_2\mathbf{Y}'\mathbf{X}_2]] =$$
$$= N^{-6}\,\mathrm{Tr}[\mathbf{X}_1\mathbf{X}_2]\,\mathrm{Tr}[\mathbf{X}_1]^2\,\mathrm{Tr}[\mathbf{X}_2]^2 + O(N^{-2}),$$
$$\mathbb{E}_{\mathbf{Y},\mathbf{Y}'}[\mathrm{Tr}[\mathbf{Y}'\mathbf{X}_1\mathbf{Y}\mathbf{X}_1\mathbf{Y}'\mathbf{X}_1\mathbf{Y}\mathbf{X}_2\mathbf{Y}\mathbf{X}_2\mathbf{Y}'\mathbf{X}_2]] =$$
$$= 3N^{-6}\,\mathrm{Tr}[\mathbf{X}_1\mathbf{X}_2]\,\mathrm{Tr}[\mathbf{X}_1]^2\,\mathrm{Tr}[\mathbf{X}_2]^2 + O(N^{-3}) = O(N^{-2}).$$
(186)

A new term with $\mathbf{X}_2\mathbf{Y}'\mathbf{X}_2\mathbf{Y}\mathbf{X}_2$:

$$\mathbb{E}_{\mathbf{Y},\mathbf{Y}'}[\mathrm{Tr}[\mathbf{Y}'\mathbf{X}_1\mathbf{Y}\mathbf{X}_1\mathbf{Y}'\mathbf{X}_1\mathbf{Y}\mathbf{X}_2\mathbf{Y}'\mathbf{X}_2\mathbf{Y}\mathbf{X}_2]] =$$
$$= N^{-6}\left(6\,\mathrm{Tr}[\mathbf{X}_1\mathbf{X}_2]^3 + 9\,\mathrm{Tr}[\mathbf{X}_1\mathbf{X}_2]\,\mathrm{Tr}[\mathbf{X}_1]^2\,\mathrm{Tr}[\mathbf{X}_2]^2\right) + O(N^{-4}) = O(N^{-3}).$$
(187)

To summarize, we observe two types of monomials. The first type has the order $O(N^{-2})$ and is composed of those monomials which take $\mathbf{X}_1\mathbf{Y}\mathbf{X}_1\mathbf{Y}'\mathbf{X}_1$ from $\widetilde{q}(\mathbf{X}_1, \mathbf{Y}, \mathbf{Y}', a_1, a_1')$ and/or $\mathbf{X}_2\mathbf{Y}'\mathbf{X}_2\mathbf{Y}\mathbf{X}_2$ from $\widetilde{q}(\mathbf{X}_2, \mathbf{Y}, \mathbf{Y}', a_2, a_2')$. The rest of the monomials $g(\cdot, \cdot, \cdot, \cdot)$, contain exactly one factor $\mathrm{Tr}[\mathbf{X}_1\mathbf{X}_2] = N\frac{r_1 - r_2}{z_1 - z_2}$, have the order $O(N^{-1})$, and satisfy

$$\mathbb{E}_{\mathbf{Y},\mathbf{Y}'}\left[g(\mathbf{X}_1, \mathbf{X}_2, \mathbf{Y}, \mathbf{Y}')\right] = \frac{1}{N}\frac{r_1 - r_2}{z_1 - z_2}\frac{g(r_1, r_2, 1, 1)}{r_1 r_2} + O(N^{-2})$$
$$= -\frac{1}{N}\frac{r_1^{-1} - r_2^{-1}}{z_1 - z_2}g(r_1, r_2, 1, 1) + O(N^{-2}).$$
(188)

Thus, the leading $O(N^{-1})$ term for the second moment with $l \neq l'$ is given by

$$R_{ll'l'l}(z_1, z_2) = -\frac{1}{N}\frac{r_1^{-1} - r_2^{-1}}{z_1 - z_2}\prod_{s=1,2}\left[r_s - (a_s + a_s')r_s^2 + a_s a_s' r_s^3\right] + O(N^{-2})$$
$$= -\frac{1}{N}\frac{r_1^{-1} - r_2^{-1}}{z_1 - z_2}\frac{r_1^{-1} r_2^{-1}}{(\lambda_l + r_1^{-1})(\lambda_{l'} + r_1^{-1})(\lambda_l + r_2^{-1})(\lambda_{l'} + r_2^{-1})} + O(N^{-2}).$$
(189)

Again, in the limit $z_1 \to z_2$, the result above coincides with previously derived (171) up to subleading $O(N^{-2})$ terms.

**Summations over $l'$ and $l$ in (44).** For the loss functional we will need not the bare second moment $R_{ll'l'l}(z_1, z_2)$ but the sum $\sum_{l'} \lambda_{l'}^2 R_{ll'l'l}(z_1, z_2)$. First, let us start with the case $z_1 = z_2$ where the second moment is given by (171). The essential sum to compute is $\sum_{l'} \frac{\lambda_{l'}^2}{(\lambda_{l'} + r^{-1})^2}$. Using the fixed-point equation (161) gives

$$\frac{1}{N} \sum_{l'} \frac{\partial_z r^{-1} \lambda_{l'}^2}{(\lambda_{l'} + r^{-1})^2} = \frac{1}{N} \sum_{l'} \frac{\partial}{\partial z} \frac{r^{-1} \lambda_{l'}}{\lambda_{l'} + r^{-1}} = \frac{\partial}{\partial z}(r^{-1} + z) = 1 + \partial_z r^{-1}. \tag{190}$$

We can substitute this result into the second moment to get the desired second moment sum.

$$\sum_{l'} \lambda_{l'}^2 R_{ll'l'l}(z, z) = \frac{\lambda_l^2 - (1 + \partial_z r^{-1}) r^{-2}}{(\lambda_l + r^{-1})^2}. \tag{191}$$

Now, we proceed to the case $z_1 \neq z_2$. Similarly, from (189) we see that the essential sum to compute is

$$\frac{1}{N} \frac{r_1^{-1} - r_2^{-1}}{z_1 - z_2} \sum_{l'} \frac{\lambda_{l'}^2}{(\lambda_{l'} + r_1^{-1})(\lambda_{l'} + r_1^{-1})} = \frac{1}{N} \frac{1}{z_1 - z_2} \sum_{l'} \left( \frac{\lambda_{l'} r_1^{-1}}{\lambda_{l'} + r_1^{-1}} - \frac{\lambda_{l'} r_2^{-1}}{\lambda_{l'} + r_2^{-1}} \right)$$
$$= 1 + \frac{r_1^{-1} - r_2^{-1}}{z_1 - z_2}, \tag{192}$$

which is basically a finite difference version of the sum for $z_1 = z_2$. Again, we substitute this result into second moment sum to get

$$\sum_{l'} \lambda_{l'}^2 R_{ll'l'l}(z_1, z_2) = \frac{\lambda_l^2 - r_1^{-1} r_2^{-1} \left(1 + \frac{r_1^{-1} - r_2^{-1}}{z_1 - z_2}\right)}{(\lambda_l + r_1^{-1})(\lambda_l + r_2^{-1})} + O(N^{-1}). \tag{193}$$

Now let us consider the sum over $l$ in (44). First, observe that the first-moment term from (44) enters in the form

$$\sum_l c_l^2 \lambda_l R_{ll}(z) = \sum_l \frac{\lambda_l c_l^2}{\lambda_l + r^{-1}(z)} = u(z). \tag{194}$$

Here we encountered the first auxiliary function $u(z)$ introduced (14). However, directly performing the respective sum for the second-moment term does not automatically reduce it to an expression with "decoupled" factors (depending only on $z_1$ or $z_2$ but not jointly on $z_1, z_2$). Yet, we can perform an additional transformation to express the result in terms of decoupled factors. Representation of fractions product as a difference, similar to (192), gives for two parts of $\sum_{l,l'} c_l^2 \lambda_{l'}^2 R_{ll'l'l}(z_1, z_2)$

$$\sum_l \frac{c_l^2 (\lambda_l^2 - r_1^{-1} r_2^{-1})}{(\lambda_l + r_1^{-1})(\lambda_l + r_2^{-1})} = \sum_l \left( \frac{\lambda_l c_l^2}{\lambda_l + r_1^{-1}} + \frac{\lambda_l c_l^2}{\lambda_l + r_2^{-1}} - c_l^2 \right) = u_1 + u_2 - \sum_l c_l^2, \tag{195}$$

$$\sum_l \frac{-c_l^2 r_1^{-1} r_2^{-1} \frac{r_1^{-1} - r_2^{-1}}{z_1 - z_2}}{(\lambda_l + r_1^{-1})(\lambda_l + r_2^{-1})} = r_1^{-1} r_2^{-1} \frac{\sum_l \frac{c_l^2}{\lambda_l + r_1^{-1}} - \sum_l \frac{c_l^2}{\lambda_l + r_2^{-1}}}{z_1 - z_2} = r_1^{-1} r_2^{-1} \frac{v_1 - v_2}{z_1 - z_2}, \tag{196}$$

where we have encountered second auxiliary function $v(z)$ defined in (14). Summarizing our computation of the second moment, we have obtained

$$\sum_{l,l'} c_l^2 \lambda_{l'}^2 R_{ll'l'l}(z_1, z_2) = u(z_1) + u(z_2) + r^{-1}(z_1) r^{-1}(z_2) \frac{v(z_1) - v(z_2)}{z_1 - z_2} - \sum_l c_l^2. \tag{197}$$

Thus, we fully expressed the population sums of resolvent moments in terms of two auxiliary functions $v(z)$ and $u(z)$, which would later define our final result for the loss functional.

### E.2 Loss functional

In this section, we using the resolvent moments derived in Section E.1.1 to compute the loss functional (44).

**KRR case.** As a sanity check, let's first compute the loss for the case of KRR learning algorithm $h_\eta(z) = \frac{z}{z+\eta}$, with the expectation to recover the original expression (1). In the absence of target noise $\sigma^2 = 0$, we have

$$L_l[h_\eta] \overset{\sigma^2=0}{=} \sum_{l'} \lambda_{l'}^2 R_{ll'l'l}(-\eta, -\eta) - 2\lambda_l R_{ll}(-\eta) + 1$$

$$= \frac{\lambda_l^2 - \eta_{\text{eff}}^2(1 - \partial_\eta \eta_{\text{eff}})}{(\lambda_l + \eta_{\text{eff}})^2} - 2\frac{\lambda_l}{\lambda_l + \eta_{\text{eff}}} + 1 = \frac{\eta_{\text{eff}}^2 \partial_\eta \eta_{\text{eff}}}{(\lambda_l + \eta_{\text{eff}})^2}. \tag{198}$$

For the contribution of noise term we have

$$\varepsilon_l^{(2)} = \lambda_l^2 \frac{1}{N}\mathbb{E}\left[\phi_l^T \widehat{\mathbf{R}}^2(-\eta)\phi_l\right] = -\lambda_l^2 \partial_\eta \frac{1}{N}\mathbb{E}\left[\phi_l^T \widehat{\mathbf{R}}(-\eta)\phi_l\right]$$

$$= -\lambda_l^2 \partial_\eta R_{ll}(-\eta) = \frac{\lambda_l^2 \partial_\eta \eta_{\text{eff}}}{(\lambda_l + \eta_{\text{eff}})^2}. \tag{199}$$

Combining noiseless and noise contributions, we obtain

$$L_{\text{KRR}}(\eta) = \frac{1}{2}\frac{\eta_{\text{eff}}^2 \partial_\eta \eta_{\text{eff}}}{(\lambda_l + \eta_{\text{eff}})^2} + \frac{\sigma^2}{2N}\frac{\lambda_l^2 \partial_\eta \eta_{\text{eff}}}{(\lambda_l + \eta_{\text{eff}})^2}, \tag{200}$$

which is the same as (1).

**General case.** Now we turn to our main goal of describing learning measures $\rho^{(2)}, \rho^{(1)}, \rho^{(\varepsilon)}$. Denote, $u(\lambda) = \lim_{y\to 0_+} u(\lambda + \mathbf{i}y)$ and $v(\lambda) = \lim_{y\to 0_+} v(\lambda + \mathbf{i}y)$. Then, for the first moment of the learning measure, we have

$$\rho^{(1)}(d\lambda) = \sum_{l,l'} c_l c_{l'} \rho_{ll'}^{(1)}(d\lambda) = \frac{1}{\pi\lambda}\lim_{y\to 0_+}\sum_l c_l^2 \lambda_l \Im R_{ll}(\lambda + \mathbf{i}y)d\lambda = \frac{\Im u(\lambda)}{\pi\lambda}d\lambda. \tag{201}$$

For noise measure $\rho_l^{(\varepsilon)}(d\lambda)$ we similarly obtain

$$\rho^{(\varepsilon)}(d\lambda) = \frac{1}{\pi\lambda^2}\lim_{y\to 0_+}\sum_l \lambda_l^2 \Im R_{ll}(\lambda + \mathbf{i}y)d\lambda = \frac{\Im w(\lambda)}{\pi\lambda^2}d\lambda, \tag{202}$$

where $w(\lambda) = \lim_{y\to 0_+} w(\lambda + \mathbf{i}y)$ for the last auxiliary function $w(z)$ defined in (14).

Now we proceed to the computation for the second moment of the learning measure $\rho^{(2)}(d\lambda_1, d\lambda_2)$ using the relation (152). We have

$$\rho^{(2)}(d\lambda_1, d\lambda_2) = \frac{1}{\pi^2 \lambda_1 \lambda_2}\lim_{\substack{y_1\to 0_+ \\ y_2\to 0_+}}\sum_{l,l'} c_l^2 \lambda_{l'}^2 R_{l_1 l_1' l_2' l_2}^{(\text{im})}(\lambda_1 + \mathbf{i}y_1, \lambda_2 + \mathbf{i}y_2)d\lambda_1 d\lambda_2. \tag{203}$$

Note that while for the first moment we simply have $R_{ll'}^{(\text{im})}(z) = \Im R_{ll'}(z)$, the relation between $R_{l_1 l_1' l_2' l_2}^{(\text{im})}(z_1, z_2)$, and previously computed $R_{l_1 l_1' l_2' l_2}(z_1, z_2)$ is less straightforward since the product of two imaginary parts has to be taken out of expectation. This can be done with the following trick: for two random variables $w_1, w_2$ we have

$$\mathbb{E}\left[\Im w_1 \Im w_2\right] = \mathbb{E}\left[\Re\frac{w_1 \overline{w_2} - w_1 w_2}{2}\right] = \Re\frac{\mathbb{E}\left[w_1 \overline{w_2}\right] - \mathbb{E}\left[w_1 w_2\right]}{2}. \tag{204}$$

Taking $w_1 = \widehat{R}_{ll'}(z_1), w_2 = \widehat{R}_{l'l}(z_2)$, and noting that $\overline{\widehat{\mathbf{R}}(z)} = \widehat{\mathbf{R}}(\overline{z})$, we get the desired second moment of imaginary parts

$$R_{ll'l'l}^{(\text{im})}(z_1, z_2) = \mathbb{E}\left[\Im\widehat{R}_{ll'}(z_1)\Im\widehat{R}_{l'l}(z_2)\right] = \Re\left[\frac{R_{ll'l'l}(z_1, \overline{z_2}) - R_{ll'l'l}(z_1, z_2)}{2}\right]. \tag{205}$$

Observe that the part $u(z_1) + u(z_2) - \sum_l c_l^2$ of $\sum_{l,l'} c_l^2 \lambda_{l'}^2 R_{l'll'l}(z_1, z_2)$ gives no contribution when substituted into (205):

$$\Re\left[\left(u(z_1) + u(\overline{z_2}) - \sum_l c_l^2\right) - \left(u(z_1) + u(z_2) - \sum_l c_l^2\right)\right] = \Re\left[\overline{u(z_2)} - u(z_2)\right] = 0. \tag{206}$$

Indeed, the remaining part has a non-trivial contribution that we will compute below in the limit $z_1 \to \lambda_1 + \mathbf{i}0$, $z_2 \to \lambda_2 + \mathbf{i}0$.

$$
\begin{aligned}
\lim_{\substack{y_1 \to 0_+ \\ y_2 \to 0_+}} & \sum_{l,l'} c_l^2 \lambda_{l'}^2 R_{l_1 l_1' l_2' l_2}^{(\text{im})}(\lambda_1 + \mathbf{i}y_1, \lambda_2 + \mathbf{i}y_2) \\
= & -\frac{1}{2} \Re \lim_{\substack{y_1 \to 0_+ \\ y_2 \to 0_+}} r^{-1}(\lambda_1 + \mathbf{i}y_1) r^{-1}(\lambda_2 + \mathbf{i}y_2) \frac{v(\lambda_1 + \mathbf{i}y_1) - v(\lambda_2 + \mathbf{i}y_2)}{\lambda_1 - \lambda_2 + \mathbf{i}(y_1 - y_2)} \\
& + \frac{1}{2} \Re \lim_{\substack{y_1 \to 0_+ \\ y_2 \to 0_+}} r^{-1}(\lambda_1 + \mathbf{i}y_1) r^{-1}(\lambda_2 - \mathbf{i}y_2) \frac{v(\lambda_1 + \mathbf{i}y_1) - v(\lambda_2 - \mathbf{i}y_2)}{\lambda_1 - \lambda_2 + \mathbf{i}(y_1 + y_2)}.
\end{aligned}
\tag{207}
$$

The first limit here can be easily taken in joint way $y_1 = y_2 = y \to 0_+$, leading to

$$
\lim_{\substack{y_1 \to 0_+ \\ y_2 \to 0_+}} r^{-1}(\lambda_1 + \mathbf{i}y_1) r^{-1}(\lambda_2 + \mathbf{i}y_2) \frac{v(\lambda_1 + \mathbf{i}y_1) - v(\lambda_2 + \mathbf{i}y_2)}{\lambda_1 - \lambda_2 + \mathbf{i}(y_1 - y_2)} = \frac{v(\lambda_1) - v(\lambda_2)}{r(\lambda_1) r(\lambda_2)(\lambda_1 - \lambda_2)}.
\tag{208}
$$

Note that this expression is regular at $\lambda_1 = \lambda_2$ since we assume $v(\lambda)$ to be differentiable.

The second limit might have a non-vanishing singularity at $\lambda_1 = \lambda_2$, for which we will need to use Sokhotski–Plemelj formula $\lim_{\varepsilon \to 0_+} \frac{1}{x - \mathbf{i}\varepsilon} = \mathbf{i}\pi\delta(x) + \mathcal{P}(\frac{1}{x})$, where $\mathcal{P}$ denotes the Cauchy principal value.

$$
\begin{aligned}
\lim_{\substack{y_1 \to 0_+ \\ y_2 \to 0_+}} & r^{-1}(\lambda_1 + \mathbf{i}y_1) r^{-1}(\lambda_2 - \mathbf{i}y_2) \frac{v(\lambda_1 + \mathbf{i}y_1) - v(\lambda_2 - \mathbf{i}y_2)}{\lambda_1 - \lambda_2 + \mathbf{i}(y_1 + y_2)} \\
& = \lim_{y_1 \to 0_+} r^{-1}(\lambda_1 + \mathbf{i}y_1) \overline{r^{-1}(\lambda_2)} \frac{v(\lambda_1 + \mathbf{i}y_1) - \overline{v(\lambda_2)}}{\lambda_1 - \lambda_2 + \mathbf{i}y_1} \\
& = |r^{-1}(\lambda_1)|^2 2\pi \Im\{v(\lambda_1)\} \delta(\lambda_1 - \lambda_2) + r^{-1}(\lambda_1) \overline{r^{-1}(\lambda_2)} \mathcal{P}\left(\frac{v(\lambda_1) - \overline{v(\lambda_2)}}{\lambda_1 - \lambda_2}\right).
\end{aligned}
\tag{209}
$$

Note that the singularity at $\lambda_1 = \lambda_2$ under the Cauchy principal value is purely imaginary and, therefore, will disappear after taking the real part in (207). Next, let us combine (208) with the second term from (209)

$$
\begin{aligned}
\frac{1}{2} \Re & \left\{ \frac{r^{-1}(\lambda_1)\overline{r^{-1}(\lambda_2)}\big(v(\lambda_1) - \overline{v(\lambda_2)}\big) - r^{-1}(\lambda_1)r^{-1}(\lambda_2)\big(v(\lambda_1) - v(\lambda_2)\big)}{\lambda_1 - \lambda_2} \right\} \\
& = \frac{\Im\{r^{-1}(\lambda_2)\}\Im\{r^{-1}(\lambda_1)v(\lambda_1)\} - \Im\{r^{-1}(\lambda_1)\}\Im\{r^{-1}(\lambda_2)v(\lambda_2)\}}{\lambda_1 - \lambda_2} \\
& = \frac{\Im\{u(\lambda_2)\}\Im\{r^{-1}(\lambda_1)\} - \Im\{u(\lambda_1)\}\Im\{r^{-1}(\lambda_2)\}}{\lambda_1 - \lambda_2},
\end{aligned}
\tag{210}
$$

where in the last line we have used the relation $r^{-1}v = \sum_l c_l^2 - u$. Finally, we combine all the terms into the learning measure second moment

$$
\begin{aligned}
\frac{\rho^{(2)}(d\lambda_1, d\lambda_2)}{d\lambda_1 d\lambda_2} = & \frac{|r^{-1}(\lambda_1)|^2}{\pi \lambda_1^2} \Im\{v(\lambda_1)\} \delta(\lambda_1 - \lambda_2) \\
& + \frac{1}{\pi^2 \lambda_1 \lambda_2} \frac{\Im\{u(\lambda_2)\}\Im\{r^{-1}(\lambda_1)\} - \Im\{u(\lambda_1)\}\Im\{r^{-1}(\lambda_2)\}}{\lambda_1 - \lambda_2}.
\end{aligned}
\tag{211}
$$

Thus, we have derived the expressions (15) and (16) of the main paper.

### E.3 POWER-LAW ANSATZ

The analysis of the loss functional given by (15) and (16) is much more tractable for continuous approximation of our basic power distributions (2)

$$
\sum_l \delta_{\lambda_l}(d\lambda) \to \frac{1}{\nu} \lambda^{-\frac{1}{\nu} - 1} d\lambda = \mu_\lambda(d\lambda), \qquad \sum_l c_l^2 \delta_{\lambda_l}(d\lambda) \to \frac{1}{\nu} \lambda^{\frac{\kappa}{\nu} - 1} d\lambda = \mu_c(d\lambda).
\tag{212}
$$

In particular, the fixed-point equation (161) for Stieltjes transform $r(z)$ becomes

$$z = -r^{-1}(z) + \frac{1}{N} \int_0^1 \frac{r^{-1}(z)\frac{1}{\nu}\lambda^{-\frac{1}{\nu}} d\lambda}{\lambda + r^{-1}(z)}. \tag{213}$$

We will encounter many integrals similar to the one above, with the general form

$$F_a(x) \equiv \int_0^1 \frac{\lambda^a d\lambda}{\lambda + x}, \quad a > -1, \quad x \notin [-1, 0]. \tag{214}$$

Such integral can be expressed in terms of Hypergeometric function $_2F_1$. This can be immediately used to get asymptotic $x \to 0$ expansion, which will be very useful later

$$F_a(x) = \frac{_2F_1\left(1, 1+a, 2+a, -\frac{1}{x}\right)}{(1+a)x} = -\frac{\pi}{\sin(\pi a)}x^a + \frac{1}{a} + \frac{x}{1-a} + O(x^2). \tag{215}$$

For this asymptotic expansion, we assume that $x$ has a cut along $\mathbb{R}_-$ and $a \notin \mathbb{Z}$.

Below we describe the essential steps in computing the loss functional for the Wishart model.

### E.3.1 SOLVING FIXED POINT EQUATION

In this section, we will analyze asymptotic $N \to \infty$ solutions of fixed-point equation (213). Note that we are interested in the solution of this equation when $z$ approaches the real line from above:

$$r(\lambda) = \lim_{\varepsilon \to 0_+} r(\lambda + \mathbf{i}\varepsilon). \tag{216}$$

First, let us find values of $\lambda$ when $\Im r(\lambda) = \pi\mu(\lambda) > 0$, which corresponds to support of the empirical spectral density $\mu(\lambda)$ of $\mathbf{K}$. For this, let us write $r^{-1}(\lambda) = -\tau - \mathbf{i}\upsilon$, and rewrite (213) in the limit $\varepsilon \to 0$ as a pair of real equations

$$\lambda = \tau + \frac{1}{N} \int \frac{\lambda'(\tau^2 + \upsilon^2) - (\lambda')^2\tau}{(\lambda' - \tau)^2 + \upsilon^2}\mu_\lambda(d\lambda')$$

$$0 = \upsilon - \frac{1}{N} \int \frac{(\lambda')^2\upsilon}{(\lambda' - \tau)^2 + \upsilon^2}\mu_\lambda(d\lambda') \tag{217}$$

$$= \upsilon \left(1 - \frac{1}{N} \int \frac{(\lambda')^2}{(\lambda' - \tau)^2 + \upsilon^2}\mu_\lambda(d\lambda')\right).$$

Now, let us fix value of $r$ corresponding to the point outside of the support of $\mu$, where $\upsilon = 0$, and therefore $\tau \notin \text{supp}(\mu_\lambda) = [0, 1]$ to ensure convergence of the integral. Since the solution of the fixed point equation for $z \in \mathbb{C}_+$ is unique, there should be no value of $\lambda$ and $\upsilon$ satisfying equations (217). But since $\lambda$ can be defined for any value of $\tau, \upsilon$, the second equation should have no solutions with $\upsilon \neq 0$. Due to the monotonicity of the expression in the brackets, this gives a necessary condition for $\tau$ corresponding to the point outside the support:

$$\frac{1}{N} \int \frac{(\lambda')^2}{(\lambda' - \tau)^2}\mu_\lambda(d\lambda') < 1. \tag{218}$$

Additionally, it is easy to see that for the values of $\tau$ not satisfying the inequality above, there is a solution of the second equation in (217) with $\upsilon < 0$, which induces the solution of the first equation with some $\lambda$, meaning that the triple $(\lambda, \tau, \upsilon)$ corresponds the point inside of the support of $\mu$.

The argument above fully characterizes the support of $\mu$: there two support edges $\lambda_-, \lambda_+$ with the respective values $\tau_- < 0, \tau_+ > 1$ given by the equality version of (218). The right edge $\tau_+$ should be at a distance $\sim N^{-1}$ from $\lambda = 1$, where we have

$$\frac{1}{N} \int \frac{(\lambda')^2}{(\lambda' - \tau_+)^2}\mu_\lambda(d\lambda') = \frac{1 + o(1)}{N} \int_0^1 \frac{\mu_\lambda(1)}{(\lambda' - \tau_+)^2} d\lambda'$$

$$= \frac{\mu_\lambda(1)}{N}\left(\frac{1}{\tau_+ - 1} - \frac{1}{\tau_+}\right)(1 + o(1)) = 1. \tag{219}$$

From the calculation above and the first equation in (217), the right edge is given by

$$
\begin{aligned}
\tau_+ &= 1 + \frac{\mu_\lambda(1)}{N}(1 + o(1)) \\
\lambda_+ &= \tau_+ - \frac{\mu_\lambda(1)}{N} \log\left(\tau_+ - 1\right)(1 + o(1)) = 1 + \mu_\lambda(1)\frac{\log N}{N}(1 + o(1)).
\end{aligned}
\tag{220}
$$

Turning to the left edge of the support, we note that it has the order $\tau_- \sim N^{-\nu}$. Thus, we can use asymptotic expansion of Hypergeometric function (215) with $a = -\frac{1}{\nu}$. leading to asymptotic form of fixed point equation

$$
z = -r^{-1}(z) + \frac{C_\nu}{N} r^{-1+\frac{1}{\nu}}(z), \quad C_\nu = \frac{\pi/\nu}{\sin(\pi/\nu)},
\tag{221}
$$

where we, for simplicity, do not write $(1 + o(1))$ correction factors in the asymptotic.

Then, for the left edge of the support we have

$$
1 = -\frac{1}{N}\frac{1}{\nu}\frac{\partial F_{-\frac{1}{\nu}}(-\tau_- - \mathbf{i}\upsilon)}{\mathbf{i}\partial\upsilon}\Bigg|_{\upsilon=0} = -\frac{C_\nu}{N}\frac{\partial(-\tau_- - \mathbf{i}\upsilon)^{1-\frac{1}{\nu}}}{\mathbf{i}\partial\upsilon}\Bigg|_{\upsilon=0} = \frac{C_\nu}{N}\frac{\nu-1}{\nu}(-\tau_-)^{-\frac{1}{\nu}},
\tag{222}
$$

which gives the respective left edge values

$$
\begin{aligned}
\tau_- &= -\left(\tfrac{\nu-1}{\nu}C_\nu\right)^\nu N^{-\nu}, \\
\lambda_- &= \frac{1}{\nu-1}(-\tau_-) = \frac{(\nu-1)^{\nu-1}\left(C_\nu\right)^\nu}{\nu^\nu} N^{-\nu}.
\end{aligned}
\tag{223}
$$

Now, let us give more explicit solutions of fixed-point equation in different scenarios.

**To the left of the support ($\lambda < \lambda_-$).** In this region the solution of fixed point equation is real, and will in fact has a lot of parallels with KRR applied to the Wishart model. Thus, we translate $\lambda$ and $r(\lambda)$ to their KRR notations: $\eta = -\lambda$ is the KRR regularization and $\eta_{\text{eff}} = r^{-1}$ is the effective implicit regularization, appearing in (1). In these notations, the asymptotic form of fixed point equation becomes

$$
1 = \frac{\eta}{\eta_{\text{eff}}} + \frac{C_\nu}{N}\eta_{\text{eff}}^{-\frac{1}{\nu}}.
\tag{224}
$$

When $\eta_{\text{eff}}$ has the scaling $s \geq \nu$, we can write $\eta_{\text{eff}} = \widetilde{\eta}_{\text{eff}} N^{-\nu}$ and $\eta = \widetilde{\eta} N^{-\nu}$, which satisfies $N$-independent equation

$$
1 = \frac{\widetilde{\eta}}{\widetilde{\eta}_{\text{eff}}} + C_\nu \widetilde{\eta}_{\text{eff}}^{-\frac{1}{\nu}}.
\tag{225}
$$

The equation above gives a nontrivial relations between $\widetilde{\eta}$ and $\widetilde{\eta}_{\text{eff}}$. However, when $N^{-\nu} \ll \eta_{\text{eff}} \ll 1$, or in other words it has the scaling $0 < s < \nu$, the relation simplify to an explicit one:

$$
\eta_{\text{eff}} = \eta + \frac{C_\nu}{N}\eta^{1-\frac{1}{\nu}}.
\tag{226}
$$

**Inside the support ($\lambda_- < \lambda < \lambda_+$).** In this region it is convenient to write $r(\lambda) = r_0(\lambda)e^{i\phi(\lambda)}$, with the phase taking values in the upper half-circle: $0 < \phi < \pi$. Substituting $r_0(\lambda)e^{i\phi(\lambda)}$ into the limit form (221) of fixed point equation we get a pair of real equations

$$
\lambda = r_0^{-1}\left(-\cos\phi + \frac{C_\nu}{N}(r_0)^{\frac{1}{\nu}}\cos\left((1-\tfrac{1}{\nu})\phi\right)\right)
\tag{227}
$$

$$
0 = r_0^{-1}\sin\phi - \frac{C_\nu}{N}r_0^{-1+\frac{1}{\nu}}\sin\left((1-\tfrac{1}{\nu})\phi\right).
\tag{228}
$$

Let us rewrite the second equation here using the left edge value $r_- = (-\tau_-)^{-1}$

$$
r_0^{-1} = r_-^{-1}\left(\frac{(1-\tfrac{1}{\nu})\sin\phi}{\sin\left((1-\tfrac{1}{\nu})\phi\right)}\right)^{-\nu} = N^{-\nu}\left(\frac{\sin\phi}{C_\nu\sin\left((1-\tfrac{1}{\nu})\phi\right)}\right)^{-\nu}.
\tag{229}
$$

Thus, we see that the solution of fixed point inside the support is mostly conveniently described by using the phase $\phi$ as a free variable, and then specify the rest of the variables by the mappings $\phi \mapsto r_0$ and $(r_0, \phi) \mapsto \lambda$ given by equations (229) and (227) respectively. In this representation, $\phi = 0$ corresponds to the left edge $\lambda_-$ of the support, while $\phi \to \pi$ corresponds to the right edge of the support.

As for outside of the support case, we see that on the scale $N^{-\nu}$ the fixed-point equation admits $N$-independent form. Specifically, set $\lambda = \widetilde{\lambda} N^{-\nu}$ and $r_0 = \widetilde{r}_0 N^{\nu}$. Then, the triplet $\widetilde{r}_0, \phi, \widetilde{\lambda}$ satisfies

$$
\begin{aligned}
\widetilde{\lambda} &= -(\widetilde{r}_0)^{-1} \cos \phi + C_\nu (\widetilde{r}_0)^{-1+\frac{1}{\nu}} \cos \left( (1 - \tfrac{1}{\nu}) \phi \right), \\
\widetilde{r}_0^{-1} &= \left( \frac{\sin \phi}{C_\nu \sin \left( (1 - \tfrac{1}{\nu}) \phi \right)} \right)^{-\nu}.
\end{aligned}
\tag{230}
$$

Finally, we turn to the values $N^{-\nu} \ll \lambda \ll 1$, corresponding to the scaling $0 < s < \nu$. In this case, the pair of equations (227), (229) becomes

$$
\begin{aligned}
\lambda &= r_0^{-1} - \frac{C_\nu}{N} r_0^{-1+\frac{1}{\nu}} \cos(\tfrac{\pi}{\nu}) = \left( \frac{\pi/\nu}{N(\pi - \phi)} \right)^\nu \left( 1 + (\pi - \phi) \cot(\tfrac{\pi}{\nu}) \right), \\
r_0^{-1} &= \left( \frac{\pi/\nu}{N(\pi - \phi)} \right)^\nu.
\end{aligned}
\tag{231}
$$

Noting that in the leading asymptotic order $r_0 = \lambda^{-1}$, the second equation can be rewritten as

$$
\pi - \phi = \frac{\lambda^{-\frac{1}{\nu}}}{N} \frac{\pi}{\nu}.
\tag{232}
$$

We can summarize the above equations in a single complex equation

$$
r(\lambda) = -\lambda^{-1} \left( 1 - C_\nu \cos(\pi/\nu) \frac{\lambda^{-\frac{1}{\nu}}}{N} \right) + \mathbf{i} \frac{\pi}{\nu} \frac{\lambda^{-1-\frac{1}{\nu}}}{N}.
\tag{233}
$$

In particular, taking the imaginary part of Stieltjes transform above gives $\Im r(\lambda) = \frac{1}{N} \pi \mu_\lambda(\lambda)$. This implies that for $N^{-\nu} \ll \lambda \ll 1$ the (normalized) empirical eigenvalue density $N\mu(\lambda)$ coincides with population density $\mu_\lambda(\lambda)$ as expected.

### E.3.2   LEARNING MEASURES

In this section, we specify the form of empirical learning measures $\rho^{(2)}(d\lambda_1, d\lambda_2)$, $\rho^{(1)}(d\lambda)$ and $\rho^{(\varepsilon)}(d\lambda)$, derived in (201),(202) and (16) respectively, for the case of power-law distributions (212). Note that all three learning measures are expressed in terms of $r(\lambda)$ and imaginary parts of 3 auxiliary functions $v(\lambda), u(\lambda), w(\lambda)$, whose asymptotic form we will now analyze. In the continuous approximation (212) these functions are given by

$$
v(\lambda) = \frac{1}{\nu} \int_0^1 \frac{(\lambda')^{\frac{\kappa}{\nu} - 1} d\lambda'}{\lambda' + r^{-1}(\lambda)}, \quad u(\lambda) = \frac{1}{\nu} \int_0^1 \frac{(\lambda')^{\frac{\kappa}{\nu}} d\lambda'}{\lambda' + r^{-1}(\lambda)}, \quad w(\lambda) = \frac{1}{\nu} \int_0^1 \frac{(\lambda')^{1 - \frac{1}{\nu}} d\lambda'}{\lambda' + r^{-1}(\lambda)}. \tag{234}
$$

Since all of these functions have the same functional form $\int \frac{(\lambda')^a d\lambda'}{\lambda' + r^{-1}(\lambda)}$, let us write an asymptotic expansion of the imaginary part of this integral, using (215) as a basis. First, we consider $\lambda \ll 1$ and use the leading term of asymptotic expansion (215) together with asymptotic formulas (227) and (229).

$$
\begin{aligned}
\Im \int_0^1 \frac{(\lambda')^a d\lambda'}{\lambda' + r^{-1}(\lambda)} &= \Gamma(1 + a) \Gamma(-a) \Im \{ r_0^{-a} e^{-\mathbf{i} a \phi} \} \\
&= \pi \frac{\sin(a\phi)}{\sin(a\pi)} \left( \frac{N \sin \phi}{C_\nu \sin \left( (1 - \tfrac{1}{\nu}) \phi \right)} \right)^{-a\nu}.
\end{aligned}
\tag{235}
$$

Importantly, away from left edge $\lambda \gg \lambda_-$, or equivalently $\pi - \phi \ll 1$, the above expression significantly simplifies

$$\Im \int_0^1 \frac{(\lambda')^a d\lambda'}{\lambda' + r^{-1}(\lambda)} = \pi \left( \frac{N(\pi - \phi)}{\pi/\nu} \right)^{-a\nu} \left( 1 + O(\pi - \phi) \right)$$
$$= \pi \lambda^a \left( 1 + O\left( \frac{\lambda^{-\frac{1}{\nu}}}{N} \right) \right). \tag{236}$$

The the leading term above is expected: since $\Im r^{-1}(\lambda) \ll \lambda$ for $\lambda \gg \lambda_-$, the fraction can be approximated with Sokhotski formula $\Im \frac{1}{\lambda' + r^{-1}(\lambda)} \approx \pi \delta_\lambda(\lambda')$. However, our approach with asymptotic expansion of Hypergeometric function in (215) also provides an estimation of correction term to the Sokhotski formula, which would be difficult to obtain directly.

Let us now take into account the omitted subleading terms in the asymptotic expansion (215). These terms make a regular power series in $x$, and therefore their imaginary part can be estimated $O(\Im x) = O(r^{-1}(\lambda)) = O\left( \frac{\lambda^{1-\frac{1}{\nu}}}{N} \right)$. Thus, we can summarize our computation for $\lambda_- \ll \lambda \ll 1$ as

$$\Im \int_0^1 \frac{(\lambda')^a d\lambda'}{\lambda' + r^{-1}(\lambda)} = \pi \lambda^a + O\left( \frac{\lambda^{a-\frac{1}{\nu}}}{N} \right) + O\left( \frac{\lambda^{1-\frac{1}{\nu}}}{N} \right). \tag{237}$$

Note that while the asymptotic form above is mostly meaningful for $\lambda \gg \lambda_-$ when the corrections as small, we can formally use at for $\lambda \sim \lambda_-$ but the corrections become of the same order as the leading terms.

The values of functions $\Im v(\lambda)$, $\Im u(\lambda)$, $\Im w(\lambda)$, can be obtained by using either (235) or (237) depending on the scale of $\lambda$. In particular, when $\lambda_- \ll \lambda \ll 1$ we have

$$\Im v(\lambda) = \pi \mu_c(\lambda) + O\left( \frac{\lambda^{1 \wedge (\frac{\kappa}{\nu} - 1) - \frac{1}{\nu}}}{N} \right),$$
$$\Im u(\lambda) = \pi \lambda \mu_c(\lambda) + O\left( \frac{\lambda^{1 \wedge \frac{\kappa}{\nu} - \frac{1}{\nu}}}{N} \right), \tag{238}$$
$$\Im w(\lambda) = \pi \lambda^2 \mu_\lambda(\lambda) + O\left( \frac{\lambda^{1 - \frac{2}{\nu}}}{N} \right).$$

As an important application of the expressions above, let us estimate the scale of the off-diagonal part $\rho_{\text{off}}^{(2)}(\lambda_1, \lambda_2)$ of the learning measure $\rho^{(2)}(\lambda_1, \lambda_2)$ given in (16). Using (238) and (233) we get

$$\rho_{\text{off}}^{(2)}(\lambda_1, \lambda_2) = \frac{\lambda_2^{1-\frac{1}{\nu}} \lambda_1^{\frac{\kappa}{\nu}} \left( 1 + O\left( \frac{\lambda_2^{-\frac{1}{\nu}}}{N} \right) + O\left( \frac{\lambda_1^{-\frac{1}{\nu}}}{N} \right) \right) - \lambda_1^{1-\frac{1}{\nu}} \lambda_2^{\frac{\kappa}{\nu}} \left( 1 + O\left( \frac{\lambda_2^{-\frac{1}{\nu}}}{N} \right) + O\left( \frac{\lambda_1^{-\frac{1}{\nu}}}{N} \right) \right)}{\pi^2 \lambda_1 \lambda_2 N (\lambda_1 - \lambda_2)}. \tag{239}$$

Now we will estimate the scale $S[\rho_{\text{off}}^{(2)}]$ of $\rho_{\text{off}}^{(2)}(\lambda_1, \lambda_2)$ assuming that $\lambda_1$ and $\lambda_2$ have the scales $s_1$ and $s_2$ respectively. First, assume that $\lambda_1 - \lambda_2$ has the same scale as their maximum $\lambda_1 \vee \lambda_2$. Then, the corrections to the leading terms in the numerator can be neglected, and the scales of both denominator and numerator are given by minimal scale of two subtracted terms. Specifically,

$$S[\rho_{\text{off}}^{(2)}](s_1, s_2) = \left( \left( \frac{\nu-1}{\nu} s_1 + \frac{\kappa}{\nu} s_2 \right) \wedge \left( \frac{\kappa}{\nu} s_1 + \frac{\nu-1}{\nu} s_2 \right) \right) - (s_1 + s_2 - 1 + s_1 \wedge s_2)$$
$$\geq 1 - s_1 - s_2. \tag{240}$$

Yet, the scale derived above may not be valid when $\lambda_1$ and $\lambda_2$ are to close to each other. This does not happen, which can be seen, for example, by writing the difference in the enumerator of $\rho_{\text{off}}^{(2)}(\lambda_1, \lambda_2)$ and observing that the whole expression behaves regularly as $\lambda_1 \to \lambda_2$. Then, similar argumentation together with (235) shows that the scale derived in (201) holds when both $s_1 = s_2 = \nu$. Thus, for all $\lambda_1, \lambda_2$ inside the support of $\mu$ we have $S[\rho_{\text{off}}^{(2)}](s_1, s_2) \geq 1 - s_1 - s_2$.

Finally, we estimate the contribution of off-diagonal part of the second moment to the loss $L_{\text{off}}[h] = \frac{1}{2} \int h(\lambda_1) h(\lambda_2) \rho_{\text{off}}^{(2)}(d\lambda_1, d\lambda_2)$. For that, we can use a two-dimensional analog of proposition 2, which can be proven similarly.

**Proposition 6.** *Let $g_N(\lambda_1, \lambda_2)$ be a sequence of functions with a scaling profile $S^{(g)}(s_1, s_2)$, and let $a_N > 0$ be a sequence of scale $a > 0$. Then, the sequence of integrals $\int_{a_N}^1 \int_{a_N}^1 |g_N(\lambda_1, \lambda_2)| d\lambda_1 d\lambda_2$ has the scale*

$$S\Big[ \int_{a_N}^1 \int_{a_N}^1 |g_N(\lambda_1, \lambda_2)| d\lambda_1 d\lambda_2 \Big] = \min_{0 \leq s_1, s_2 \leq a} \big( S^{(g)}(s_1, s_2) + s_1 + s_2 \big). \tag{241}$$

Applying this statement to $L_{\text{off}}[h] = \frac{1}{2} \int h(\lambda_1) h(\lambda_2) \rho_{\text{off}}^{(2)}(d\lambda_1, d\lambda_2)$, we get

$$S\left[ L_{\text{off}} \right] \geq \min_{0 \leq s \leq \nu} \Big[ 1 - s_1 - s_2 + S^{(h)}(s_1) + S^{(h)}(s_2) + s_1 + s_2 \Big] \geq 1. \tag{242}$$

In other words, we have shown that

$$\int h(\lambda_1) h(\lambda_2) \rho_{\text{off}}^{(2)}(d\lambda_1, d\lambda_2) = O(N^{-1}). \tag{243}$$

Actually, the estimation above is tight, which can be shown, for example, by taking $h(\lambda) = 1$ computing the leading order term above, which will be exactly $\sim N^{-1}$. Note the presence of $N^{-1}$ term in the loss, regardless of the value of $\kappa, \nu$ is unnatural for our problem because we expect the rate of the optimal algorithm to be $N^{-\kappa \wedge 2\nu}$, which can be smaller than $N^{-1}$. The reason for this unnatural term can be traced back to our calculation of the resolvent second moment in (189) and (178), where in the application of the Wick theorem we took into account only the pairings producing leading order terms in $N$. Thus, we might expect that taking into account subleading order pairings in Wick theorem would lift the (243) from $O(N^{-1})$ to $O(N^{-2})$. What would be the role of the off-diagonal $\rho_{\text{off}}^{(2)}$ if we took into account all pairings (i.e. performed non-perturbative computation) is not clear at the moment and is an interesting direction for future work.

### E.3.3 NOISY OBSERVATIONS

For noisy observations, our goal is to show the equivalence between full loss functional and the NMNO functional (Theorem 2). In terms of the population densities $\mu_\lambda(\lambda)$ and $\mu_c(\lambda)$, the NMNO is written as

$$L^{(\text{nmno})}[h] = \frac{1}{2} \int_{N^{-\nu}}^1 \left[ \frac{\sigma^2}{N} h^2(\lambda) \mu_\lambda(\lambda) + \big(1 - h(\lambda)\big)^2 \mu_c(\lambda) \right] d\lambda. \tag{244}$$

Let us decompose the difference between two functionals as

$$
\begin{aligned}
L[h] - L^{(\text{nmno})}[h] = \frac{1}{2} \int_{\lambda_-}^1 &\left[ \frac{\sigma^2}{N} \Big( \frac{\Im w(\lambda)}{\pi \lambda^2} - \mu_\lambda(\lambda) \Big) h^2(\lambda) \right. \\
&+ \Big( \frac{|r^{-1}(\lambda)|^2 \Im v(\lambda)}{\pi \lambda^2} - \mu_c(\lambda) \Big) h^2(\lambda) \\
&\left. - 2\Big( \frac{\Im u(\lambda)}{\pi \lambda} - \mu_c(\lambda) \Big) h(\lambda) \right] d\lambda \\
+ \frac{1}{2} \int_{\lambda_-}^{N^{-\nu}} &\left[ \frac{\sigma^2}{N} h^2(\lambda) \mu_\lambda(\lambda) + \big(1 - h(\lambda)\big)^2 \mu_c(\lambda) \right] d\lambda + O(N^{-1}).
\end{aligned}
\tag{245}
$$

Now, similarly to the translation-invariant model on a circle, we write down the scales of all terms in the difference between two functionals, assuming that $\lambda$ has the scale $s$. For that, we use asymptotic expressions (238) for functions $v, u, w$, and also expression (233) for $r^{-1}(\lambda)$.

$$S\left[\frac{\sigma^2}{N}\left(\frac{\Im w(\lambda)}{\pi\lambda^2}-\mu_\lambda(\lambda)\right)h^2(\lambda)\right] \geq 1-s-\frac{s}{\nu}+(1-\frac{s}{\nu})+2S^{(h)}(s), \tag{246}$$

$$S\left[\left(\frac{|r^{-1}(\lambda)|^2\Im v(\lambda)}{\pi\lambda^2}-\mu_c(\lambda)\right)h^2(\lambda)\right] \geq \frac{\kappa}{\nu}s-s+(1-\frac{s}{\nu})+0\wedge(\frac{2\nu-\kappa}{\nu}s)+2S^{(h)}(s), \tag{247}$$

$$S\left[\left(\frac{\Im u(\lambda)}{\pi\lambda}-\mu_c(\lambda)\right)h(\lambda)\right] \geq \frac{\kappa}{\nu}s-s+(1-\frac{s}{\nu})+0\wedge(\frac{\nu-\kappa}{\nu}s)+S^{(h)}(s), \tag{248}$$

$$S\left[\frac{\sigma^2}{N}h^2(\lambda)\mu_\lambda(\lambda)\right] = 1-s-\frac{s}{\nu}+2S^{(h)}(s), \tag{249}$$

$$S\left[\left(1-h(\lambda)\right)^2\mu_c(\lambda)\right] = \frac{\kappa}{\nu}s-s+2S^{(1-h)}(s). \tag{250}$$

For the last two terms, we do it on the level of the whole integral, which is taken over a single scale $s=\nu$.

$$S\left[\int_{\lambda_-}^{N^{-\nu}}\frac{\sigma^2}{N}h^2(\lambda)\mu_\lambda(\lambda)d\lambda\right] = 2S^{(h)}(\nu), \tag{251}$$

$$S\left[\int_{\lambda_-}^{N^{-\nu}}\left(1-h(\lambda)\right)^2\mu_c(\lambda)d\lambda\right] = \kappa+2S^{(1-h)}(\nu). \tag{252}$$

Using the scales derived above, we need to obtain the conditions on the learning algorithm scales $S^{(1-h)}, S^{(h)}$ for which the scale of all corrections to the loss is greater than the scale of NMNO loss given, as usual, by

$$S\left[L^{(\text{nmno})}[h]\right] = \min_{0\leq s\leq\nu}\left[\left(\frac{\kappa}{\nu}s+2S^{(1-h)}\right)\wedge\left(1-\frac{s}{\nu}+2S^{(1-h)}(s)\right)\right] \leq \frac{\kappa}{\kappa+1}. \tag{253}$$

Again, this means that we can neglect all corrections whose scale (after integration with $d\lambda$) is greater than $\frac{\kappa}{\kappa+1}$. In particular, we can neglect $O(N^{-1})$ correction coming from off-diagonal part of the second moment (243).

First, note that the terms (251) and (252) exactly equal the contribution of noise (249) and signal (250) parts at scale $s=\nu$. Thus, if these terms are to be neglected, $s=\nu$ should not be a localization scale of $L^{(\text{nmno})}[h]$.

Second, observe that the right parts of $0\wedge(\frac{2\nu-\kappa}{\nu}s)$ and $0\wedge(\frac{\nu-\kappa}{\nu}s)$ in (247) and (248), when activated, give at most $O(N^{-1})$ contribution to the total loss, and therefore can also be neglected. Then, after choosing the left 0 option, we see that (248) is never less than (247). Thus, the total contribution of these two terms can be effectively described by the scale $\frac{\kappa}{\nu}s-s+(1-\frac{s}{\nu})+S^{(h)}(s)$.

Lastly, we can see that (246) is always larger than corresponding NMNO noise scale (249), except for $s=\nu$ where they are equal. But since we already require $s=\nu$ not being a localization scale of $L^{(\text{nmno})}[h]$, the term (246) can be neglected.

Thus, all requirements in addition to $s=\nu$ not being a localization scale of $L^{(\text{nmno})}[h]$ can be summarized as

$$\min_{0\leq s\leq\nu}\left[\frac{\kappa}{\nu}s+(1-\frac{s}{\nu})+S^{(h)}(s)\right] > \min_{0\leq s\leq\nu}\left[\left(\frac{\kappa}{\nu}s+2S^{(1-h)}\right)\wedge\left(1-\frac{s}{\nu}+2S^{(1-h)}(s)\right)\right]. \tag{254}$$

However, we have

$$\min_{0\leq s\leq\nu}\left[\frac{\kappa}{\nu}s+(1-\frac{s}{\nu})+S^{(h)}(s)\right] \geq 1\wedge\kappa > \frac{\kappa}{\kappa+1}, \tag{255}$$

therefore, the requirement (254) is satisfied automatically. Summarizing, the corrections to the NMNO functional are asymptotically vanishing when $s=\nu$ is not a localization scale of $L^{(\text{nmno})}[h]$.

### E.3.4 NOISELESS OBSERVATIONS

Now, we take $\sigma^2 = 0$ and analyze the respective loss functional. Recall that our current calculations are accurate up to $O(N^{-1})$, as we discussed before. Therefore, we will neglect all the terms that give contributions of the order $O(N^{-1})$, since they are beyond our current level of accuracy. This means several things:

1. We ignore the contribution to the loss of the off-diagonal part $\int h(\lambda_1) h(\lambda_2) \rho_{\text{off}}^{(2)}(d\lambda_1, d\lambda_2)$ of the functional.

2. We ignore the $O(\frac{\lambda^{1-\frac{1}{\nu}}}{N})$ correction to $\Im v(\lambda)$ and $\Im u(\lambda)$ in (238): in the previous section we have shown that they have at most $O(N^{-1})$ contribution to the loss.

3. We restrict ourselves with the values of signal exponent $\kappa < 1$. Since the optimal scaling of the loss in the noiseless case is expected to be $N^{-\kappa}$, we will not be able to capture it for $\kappa \geq 1$.

4. The previous point also implies that we cannot access values $\kappa > 2\nu > 2$ corresponding to the noiseless saturation phase since eigenvalue exponent values are confined to the physical region $\nu > 1$.

With the above remarks taken into account, we start deriving the loss functional and, for compactness, ignore all the $O(N^{-1})$ terms. First, due to the diagonality of the remaining part of the learning measure $\rho^{(2)}(d\lambda_1, d\lambda_2)$, we can immediately specify the optimal algorithm and the respective decomposition of the functional

$$L[h] = \frac{1}{2} \int_{\lambda_-}^{1} \left[ \frac{|r^{-1}(\lambda)|^2 \Im v(\lambda)}{\pi \lambda^2} \left( h(\lambda) - h^*(\lambda) \right)^2 + \left( \mu_c(\lambda) - \frac{\left( \Im u(\lambda) \right)^2}{\pi |r^{-1}(\lambda)|^2 \Im v(\lambda)} \right) \right] d\lambda, \quad (256)$$

$$h^*(\lambda) = \frac{\lambda \Im u(\lambda)}{|r^{-1}(\lambda)|^2 \Im v(\lambda)} = 1 + O(\frac{\lambda^{-\frac{1}{\nu}}}{N}), \quad (257)$$

where we have used asymptotics (238) to estimate the deviation of the optimal algorithm from $h(\lambda) = 1$. Again using (238), we see that the free term in (256) has the order $O(\frac{\lambda^{\frac{\kappa-1}{\nu}-1}}{N})$, which, given $\kappa < 1$, always localizes on the maximal scale $s = \nu$. If we additionally assume that the learning algorithms fits higher eigenvalues well enough: $|1 - h(\lambda)| = O(\sqrt{\frac{\lambda^{-\frac{1}{\nu}}}{N}})$, the first term in (256) will also always localize on the maximal scale $s = \nu$.

Now, relying on the fact that the loss localizes on $s = \nu$, we can use expressions of all the terms in the functional through the phase $\phi$. Eigenvalue $\lambda$ is defined by an explicit form of (227)

$$\lambda = N^{-\nu} \left( \frac{\sin \phi}{C_\nu \sin(\phi - \frac{\phi}{\nu})} \right)^{-\nu} \sin \phi \left( \cot(\frac{\nu-1}{\nu}\phi) - \cot \phi \right). \quad (258)$$

Using (235) and (229), the terms of the functional become

$$\frac{|r^{-1}(\lambda)|^2 \Im v(\lambda)}{\pi \lambda^2} = \frac{1}{\nu} \left( \frac{\sin \phi}{C_\nu \sin(\phi - \frac{\phi}{\nu})} \right)^{-\kappa+\nu} \frac{\sin(\phi - \frac{\kappa}{\nu}\phi)}{\sin(\frac{\kappa}{\nu}\pi) \sin^2 \phi \left( \cot(\frac{\nu-1}{\nu}\phi) - \cot \phi \right)^2} \quad (259)$$

$$\mu_c(\lambda) = \frac{1}{\nu} \left( \frac{\sin \phi}{C_\nu \sin(\phi - \frac{\phi}{\nu})} \right)^{-\kappa+\nu} \left( \sin \phi \left( \cot(\frac{\nu-1}{\nu}\phi) - \cot \phi \right) \right)^{\frac{\kappa}{\nu}-1} \quad (260)$$

$$\frac{\left( \Im u(\lambda) \right)^2}{\pi |r^{-1}(\lambda)|^2 \Im v(\lambda)} = \frac{1}{\nu} \left( \frac{\sin \phi}{C_\nu \sin(\phi - \frac{\phi}{\nu})} \right)^{-\kappa+\nu} \frac{\sin^2(\frac{\kappa}{\nu}\phi)}{\sin(\phi - \frac{\kappa}{\nu}\phi) \sin(\frac{\kappa}{\nu}\pi)}, \quad (261)$$

The optimal algorithm is

$$h^*(\lambda) = \frac{\cot(\frac{\nu-1}{\nu}\phi) - \cot \phi}{\cot(\frac{\kappa}{\nu}\phi) - \cot \phi}. \quad (262)$$

Substituting everything in the loss functional gives

$$
L[h] = \frac{N^{-\kappa}}{2} \int\limits_0^{\pi - \frac{\pi}{\nu N}} \left( \frac{\sin \phi}{C_\nu \sin(\phi - \frac{\phi}{\nu})} \right)^{-\kappa + \nu} \left[ \frac{\sin(\phi - \frac{\kappa}{\nu}\phi)\big(h(\lambda) - h^*(\lambda)\big)^2}{\nu \sin(\frac{\kappa}{\nu}\pi) \sin^2 \phi \big( \cot(\frac{\nu - 1}{\nu}\phi) - \cot \phi \big)^2} \right.
$$
$$
\left. + \frac{1}{\nu} \Big( \sin \phi \big( \cot(\tfrac{\nu - 1}{\nu}\phi) - \cot \phi \big) \Big)^{\frac{\kappa}{\nu} - 1} - \frac{\sin^2(\frac{\kappa}{\nu}\phi)}{\nu \sin(\phi - \frac{\kappa}{\nu}\phi) \sin(\frac{\kappa}{\nu}\pi)} \right] \frac{d(N^\nu \lambda)}{d\phi} d\phi, \tag{263}
$$

Finally, we note that the $\phi \to \pi$ asymptotic of the expressions under the integral above can be inferred from the respective $\frac{\lambda^{-\frac{1}{\nu}}}{N} \to 0$ asymptotic of the original functional, since $\pi - \phi = \frac{\pi}{\nu} \frac{\lambda^{-\frac{1}{\nu}}}{N}(1 + o(\pi - \phi))$. Thus, we can write

$$
L[h] = \frac{N^{-\kappa}}{2} \int\limits_0^{\pi - \frac{\pi}{\nu N}} \left[ O\Big( (\pi - \phi)^{-\kappa + \nu} \Big) \big( h(\lambda_\phi) - h^*(\lambda_\phi) \big)^2 + O\Big( (\pi - \phi)^{-\kappa + \nu + 1} \Big) \right] d\phi. \tag{264}
$$

From the above expression we see that, if $|h(\lambda_\phi) - h^*(\lambda_\phi)| = O(|h(\lambda_\phi) - 1|) = O(\sqrt{\pi - \phi})$, the expression under the integral is integrable at $\phi = \pi$. Therefore, the upper integration limit can be shifted to $\pi$ without changing leading order asymptotic of the loss.

**Overlearning transition.** As we discussed above, our analysis of Wishart model in the noiseless case is restricted to the values of target exponent $\kappa < 1$ due to $O(N^{-1})$ accuracy of calculations. Thus, if $\nu < 2$, we can access overlearning transition point $\kappa = \nu - 1 < 1$ and verify whether the transition holds for the Wishart model. Proceeding similarly to respective Lemma 2 for the circle model, we have

**Lemma 3.** *Consider the optimal algorithm* (262)*:*

$$
h^*(\lambda) = \frac{\cot(\frac{\nu - 1}{\nu}\phi) - \cot \phi}{\cot(\frac{\kappa}{\nu}\phi) - \cot \phi} \tag{265}
$$

*with eigenvalue $\lambda = \lambda(\phi), \phi \in (0, \pi)$ parameterized according to* (258)*. Then, assuming $\kappa < 1$, for any $\phi \in (0, \pi)$*

$$
h^*(\lambda) \begin{cases} < 1, & \kappa + 1 < \nu, \\ = 1, & \kappa + 1 = \nu, \\ > 1, & \kappa + 1 > \nu. \end{cases} \tag{266}
$$

*Proof.* Observe that both the nominator and denominator of (265) have the form $g(a, \phi) \equiv \cot(a\phi) - \cot(\phi)$ with $a \in (0, 1)$, $\phi \in (0, \pi)$. Since $\cot(\phi)$ is strictly decreasing on $(0, \pi)$, the function $g(a, \phi)$ is 1) positive 2) at fixed $\phi$ is strictly decreasing in $a$. Thus, for the ratio of such functions, we have

$$
\frac{g(a, \phi)}{g(b, \phi)} \begin{cases} < 1, & a > b, \\ = 1, & a = b, \\ > 1, & a < b. \end{cases} \tag{267}
$$

The above is equivalent to the statement of the lemma once $a = \frac{\nu - 1}{\nu}$, $b = \frac{\kappa}{\nu}$. $\qquad \square$

The above result shows that the behavior of the sign of $h(\lambda) - 1$, which indicates overlearning/underlearning, is exactly the same in the Wishart and Circle models. This makes us believe that overlearning transition can be a quite general phenomenon going beyond two data models considered in the current work.

## F  EXPERIMENTS

### F.1  FIGURE 1: DETAILS AND DISCUSSION.

Let us start by describing the experiment setting and details. Both KRR and GF plots use optimally scaled regularization $\eta$ and time $t$, as derived in Section C. For all three data models, we consider

ideal power-law population spectrum: $\lambda_l = l^{-\nu}$, $c_l^2 = l^{-\kappa-1}$ (truncated at $P = 4 \times 10^4$ due to computational limitations), and an adapted version $\lambda_l = (2(|l|+1))^{-\nu}$, $|c_l|^2 = (2(|l|+1))^{-\kappa-1}$, $l \in \mathbb{Z}$ for Circle model. The extra factor of 2 here is needed to asymptotically align population spectrum at small $\lambda$: for all three models we have $\mu_c([0,\lambda]) \to \frac{1}{\kappa}\lambda^{\frac{\kappa}{\nu}}$ and $\mu_\lambda([\lambda,\infty)) \to \lambda^{-\frac{1}{\nu}}$ as $\lambda \to 0$.

The figure contains 3 types of data that are computed in different ways. The first type is scatter plot markers and corresponds to the estimation of generalization loss via direct simulation. For Wishart and Cosine Wishart (see Section F.3) models, this amounts to sampling empirical kernel matrix $\mathbf{K}$ and observation vector $\mathbf{y}$, calculating the generalization error for the resulting sampled realization, and finally averaging the result over $n = 100$ repetitions of the above procedure to estimate the expectation over training dataset $\mathcal{D}_N$ in (3). Due to computational limitations, we were able to execute this procedure for sizes of empirical kernel matrix only up to $N = 4 \times 10^3$. For Circle model, Theorem 1 gives an exact value of the expected loss (3) (i.e. no approximations are made during the derivation). Since for the considered type of population spectra the N-perturbations (10) can be expressed in terms of Hurwitz zeta function (see (127)), we compute analytically all the terms in (11). This way, we are able to reach much larger values of $N$ for the circle model.

Let us also comment on the estimation of the generalization error (3) of a given realization of Wishart or Cosine Wishart models. The expectation over $\mathbf{x}$ requires scalar products

$$\langle K(\mathbf{x}_i, \mathbf{x}), K(\mathbf{x}, \mathbf{x}_j)\rangle = \sum_l \lambda_l^2 \phi_l(\mathbf{x}_i)\phi_l(\mathbf{x}_j), \qquad \langle f^*(\mathbf{x}), K(\mathbf{x}, \mathbf{x}_j)\rangle = \sum_l \lambda_l c_l \phi_l(\mathbf{x}_j) \quad (268)$$

for points $\mathbf{x}_i, \mathbf{x}_j$ from sampled dataset $\mathcal{D}_N$. The expressions above can be used to calculate these scalar products experimentally since we have access to $\lambda_l, c_l$ and already sampled feature values $\phi_l(\mathbf{x}_i)$.

The second type of data is depicted with solid lines and corresponds to the direct calculation of NMNO loss (17), which becomes a sum for discrete population spectrum. Assuming $\lambda_l$ are sorted in descending order, NMNO loss becomes

$$L^{(\mathrm{nmno})}[h] = \frac{1}{2}\sum_{l=1}^{N}\left[c_l^2(1 - h(\lambda_l))^2 + \frac{\sigma^2}{N}h(\lambda_l)^2\right], \quad (269)$$

which we can easily compute for quite large values of $N$.

Finally, the third type of data is $N \to \infty$ loss asymptotic $L = CN^{-\#}(1 + o(1))$ depicted with dashed lines. The constant $C$ we compute analytically, similarly to limiting expressions (138), (135) for noiseless Circle model and (263) for Wishart model. For that, we 1) recall (see Section C) the loss localization scale $s_{\mathrm{loc}}$ for the considered algorithms: $s_{\mathrm{loc}} = \frac{\nu}{\kappa+1}$ for GD and non-saturated KRR, and $s_{\mathrm{loc}} \in \{0, \frac{\nu}{2\nu+1}\}$ for saturated KRR 2) and then get the limiting expressions of NMNO loss functional (17).

For GF, we have algorithm profile $h_t(\lambda) = 1 - e^{-t\lambda}$ and loss localization scale is $s_{\mathrm{loc}} = \frac{\nu}{\kappa+1}$. Thus, we make a change of variables $\lambda = \lambda' N^{-\frac{\nu}{\kappa+1}}$ and $t = t' N^{\frac{\nu}{\kappa+1}}$, leading to the following limiting loss

$$L^{(\mathrm{nmno})}[h_t] = \frac{1}{2}N^{-\frac{\kappa}{\kappa+1}}\frac{1}{\nu}\int_0^\infty \left[e^{-2t'\lambda'}(\lambda')^{\frac{\kappa}{\nu}} + \sigma^2(1 - e^{-t'\lambda'})^2(\lambda')^{-\frac{1}{\nu}}\right]\frac{d\lambda'}{\lambda'}. \quad (270)$$

Note that the integral above converges both at $\lambda' \to 0$ and $\lambda' \to \infty$, which reflects that our change of variables has used the correct loss localization scale. The integral in (270) can be computed either numerically or analytically by reducing it to Gamma functions.

For KRR algorithm profile is $h_\eta(\lambda) = \frac{\lambda}{\lambda+\eta}$ and loss localization changes between saturated and non-saturated phases. In the non-saturated phase, we make a change of variables $\lambda = \lambda' N^{-\frac{\nu}{\kappa+1}}$ and $\eta = \eta' N^{-\frac{\nu}{\kappa+1}}$, leading to

$$L^{(\mathrm{nmno})}[h_\eta] = \frac{1}{2}N^{-\frac{\kappa}{\kappa+1}}\frac{1}{\nu}\int_0^\infty \frac{(\eta')^2(\lambda')^{\frac{\kappa}{\nu}} + \sigma^2(\lambda')^{2-\frac{1}{\nu}}}{(\lambda'+\eta')^2}\frac{d\lambda'}{\lambda'}, \quad \kappa < 2\nu. \quad (271)$$

In the saturated phase, the noise part localize at $s_{\mathrm{loc}} = \frac{\nu}{2\nu+1}$, which coincides with the scale of regularization $\eta = \eta' N^{-\frac{\nu}{2\nu+1}}$. The signal part localizes at $s_{\mathrm{loc}} = 0$ where the loss functional

remains discrete, and we can use a perturbative expression for the learning algorithm $h_\eta(\lambda) \approx \frac{\lambda}{\eta}$. This leads to

$$L^{(\text{nmno})}[h_\eta] = \frac{1}{2} N^{-\frac{2\nu}{2\nu+1}} \left[ \frac{1}{\nu} \int_0^\infty \sigma^2 \frac{(\lambda')^{2-\frac{1}{\nu}}}{(\lambda'+\eta')^2} \frac{d\lambda'}{\lambda'} + (\eta')^2 \sum_l \frac{|c_l|^2}{\lambda_l^2} \right], \quad \kappa > 2\nu. \quad (272)$$

As for the GF, the integrals in (271) and (272) can be computed either numerically or analytically by reducing them to Beta function integrals with a change of integration variable $z = \frac{\eta'}{\lambda'+\eta'}$.

**Discussion.** The main conclusion of the figure is, indeed, the match between the actual values of the loss for a given model (scatter markers) and NMNO values (solid lines) for large enough $N$. This validates experimentally the statement of Theorem 2. Importantly, the Cosine Wishart model is not covered by our theory but still demonstrates the equivalence. We interpret it as an indication that the equivalence holds for a broader class of models. At the moment, we are not ready to give any potential candidates for such class of models since it requires a different set of tools than the one used in this work.

Observe that the match between the actual loss of a data model and its NMNO analog happens at quite small $N$ for big value of target exponent $\kappa = 5$ (two right subfigures) whereas much larger $N$ are required for small $\kappa = 0.5$. This is a general manifestation of the fact that slower power-laws require much larger sizes $N$ for asymptotic $N \to \infty$ behavior to start working. For instance, NMNO loss (17) ignores the error associated with unlearned signal at eigenvalues $\lambda < \lambda_{\min}$. The contribution of this part can be roughly estimated as $\mu_c([0, \lambda_N])/\mu_c([0, 1]) \approx N^{-\kappa} = e^{-\log N \kappa}$ which becomes negligible at exponentially large values of dataset size $N \gg e^{\frac{1}{\kappa}}$.

Finally, we note that only the third subplot of the figure (corresponding to saturated KRR) has different loss asymptotics for the Circle model on one side and Wishart and Cosine Wishart on the other side. This difference is due to the respective population spectra which, although matching asymptotically at $\lambda \to 0$, are different at the head of the spectrum $\lambda \sim 1$. Then, since the loss for saturated KRR localizes at the scale $s = 0$ (i.e. $\lambda \sim 1$) as demonstrated in Figure 2, the difference between population spectra at $\lambda \sim 1$ start to affect the loss values. On the contrary, when there is no saturation (the rest of the plots on the figure), the loss localizes on some scale $s > 0$ where the two population spectra become (asymptotically) the same. This is the reason why 1,2,4 subplots have a single common loss asymptotic colored in grey.

### F.2 FIGURE 3: DETAILS AND DISCUSSION.

This figure's primary focus is the comparison of different algorithms. Profiles of each of 4 considered algorithms were obtained as follows: interpolation is simply $h(\lambda) = 1$, the optimal algorithm was calculated from (23), optimally stopped GF was obtained by first evaluating the asymptotic loss (138) on a wide and dense grid of $t$ values subsequently picking $t^*$ as the optimal among those values, and for optimally regularized KRR the same procedure (including both negative and positive regularization values) was used.

On the first plot, we validate the asymptotic loss value $L^{(\text{asym})} = CN^{-\kappa}$ (dashed lines) computed from (138) with exact loss value at finite $N$ given by (11) (scatter markers) computed similar to the respective values from Figure 1. On the remaining 3 plots we already exclusively use asymptotic expression (138).

**Discussion.** The first plot indeed confirms that asymptotic expression $L^{(\text{asym})} = CN^{-\kappa}$ given by (138) can accurately describe the loss values with the correct constant $C$.

The second plot examines the behavior of the constant $C$ across the range of target exponent values $\kappa \in (0, 2\nu)$, while saturated values $\kappa > 2\nu$ have different rate $N^{-2\nu}$ and thus not considered on the figure. In particular, in the vicinity of the saturation transition the constant starts to blow up $C \to \infty$ as $\kappa \to 2\nu - 0$. The overlearning transition at $\kappa = \nu - 1$ is also quite visible: for $\kappa < \nu - 1$ all the considered algorithms have very close loss values, while for $\kappa > \nu - 1$ a significant gap appears between optimal KRR and the optimal algorithm on one side and interpolation and optimal GF on the other side. This demonstrates that significant overlearning $h(\lambda) > 1$ is required for strong

performance. Note that GF profile $h_t(\lambda) = 1 - e^{-t\lambda} \leq 1$ which forces $t^* = \infty$ for $\kappa > \nu - 1$, thus explaining exact match between green and grey lines for $\kappa < \nu - 1$.

Actually, it is quite difficult to distinguish loss curves of different algorithms (except for the over-learning gap discussed above) on the two left plots. However, there is a well-defined (i.e. not due to numerical errors) difference between the algorithms (high zoom is required to see it!). Such a small difference seems to be an intrinsic property of the Circle model: we have tried wide ranges of $\kappa$ and $\nu$ and observed that in all of them, the relative difference between different algorithms was of the order of $1\%$ or smaller.

### F.3   COSINE WISHART MODEL

The model is constructed as follows. Given a population spectrum $\lambda_l, c_l \in \mathbb{R}, \ l = 0, 1, 2, 3, \ldots,$ the features $\phi_l(x)$ with $x \in \mathcal{S}$ are defined as

$$\phi_l(x) = \begin{cases} 1, & l = 0, \\ \sqrt{2}\cos(lx), & l \geq 1, \end{cases} \tag{273}$$

and the training dataset inputs $x_i \in \mathcal{D}_N$ are sampled i.i.d. from a uniform distribution on the circle $\mathcal{S}$. In particular, (273) ensures that $\mathbb{E}_x[\phi_l(x)\phi_{l'}(x)] = \delta_{ll'}$.

The Cosine Wishart model can be thought as something intermediate between our main Wishart and Circle models. On the one side, the population quantities of Cosine Wishart models are the same as those for the Circle given in (6), except for the presence of $e^{il(x+x')}$ terms in the kernel $K(x, x')$ making it non-translation invariant. This last aspect does not move Cosine Wishart model too much from Circle, as the respective empirical kernel matrix $\mathbf{K}$ would still be almost diagonalized by discrete Fourier harmonics $e^{i\frac{2\pi ki}{N}}$ if the inputs were forming a regular lattice. The more important difference is that inputs $x_i$ of the Cosine Wishart model are sampled i.i.d., which completely eliminates the possibility to analytically diagonalize $\mathbf{K}$, which is a core step of the Circle model solution.

On the other side, for a given population spectrum $\lambda_l, c_l$, the structure of Cosine Wishart model and Wishart model are similar in the sense that in both cases, random realizations of empirical kernel matrix $\mathbf{K}$ are obtained by sampling components of the feature matrix $\mathbf{\Phi}$ with first to moments being $\mathbb{E}\Phi_{li} = 0$ and $\mathbb{E}\Phi_{li}^2 = 1$, except for $\mathbb{E}\Phi_{0i} = 1$ for Cosine Wishart which only amounts to a single spike in $\mathbf{K}$. However, the most important difference is that for Wishart model, all entries of $\mathbf{\Phi}$ are independent, while for Cosine Wishart there is a very strong correlation of entries $\Phi_{li} = \sqrt{2}\cos(lx_i)$ with different $l$ but the same $i$. This correlation turns out to be crucial and makes the Stieltjes transform $r(z) = \mathrm{Tr}\left[(\mathbf{K} - z\mathbf{I})^{-1}\right]$ no longer deterministic, which was the core assumption for our analysis of Wishart model based on the fixed-point equation (13).

