# OpenReview forum: "Generalization error of spectral algorithms"
_ICLR.cc/2024/Conference — ICLR 2024 spotlight_

### Official Review · Reviewer_wTWc · 2023-10-19

**Soundness:** 3 good
**Presentation:** 3 good
**Contribution:** 3 good
**Rating:** 8
**Confidence:** 3

**Summary:**

This paper considers a general framework that expresses the generalization error in spectral algorithms. A generic formula is derived for an arbitrary choice of kernel and profile. Then, the analysis is specialized to account for the circle model and the Wishart model. The theory can explain the KRR saturation phenomenon. Finally, a simplified naive model for noisy observations (NMNO) was proposed, for which the generalization error can be neatly expressed, and it is shown that under certain conditions, the circle model and the Wishart model collapse into NMNO.

**Strengths:**

* The paper takes a generic point of view. It seems that the result can be applied to many different situations, which improves the power of the result.
* The theory is nicely applied to provide a new perspective to already observed phenomena.
* Overall, the paper is clearly written and most results are motivated.

**Weaknesses:**

* The analysis in the paper does not account for the errors induced by discretization. For instance, although it is claimed at the beginning of the paper that the theory covers gradient descent, what is really done is for gradient flow. Also, when the kernel $K$ is discretized using the samples, the following assumptions are all made about this discrete kernel, and it is not obvious how it can be connected to the continuous kernel.
* Certain parts of the analysis do not obtain nice closed formulas. For example, those in `section 4` for the Wishart model and those in `section 5` for the noise-free case.
* Although this is a theoretical paper, it would be helpful to include some experiments, even if they are on synthetic datasets. This helps explain (and justify) the theory.

**Questions:**

* On page 3, when you define $\boldsymbol{\Lambda}$, what is the number $P$? Is it the exact rank of $\mathbf{K}$ or its numerical rank, or it does not matter too much?
* In the statement of Proposition 1, I do not think $\rho$ has been defined before and I do not think it is a standard notation outside this community. What is the precise definition of it?
* Can you make your conjecture on page 6 slightly more formal?
* It's not a question, but I think your discussion of the circle model in the GF case might be related to this line of work below. Did you happen to come across these works and see some potentially interesting connections to your paper?
    * Sanjeev Arora, Simon Du, Wei Hu, Zhiyuan Li, and Ruosong Wang, Fine-grained analysis of optimization and generalization for overparameterized two-layer neural networks, International Conference on Machine Learning, pp. 322-332, PMLR, 2019.
    * Ronen Basri, David Jacobs, Yoni Kasten, and Shira Kritchman, The convergence rate of neural networks for learned functions of different frequencies, Neural Information Processing Systems 32, 2019
    * Annan Yu, Yunan Yang, and Alex Townsend, Tuning frequency bias in neural network training with nonuniform data, International Conference on Learning Representations, 2023

---

> ### Author Response · Authors · 2023-11-23
>
> Thank you for your valuable comments and questions and for the overall positive evaluation of our work! Let us address your question separately below
>
> > The analysis in the paper does not account for the errors induced by discretization. For instance, although it is claimed at the beginning of the paper that the theory covers gradient descent, what is really done is for gradient flow. Also, when the kernel
>  is discretized using the samples, the following assumptions are all made about this discrete kernel, and it is not obvious how it can be connected to the continuous kernel.
>
> Regarding the first discretization of Gradient Flow (GF) to Gradient Descent (GD), you are right that in the paper, we demonstrate results only for GF. To address this, in the revised version of the paper we double-checked that all claims including gradient-based algorithms are made about GF and not GD. However, are loss functionals derived for Wishart and circle model do not assume any specific shape of the profile, and therefore GD profile $h_t(\lambda)=1 - (1-\alpha \lambda)^t$ can be substituted in the functional to characterize GD generalization. We focused on GF due to its simplicity, and because we do not expect significant differences between GD and GF results in the setting of our paper(but we acknowledge the significant difference in other settings, e.g. for non-linear optimization or when convergence-divergence questions are involved).
>
> Regarding the second discretization of the kernel using samples, I believe that we describe jointly the properties of the continuous kernel via mentioning population spectrum $\lambda_l$ and features $\phi_l(\mathbf{x})$, and discrete empirical kernel matrix $\mathbf{K}$ by mentioning how the training inputs $\mathbf{x}_i$ are generated. For Circle model this is done explicitly. For Wishart model, this is done rather implicitly by the statistics of $\phi_l(\mathbf{x}_i)$ without defining separately features $\phi_l(\mathbf{x})$ and distribution of inputs $\mathbf{x}_i$. Yet, such implicit characterization is sufficient for generalization error in eq. (3) to be well defined, and we can even access it experimentally as we describe in a new appendix section G.1.
>
> > Although this is a theoretical paper, it would be helpful to include some experiments, even if they are on synthetic datasets. This helps explain (and justify) the theory.
>
> That is an excellent comment! During the rebuttal time, we have primarily focused on performing numerical experiments validating our theoretical results. Please see our general response for more details and the revised version of the paper for the new experiment results.
>
> > On page 3, when you define $\mathbf{\Lambda}$, what is the number $P$? Is it the exact rank of  $\mathbf{K}$ or its numerical rank, or it does not matter too much?
>
> Throughout the paper, we use both finite and infinite values $P$. For example, all the theoretical calculations use the value $P=\infty$, while in the numerical experiments we take large enough but finite $P$ to stay close to the theoretical setting. While accurate and rigorous treatment of $P=\infty$ requires specific settings for it to be well defined, in our theoretical analysis we follow the "physical level of rigor", prioritizing shorter and more accessible derivations in favor of еру accurate mathematical definitions of all the objects involved.
>
> > In the statement of Proposition 1, I do not think $\rho$ has been defined before and I do not think it is a standard notation outside this community. What is the precise definition of it?
>
> Thank you for mentioning this! Indeed, the version of proposition 1 in the submission was not clear, leaving some ambiguity in the status of measures $\rho$. Conceptually, we believe that these learning measures are new objects introduced in our work (which are essentially equivalent to the loss functional itself). In the revised version of the paper, we improved the wording of proposition 1, including the reference to a general expression of these measures through the expectations over the training dataset $\mathcal{D}_N$.
>
> > Can you make your conjecture on page 6 slightly more formal?
>
> Indeed, that would be a great addition to the work. However, at the moment we feel we are not ready to formulate a rigorous version yet. This would require defining a family of kernel/data models and providing the analysis able to cover all models within the family. It is difficult to do at the moment since our current approach relies on explicit derivation of generalization error, with techniques tailored towards to data models we considered. Intuitively, we expect the condition of Theorem 2 regarding the absence of localization on the largest scale $s=\nu$ to be the principal requirement for the model equivalence.

---

> ### Author Response · Authors · 2023-11-23
> **Continuation of the response**
>
> > It's not a question, but I think your discussion of the circle model in the GF case might be related to this line of work below. Did you happen to come across these works and see some potentially interesting connections to your paper?
>
> Thank you for pointing us to these important works. While we have encountered papers [1,2] before we were not aware of the work [3]. Generally, there is a significant connection in the sense that for both works (ours and [1,2,3]) the characterization of the spectrum of the underlying kernel method plays the central role in the analysis. Moreover, [2] also focuses on the Fourier analysis on circle and sphere, which is also at the heart of our Circle model. On the other hand, there seem to be several important differences in our approaches and results:
> 1. We aim to develop explicit expressions for the risk, and not upper bounds as is mostly done in the mentioned papers.
> 2. We do not require spectral gap assumptions as our main setting of interest is the power-law distribution of the eigenvalue which does not have a gap.
> 3. We do not limit our analysis to particular types of kernel. For example, papers [1,2] focus on the NTK of infinitely wide two-layer ReLU network (denoted as $\mathbf{H}^\infty$). The work [2] shows that such kernel has the spectrum behaving as $\lambda_k \sim k^{-2}$ as $k\to\infty$, which roughly corresponds to fixing $\nu=2$ in the setting of our paper. We chose to consider population spectrum $\lambda_l,c_l$ as free variables so that we can access a wide range of values of $\nu,\kappa$, which makes it possible to discuss effects like saturation transition at $\kappa=2\nu$ and overlearning transition at $\kappa=\nu-1$.
>
> It would be nice to do a deeper comparison of the results of our work and papers [1-3] under the conditions on the model and data that make jointly valid the results of both these papers and our paper so that the results can be compared directly. We leave it for future work.

---

### Official Review · Reviewer_jkPS · 2023-10-19

**Soundness:** 4 excellent
**Presentation:** 3 good
**Contribution:** 4 excellent
**Rating:** 8
**Confidence:** 3

**Summary:**

This paper defines the spectral algorithms for regression task with Kernel Ridge Regression (KRR) and Gradient Descent (GD) as its special case. Then they derive and compute the generalisation error as a functional of the learning profile $h(\lambda)$ where $\lambda>0$ is the ridge and $h$ a function acting point-wisely on diagonal matrices.

Then the paper assumes the two data models: high-dimensional Gaussian and low-dimensional translation-invariant model to further derive and interpret the generalisation error, recovering the result from previous study, as well as explaining some phenomenons in a new perspective, for example the saturation effect of KRR.

**Strengths:**

Originality: The idea of spectral algorithm is novel and can be seen as unifying different common machine learning methods like KRR and GD.

Quality: The paper contains both theoretical deductions and experimental validations. The citation is also very up-to-date.

Clarity: The paper is organised in a clear manner by mentioning the motivation at first and postponing the technical details in the appendix.

Significance: Although the computation limit on the two simple models, this paper is a first step into understanding the generalisation error of spectral algorithm in a unifying way.

**Weaknesses:**

There is almost no weakness except a few typos, for example, on section 4 EXPLICIT FORMS OF THE LOSS FUNCTIONAL (Circle model), on the fourth line it should
$$
\hat{\lambda}\_k = \sum\_{n=-\infty}^\infty \lambda\_{k+Nn}.
$$

Also, due to the complex form of the generalisation error, only two simple data models are considered.

**Questions:**

I really like the idea of the generalisation error of spectral algorithm. What are the biggest obstacle to generalise the argument of this paper to more kernels?

---

> ### Author Response · Authors · 2023-11-23
> **Response by authors**
>
> Thank you very much for emphasizing the strong sides of our work, and we are glad you found the ideas of the paper interesting.
>
> Also, thank you for pointing out the typo, which we might have missed otherwise.
>
> **Extension to more general kernel/data models.** Our approach in this paper is based on explicitly computing the generalization error (or its leading term in the $\tfrac{1}{N}$-expansion in the case of Wishart model). One relatively straightforward generalization would be the high-dimensional version of the circle model (i.e. a grid on a torus). Besides that, we are currently unaware of any other models that could potentially admit an analytical solution on the same level of detail as the two currently considered models. One potentially interesting direction is related to symmetries: one may argue that analytical solutions for our two data models is due to high level of symmetry in both of them. Then, one can wonder whether there are some other symmetries that would allow for analytical characterization of the loss functional of the underlying model.
>
> However, if one is concerned only about the equivalence result of data models and NMNO, more options for generalization potentially appear. The equivalence is mainly based on the argument that at empirical eigenvalues far away from the tail (i.e. on the scale $s<\nu$), the empirical quantities are close to the population ones with the difference between the two being sufficiently small. Thus an upper bound on this difference will suffice and the exact value is not required. We hope that it is possible to provide such an upper bound for a broader class of models, but this would require a very different set of technical tools than those we used in the current work.

---

### Official Review · Reviewer_v7ue · 2023-10-26

**Soundness:** 3 good
**Presentation:** 3 good
**Contribution:** 3 good
**Rating:** 8
**Confidence:** 2

**Summary:**

This paper develops optimization algorithms for KRR estimators using a new meta function termed ``spectral profile'', and shows its versatility to generalize to settings beyond KRR.

**Strengths:**

* Equation (7) in introducing the profile $h(\lambda)$ seems new and interesting.

* The theory appears to be rich, reasonable and self-contained.

**Weaknesses:**

* The paper may seem a bit hard to follow at the beginning due to certain missing definitions (say, $\nu$ in equation (2)), which need further context derived from reading Section 2.

**Questions:**

* Is Figure 2 mentioned in the context? I'm kind of missing what those curves represent here.

---

> ### Author Response · Authors · 2023-11-23
> **Response by Authors**
>
> Thank you for the positive evaluation of our work!
>
> We agree that our introduction section has some things that are properly defined only in the next setting section. While we are also concerned about this issue, we intentionally decided to sacrifice some sequentiality of exposition in favor of a shorter informal discussion covering the main points.
>
> Thank you for your question about Figure 2. While there was a reference to it in the text, your concern is indeed valid and touches a bigger problem of Section 5.3. being too compressed and, therefore, hard to understand. We tried to improve this point in the revised version of the paper - please see our general response for more details.

---

### Public Comment · ~Daniel_Beaglehole1 · 2023-11-18
**Relevant work**

Dear authors,

In prior work, we have also used the circle model (and in higher dimensions, the torus) to derive exact expressions for the test error of kernel regression with translation-invariant kernels. We also noted that the spectrum of the kernel matrix is deterministic and has the form you write down. In particular, the eigenvalues of the kernel matrix have a closed form in terms of the Fourier coefficients of the kernel function itself. Specifically, a version of Theorem 1 in the submission here, applied to kernel ridgeless regression, appears as Lemma 5.2 on page 10 of our arXiv paper  (please see the link below).

We apply these expressions to understand the consistency of kernel interpolation in terms of the Fourier coefficients of the kernel on the circle/torus. We also use these expressions to understand the benefit of regularization in alleviating test error due to noise.

While many of the results in this submission go well beyond our work and this should not detract from the novelty of the paper, our work seems to be quite relevant. It was recently published in the SIAM Journal on Mathematics of Data Science (SIMODS). Please find links to this work here:

arXiv version: https://arxiv.org/abs/2205.13525

Journal version: https://epubs.siam.org/doi/10.1137/22M1499819

Best,

Daniel Beaglehole, Mikhail Belkin, Parthe Pandit

---

> ### Author Response · Authors · 2023-11-23
> **Official comment by authors**
>
> Dear colleagues,
>
> we would like to thank you for pointing our attention to your paper. We find your results very interesting and they are certainly relevant to our work. Additionally, we would like to thank you for acknowledging the novelty of our paper with respect to the related works.  We have added the reference to your paper to our manuscript.

---

### Author Response · Authors · 2023-11-23
**General response**

We are grateful to the reviewers for a high evaluation of our work and many useful suggestions and interesting questions. Following this feedback, we have slightly revised our paper. In addition to various small adjustments and corrections, we addressed the following bigger aspects.

**Numerical experiments** We totally agree with the reviewer wTWc on the role of the experiments in justifying and explaining the theory. In that regard, we have added a new figure 1 to the main paper as well as significantly modified the last figure (figure 3 in the revised version). Also, we have added a new section G to the appendix dedicated to the details and discussion of the experiments. The main goal of figure 1 is to provide a direct experimental verification of Theorem 2 about the equivalence between our two data models and NMNO model. In particular, we have included a new *Cosine Wishart* model which is not covered by our theory but still shows the equivalence to NMNO model.

**Section 5.3. on noiseless observations.**  It appears that the last section was too condensed and hard to read. To remedy this, we revised it by focusing exclusively on the Circle model (the respective Wishart model results are moved to sec. E.3.4.). Additionally, the new version of figure 3 provides an illustration of the phenomena discussed in the section, which is additionally discussed in the new section G.2.

---

### Meta-Review · Area_Chair_hBUG · 2023-12-11

**Metareview:**

This paper provides a theoretical analysis of the generalization error of a family of kernel-based algorithms: spectral algorithms. The paper is notable for its originality and substantial contribution, though it faced initial concerns regarding clarity and lack of experimental validation, which were resolved in the rebuttal stage.

**Justification For Why Not Higher Score:**

The paper's focus on the "spectral algorithms" is highly specialized. While this represents a significant contribution within its domain, it may not have the broad appeal or wide-reaching implications typically associated with papers selected for oral presentations.

**Justification For Why Not Lower Score:**

The overall reviewer enthusiasm supports a spotlight presentation.

---

### Decision · Program_Chairs · 2024-01-16

Accept (spotlight)